# Clonal dynamics and somatic evolution of haematopoiesis in mouse

Chiraag D. Kapadia[1], Nicholas Williams[2], Kevin J. Dawson[2], Caroline Watson[3], Matthew J. Yousefzadeh[4,5], Duy Le[6], Kudzai Nyamondo[2,7], Sreeya Kodavali[2,7], Alex Cagan[2,8], Sarah Waldvogel[1], Xiaoyan Zhang[1], Josephine De La Fuente[1], Daniel Leongamornlert[2], Emily Mitchell[2,7,9], Marcus A. Florez[6], Krzysztof Sosnowski[2,7], Rogelio Aguilar[1], Alejandra Martell[1], Anna Guzman[1], David Harrison[10], Laura J. Niedernhofer[4], Katherine Y. King[6], Peter J. Campbell[2,7], Jamie Blundell[3], Margaret A. Goodell[1 ✉] & Jyoti Nangalia[2,7,9 ✉]

Haematopoietic stem cells maintain blood production throughout life[1]. Although extensively characterized using the laboratory mouse, little is known about clonal selection and population dynamics of the haematopoietic stem cell pool during murine ageing. We isolated stem cells and progenitors from young and old mice, identifying 221,890 somatic mutations genome-wide in 1,845 single-cell-derived colonies. Mouse stem cells and progenitors accrue approximately 45 somatic mutations per year, a rate only approximately threefold greater than human progenitors despite the vastly different organismal sizes and lifespans. Phylogenetic patterns show that stem and multipotent progenitor cell pools are established during embryogenesis, after which they independently self-renew in parallel over life, evenly contributing to differentiated progenitors and peripheral blood. The stem cell pool grows steadily over the mouse lifespan to about 70,000 cells, self-renewing about every 6 weeks. Aged mice did not display the profound loss of clonal diversity characteristic of human haematopoietic ageing. However, targeted sequencing showed small, expanded clones in the context of murine ageing, which were larger and more numerous following haematological perturbations, exhibiting a selection landscape similar to humans. Our data illustrate both conserved features of population dynamics of blood and distinct patterns of age-associated somatic evolution in the short-lived mouse.

The haematopoietic system sustains mammalian life through the generation of oxygenating red blood cells, immune cells and platelets. In humans, blood production accounts for 86% of daily cellular turnover, generating approximately 280 billion cells per day[1]. This process relies on a hierarchy of progenitors that successively amplify cellular output towards differentiated blood cells, in turn derived from a heterogeneous pool of haematopoietic stem cells (HSCs)[2–4].

Like all somatic cells, HSCs accumulate somatic mutations with age[5,6]. In humans, some mutations promote cellular fitness, driving clonal outgrowth[5]. Such 'clonal haematopoiesis', although at very low levels in younger individuals, drives ubiquitous and dramatic loss of clonal diversity in older people[5]. Clonal haematopoiesis is a known risk factor for blood cancers and age-associated non-cancerous disease[7,8].

Whether these patterns of somatic evolution are features of haematopoietic ageing in other species is unknown. In Mammalia, the rate of somatic mutation accrual in colonic epithelium inversely scales with lifespan such that species acquire a similar magnitude of mutations by the end of life[9]. It is unclear if this pattern extends beyond the colon and whether the consequences of somatic evolution over human life scale to shorter-lived species.

The inbred laboratory mouse has been extensively used to study haematopoiesis and establish fundamental tenets of stem cell biology. The most common strain, C57BL/6J, has a median lifespan of 28 months[10], 35-fold shorter than that of humans, and recapitulates many phenotypes of human ageing, with initial data showing a lower rate of clonal haematopoiesis[11]. Here, we study the ontogeny, clonal dynamics and selection landscapes of murine HSCs in vivo to understand the evolutionary processes shaping the maintenance and ageing of blood production.

## Whole-genome sequencing of stem cells

We purified HSCs from three young (3 months) and three aged (30 months) healthy C57BL/6J female mice (Fig. 1a and Extended Data

[1]Department of Molecular and Cellular Biology, Baylor College of Medicine, Houston, TX, USA. [2]Wellcome Sanger Institute, Wellcome Genome Campus, Hinxton, UK. [3]Early Cancer Institute, University of Cambridge, Cambridge Biomedical Campus, Cambridge, UK. [4]Institute on the Biology of Aging and Metabolism, Department of Biochemistry, Molecular Biology and Biophysics, University of Minnesota, Minneapolis, MN, USA. [5]Columbia Center for Translational Immunology, Columbia Center for Human Longevity, Department of Medicine, Columbia University Medical Center, New York, NY, USA. [6]Department of Pediatrics, Division of Infectious Diseases, Baylor College of Medicine, Houston, TX, USA. [7]Wellcome-MRC Cambridge Stem Cell Institute, Jeffrey Cheah Biomedical Centre, Cambridge, UK. [8]Departments of Genetics, Pathology & Veterinary Medicine, University of Cambridge, Cambridge, UK. [9]Department of Haematology, University of Cambridge, Cambridge, UK. [10]The Jackson Laboratory, Bar Harbor, ME, USA. ✉e-mail: goodell@bcm.edu; jn5@sanger.ac.uk

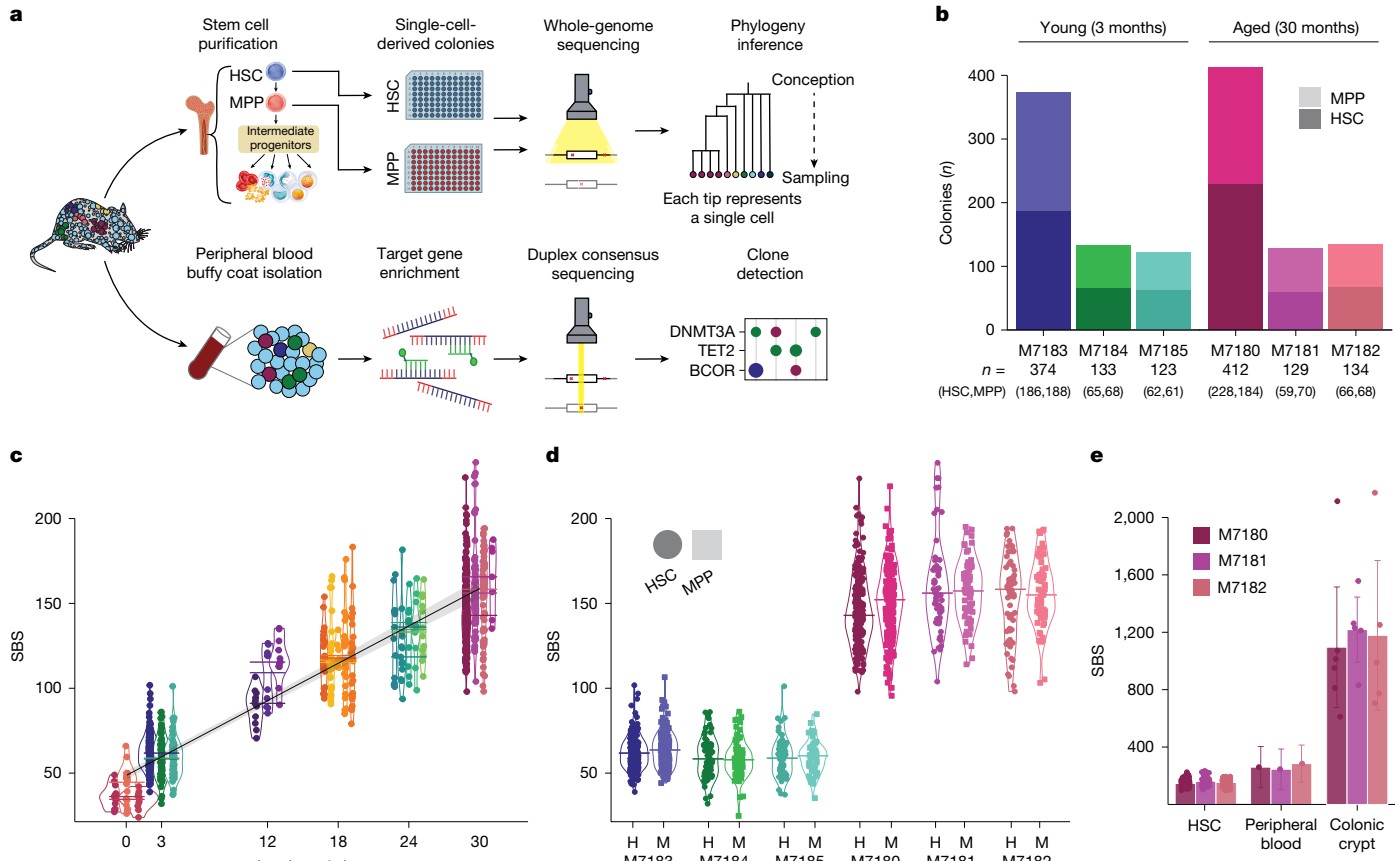

**Fig. 1 | Somatic mutations in murine stem cell-derived haematopoietic colonies. a**, Study approach. Top, single-cell-derived colony WGS of long-term HSCs and MPPs to study somatic mutations, lineage relationships and population dynamics. Bottom, targeted duplex-sequencing of peripheral blood to identify small clonal expansions and fitness landscapes. **b**, Number of whole genomes ($n$ = 1,305) of HSC- and MPP-derived colonies that underwent phylogenetic construction for each female mouse ($n$ = 6). Plots are coloured according to HSC- or MPP-derived colonies, darker and lighter shades, respectively. **c**, Burden of individual SBSs observed in HSCs ($n$ = 908) from each donor. Points denote single HSCs, and horizontal lines denote median sample burden, coloured per sample as in **b**. Line shows linear mixed-effect regression of mutation burden observed in colonies. Shaded areas indicate the 95% CI.

**d**, Comparison of SBS burden between HSC- and MPP-derived colonies from the same mice. SBS burden from HSCs is shown as circles, and burden from MPPs is shown as squares. Horizontal lines depict median burden. The exact number of HSC and MPP colonies for each sample is as listed in **b**. H, HSC; M, MPP, shown above animal ID. **e**, SBS burden across HSCs (data as in **c**), whole blood and individual colonic crypts in the three aged mice. Bar height represents mean SBS burden, and error bars denote 95% CI. Peripheral blood and colonic crypt somatic mutation burdens were measured with nanorate sequencing and WGS, respectively. HSC colony counts per sample are described in **b**; nanorate sequencing results are from a single replicate per animal, and $n$ = 6, 5 and 5 colonic crypts for M7180, M7181 and M7182.

Fig. 1a), ages estimated to be the human lifespan equivalent of approximately 20 and 85–90 years, respectively (Supplementary Note 1). Longstanding views have held that haematopoiesis is supported by long-term HSCs, which give rise to multipotent progenitors (MPPs, or short-term HSCs). Both HSCs and MPPs, distinguished by cell-surface markers, support haematopoiesis and produce all differentiated blood cell types, but only HSCs engraft following serial transplantation[12–16]. To examine both populations in vivo, single HSC- and MPP-derived colonies from 3-month and 30-month animals (Fig. 1a and Extended Data Fig. 1b) underwent whole-genome sequencing (WGS) to an average depth of 14× (61–235 HSCs, 70–191 MPPs per animal; Fig. 1b). We purified HSCs ($n$ = 242) from 17 additional mice aged 1 day to 30 months (9–24 colonies per animal). Following exclusion of 139 colonies (Extended Data Fig. 1c), 1,547 whole genomes (908 HSCs, 639 MPPs) went forward for somatic mutation identification and phylogenetic reconstruction.

## Somatic mutation accumulation

We observed a constant rate of somatic mutation accumulation with age (Fig. 1c). At 3 months, HSC/MPP had an average of 59.5 single-base substitutions (SBSs) (95% confidence interval (CI) 57.3–61.7), with 161.4 SBSs (CI 155.1–167.8) by 30 months. Annually, 45.3 SBSs (CI 42.2–48.4) were acquired, with a somatic mutation occurring every 8–9 days, corresponding to a mutation rate of $8.3 \times 10^{-9}$ per base pair per year (CI $7.7–8.9 \times 10^{-9}$ per base pair per year). Few DNA insertions or deletions were captured with no chromosomal changes observed. Previous studies indicate that MPPs are a more rapidly cycling population[3,17,18] thought to amplify cell production from HSCs, which could result in a greater mutation burden. We observed no difference in mutation burden between HSCs and MPPs (Fig. 1d), consistent with the lack of differences in mutation burdens between human HSCs and differentiated blood cells[19].

The murine HSC SBS rate is approximately three times that of humans (14–17 SBSs per year)[5,6,20,21], consistent with the concept that somatic mutation rates negatively correlate with lifespan[9]. However, the tenfold difference in ultimate mutation burden (approximately 150 in aged murine HSCs versus more than 1,500 in older human HSCs) is greater than the end-of-life somatic mutation burden variation observed across mammalian intestinal crypts[9]. Thus, we wished to validate the lower-than-expected somatic mutation burden observed in aged murine stem cells.

First, we compared genome-wide mutation burdens in HSCs to matched microdissected intestinal crypts from the same aged mice ($n = 16$, range 5–6 per animal; Extended Data Fig. 1d). Colonic epithelium had a higher mutation burden (Fig. 1e), as previously reported[9], confirming that we were not underestimating mutations in HSCs. Next, we undertook nano-error rate whole-genome duplex sequencing[19] of matched blood to orthogonally validate HSC mutation burden. Mutation burden was not statistically different from that of haematopoietic colonies (Fig. 1e) but showed a non-significant trend towards higher mutation burden in whole blood, probably due to the presence of lymphoid cells that have more mutations[22]. Together, these data confirm that somatic mutation rates in blood do not inversely scale with lifespan to the same degree as observed in colon.

## Aetiology of mutational processes in mouse

The pattern of sharing of somatic mutations across individual colonies can be used to reconstruct a phylogenetic tree that depicts their ancestral lineage relationships. We use 'lineage' here to represent the direct line of descent rather than different blood cell types. Figure 2 shows the phylogenetic trees for a 3-month-old mouse and a 30-month-old mouse, with more phylogenies in Extended Data Fig. 2. Tips of the trees are individual HSC- (blue) and MPP-derived colonies (red); branches that trace upwards from tip to root reflect the somatic mutations present in an individual colony and how these mutations are shared across other colonies. Individual branchpoints ('coalescences') represent ancestral cell divisions wherein descendants of both daughter cells were captured at sampling. Colonies that share a common ancestor on the phylogeny represent a clade.

We explored the aetiology of the higher rate of mutation accumulation in murine HSC/MPPs compared to human. DNA replication during cell division is one source of somatic mutations reflecting DNA polymerase base incorporation errors. Nodes on the phylogenies with more than two descendant lineages (polytomies) are evidence of ancestral cell divisions not associated with a somatic mutation and can be used to infer the number of mutations acquired per cell division (Extended Data Fig. 3). We observed 266 lineages by 12 mutations of molecular time in 5 donors that had adequate (more than 10 lineages) diversity. Of the 265 symmetrical self-renewing cell divisions that would have been required, 44 were mutationally silent, providing an estimate of 1.80 (CI 1.46–2.19) mutations per cell division during early life (Extended Data Fig. 3). This is not significantly different from that observed in humans (1.84 mutations per cell division, $P = 0.5$) (ref. 23), indicating that excess mutation accumulation in murine HSCs does not reflect poorer fidelity during DNA replication.

Mutagenic biological processes yield distinguishable patterns of base substitutions at trinucleotide sequence contexts, termed mutation signatures. We identified three mutational processes (Extended Data Fig. 4a and Methods): SBS1, reflecting the spontaneous deamination of methylated cytosines; SBS5, attributed to cell-intrinsic damage and repair; and SBS18, characterized by C>A transversions potentially linked to oxidative damage. As in human HSCs, SBS1 and SBS5 increased with age in keeping with their clock-like nature (Extended Data Fig. 4b). Mutations attributed to SBS18, previously identified in murine colorectal crypts[9], preferentially appeared early in life (Extended Data Fig. 4c,d) reminiscent of their presence in human placenta and human fetal HSCs[23,24]. Overall, the higher relative somatic mutation accumulation rate in mice seems underlaid by context-specific mutational processes (SBS18), together with a higher rate of endogenous DNA damage and/or reduced repair (SBS1/SBS5).

## Parallel establishment of HSCs and MPPs

The classic view of haematopoietic differentiation, based largely on transplantation studies, held that MPPs derived from HSCs[25,26].

Recent barcoding and single-cell approaches in non-perturbed haematopoiesis indicate a more nuanced picture with several self-renewing progenitor populations[2,4]. Using the phylogenies, we retraced stem cell ontogeny in vivo during unperturbed haematopoiesis. We inferred the identity of ancestral branches using the identity of their nearest sibling cell (Supplementary Note 2). Clear vertical bands of HSC-only (blue) and MPP-only (red) ancestral lineages were observed across the trees, representing independent clades (Fig. 2a,b and Extended Data Fig. 2), with only a minority of HSCs arising from MPPs and vice versa. The phylogenetic separation of MPPs and HSCs indicates that most HSCs are derived from HSC self-renewing divisions, and most MPPs are derived from MPP self-renewing divisions, with the two populations independently contributing to blood production in parallel over life (Fig. 2c).

The lack of intermixing between HSCs and MPPs indicates long-term inheritance of an HSC or MPP state. Such stable heritable identity appears by 25 mutations of molecular time, indicating that a substantial proportion of HSC and MPP populations appear and diverge early in life. Around 50 somatic mutations seem to be acquired before birth (mixed effects model intercept 48.2, CI 45.61–50.8). Thus, HSCs and MPPs are established in parallel during fetal development.

To explore when in utero identity was established, we next evaluated somatic mutations present in both HSCs and matched colonic crypts. Blood is mesoderm-derived and colonic epithelium from endoderm; thus, any shared mutations occurred in embryonic cells prior to gastrulation. Mutations on the haematopoietic phylogeny were observed in sampled colonic crypts down to 9–11 mutations of molecular time (Extended Data Fig. 3), timing these shared mutations to have occurred during gastrulation. Indeed, branches with an inferred HSC or MPP identity did not share mutations with the colon, consistent with these lineages being established after germ layer specification.

Given the probable embryonic establishment of distinct HSC and MPP pools, we considered the simplest series of cell state changes (such as HSC to MPP) to capture the observed cell identities. We first considered the prevailing view that MPPs are generated from HSCs. We counted the number of cell identity changes required to reach the sampled cell identity. Surprisingly, the HSC-to-MPP model was equivalently parsimonious (that is, requiring a similar number of cell state changes) to a model where all cells start as MPP (with HSCs arising from MPPs), an ontogeny not generally considered probable (Fig. 2d and Supplementary Note 2). Overall, our data indicate that many long-term HSC and MPP lineages are established independently and in parallel during early development and that MPPs do not always arise from HSCs, contrary to classical views but consistent with recent barcoding evidence[2,4].

## Modelling HSC and MPP transitions

To estimate the rate of HSC–MPP transitions through time, we developed a hidden Markov tree model. We defined three unobservable ancestral states—embryonic precursor cell (EMB), HSC and MPP—and inferred the sequence and transition rates between these states during life. We considered all cells prior to gastrulation (fewer than ten mutations) as EMB and assumed a fixed probability of transitioning out of this state to either HSC or MPP (with subsequent fixed probabilities of further transitions) per unit of molecular time.

By fitting the model to each donor, each age group or the whole cohort, we found that a model fit to young and old mice separately best fit the data (Supplementary Note 2 and Supplementary Table 1). A model in which EMB transitions to either HSC or MPP was significantly better than an 'HSC-first' model, where EMB must transition to HSC prior to MPP specification. However, we were only able to reject an 'HSC-first' ontogeny model in older mice and not younger mice, as our data indicated more frequent HSC-to-MPP transitions earlier in life (Fig. 2e). This difference between young and old mice could be explained if the HSCs that produce MPPs early in life are extinguished by old age and thus are not sampled in the phylogeny.

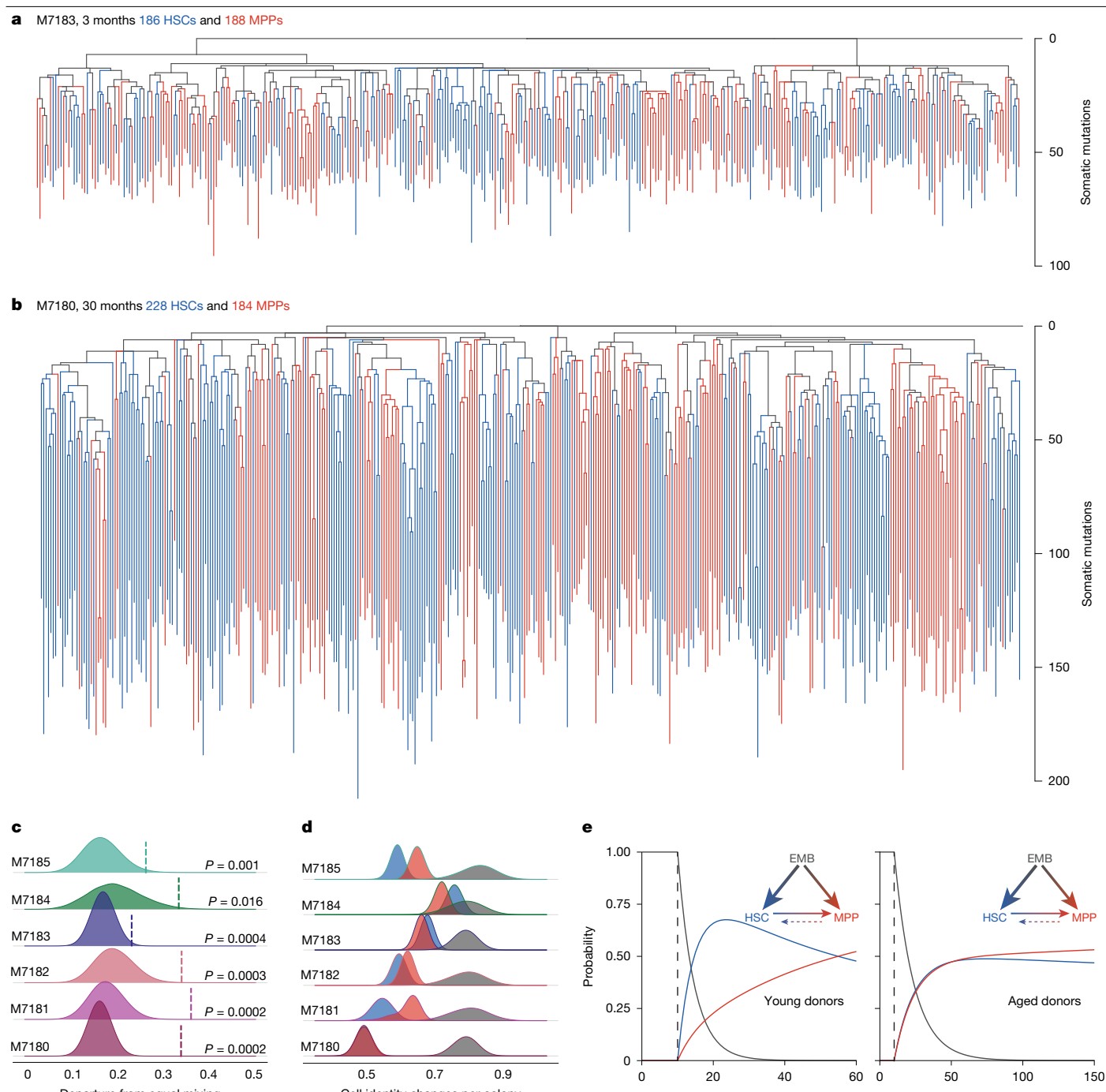

**Fig. 2 | Phylogenetic trees of HSCs and MPPs from a young mouse and an old mouse. a,b,** Phylogenetic trees of a young mouse (3 months; **a**) and an aged mouse (30 months; **b**) depicting the pattern of sharing of somatic mutations among HSC (blue) and MPP (red) colonies. Each tip represents a single colony. Branch lengths represent mutation number, corrected for sensitivity for mutation detection. Branch colours reflect the identity of descendent colonies (Supplementary Note 2). **c,** Phylogenetic relatedness of HSCs and MPPs, quantified as HSC–MPP intermixing within clades established after 25 mutations molecular time. The mixing metric for a clade is the absolute difference between the proportion of HSCs in a clade and the expected value under equal sampling, averaged for all clades in a phylogeny. The vertical bar denotes the observed mixing metric; the filled distributions reflect mixing metrics expected by random chance, estimated by reshuffling the HSC/MPP tip states. One-tailed significance values are derived from the rank of the

observed metric in the corresponding reshuffled distribution, with no adjustment for several comparisons between samples. **d,** Distributions of the number of cell identity changes required per colony to capture the observed tip states. The number of cell identity changes assuming a unidirectional 'HSC-first' model (HSCs give rise to MPPs) is shown in blue. The required cell identity changes for the opposite 'MPP-first' model, in which MPPs give rise to HSCs, is shown in red. The null distribution, in which tip states are randomly reshuffled, is shown in grey. **e,** Cell-type probability trajectories from a three-state ontogeny model in which EMBs differentiate to HSCs or MPPs, followed by HSC-to-MPP or MPP-to-HSC transitions. The displayed trajectories for 30-month (right) and 3-month (left) donors are based on iterating the hidden Markov model starting at EMB. Thickness of arrows reflect the proportion of overall transitions between states; transition rates are derived in Supplementary Note 2.

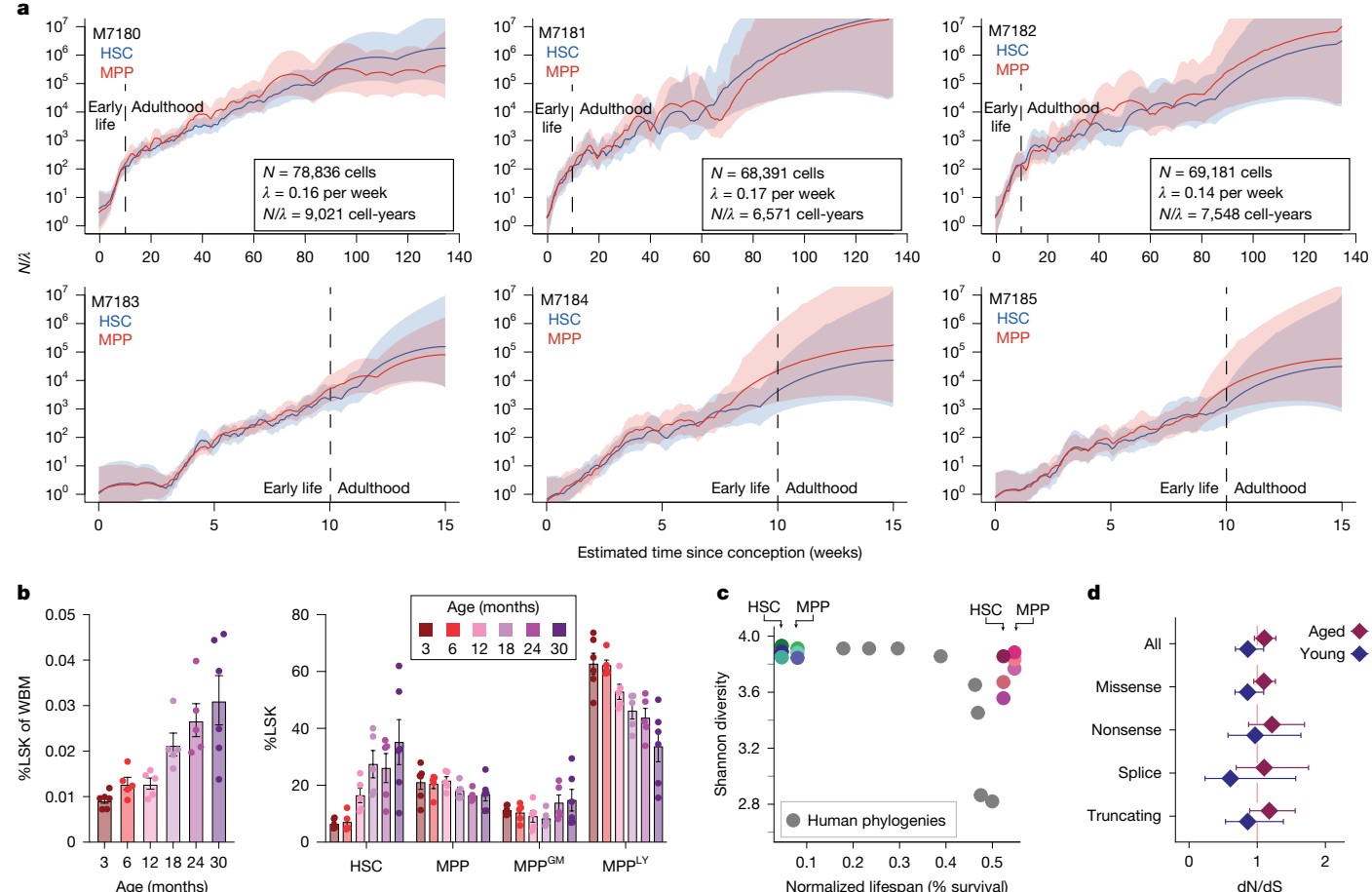

**Fig. 3 | Population dynamics and selection in the murine stem cells.**
**a**, Population trajectories estimated separately in HSCs and MPPs using
Bayesian phylodynamics for the six samples shown in Fig. 2a,b and Extended
Data Fig. 2. The dark blue (HSC) and red (MPP) lines indicate the mean effective
population trajectory; shaded areas are 95% CIs. Vertical dashed lines separate
trajectories into early life and adulthood age periods, in which different
population size behaviours are observed. Inset values indicate posterior density
estimates of population size ($N$), symmetric cell division rate per week ($\lambda$) and
their ratio ($N/\lambda$) in HSC-years, as derived from ABCs. **b**, Haematopoietic stem and
progenitor cell (HSPC) prevalence during murine ageing. The relative abundance
of total HSPCs (left, defined as the LSK compartment) and individual HSPC

subpopulations (right) are compared. Bar height denotes mean among samples
(individual dots); error bars denote s.e.m. Each data point per timepoint is a
biologically independent animal examined over two independent experiments.
MPP$^{Ly}$ are lymphoid-biased progenitors, MPP$^{GM}$ are myeloid-biased progenitors,
based on current immunophenotypic definitions[51]. WBM, whole bone marrow.
**c**, Shannon diversity index for each phylogeny calculated using the number and
size of unique clades present at 50 mutations molecular time. Mouse points
are coloured as in Fig. 1b. Grey dots depict results from data published in ref. 5.
**d**, dN/dS for somatic mutations observed across aged and young animals
overlaps with 1, indicating no departure from neutrality. Error bars denote 95% CI.

Alternatively, the rate of HSCs that transition to MPPs may be greater
earlier in life. Our model indicates that 50% of all HSC and MPP lineages
in young and aged mice had committed to their cell state before 50
mutations of molecular time, probably before birth, confirming our
earlier qualitative observations. As might be expected, HSC-to-MPP
transitions were more frequent than MPP-to-HSC transitions, which
were extremely rare (1 in 1,000 transitions) and within the plausible
limits of cell-sorting accuracies.

## Stem cell population dynamics over life

The pattern of coalescences in the phylogenetic tree reflects the ratio
($N/\lambda$) of population size ($N$) and the rate of HSC self-renewing cell divi-
sions ($\lambda$)—both smaller populations and more frequent cell divisions
shorten the interval between coalescences. Human haematopoietic
phylogenies have a profusion of early coalescences, reflecting rapid
cell division during embryonic growth. Coalescences are then infre-
quently observed during adulthood because of a large stable HSC
population, reappearing in older age in clonal expansions as clonal
diversity collapses[5].

By contrast, young and old murine haematopoietic phylogenies
display coalescences continuing down the tree (Fig. 2a,b and Extended
Data Figs. 2 and 5). Using a phylodynamic Bayesian framework to infer
the HSC population trajectory ($N/\lambda$), we infer an early period of expo-
nential HSC growth followed by progressively increasing $N/\lambda$ over life
(Fig. 3a), consistent with increased HSC numbers with age by flow
cytometry (Fig. 3b) and other studies[27,28]. Our findings contrast with
haematopoietic progenitor population trajectories in humans[5], which
exhibit a population growth plateau during adulthood. Interestingly, we
infer entirely overlapping $N/\lambda$ trajectories for HSCs and MPPs. Together
with their similar mutation burdens and lineage independence, HSC
and MPP clonal dynamics during steady state in vivo haematopoiesis
seem indistinguishable.

We developed an HSC/MPP population dynamics model in which
stem cells grow towards the target population size, taking into account
loss of cells through cell death or differentiation. Using approximate
Bayesian computation (ABC)[29], which generates simulations of phy-
logenetic trees to estimate the most likely posterior distributions of
population size and symmetrical self-renewing division rates (Fig. 3
and Extended Data Fig. 6), we estimate that the murine HSC–MPP

population grows to around 70,000 cells (median 72,414, CI 25,510–98,540). Symmetric cell divisions occur roughly every 6 weeks (median 6.4 weeks, CI 1.8–13.2 weeks). Stem cells exit the population by either death or differentiation about once every 18 weeks (CI 2.3–357 weeks).

## Contribution to progenitors and blood

Given the observed similarities between HSC and MPP, we studied if HSCs and MPPs might differentially contribute to downstream blood cell production. Using further WGS of colonies ($n = 298$) derived from a mixed progenitor compartment including granulocyte/macrophage-biased MPPs (MPP[GM]) and lymphoid-biased MPPs (MPP[Ly]), we expanded the phylogenetic trees of the three aged animals. We observed no preferential bias in MPP[GM] or MPP[Ly] emerging from HSC versus MPP (Extended Data Fig. 7a–e), indicating that HSCs and MPPs produce downstream progenitors at similar proportions. However, this analysis is limited by relatively low numbers of sampled MPP[GM] and MPP[Ly].

To assess lineage contribution to mature blood production, we used deep sequencing of matched blood to measure the fraction of cells harbouring mutations from the phylogenetic trees. Mutations corresponding to both ancestral HSCs and MPPs were recaptured in blood, indicating that both cell types actively contribute to mature blood production. Mutations private to single cells on the phylogeny were highly subclonal (less than 0.1% variant allele fraction (VAF); Extended Data Fig. 8a), in line with each HSC/MPP contributing only a small amount of overall blood production. We observed a slight bias towards increased representation of ancestral MPP lineages compared to HSCs, although this difference was subtle (Extended Data Figs. 8b and 9) and may be due to increased proliferation of MPP descendants or differences in compartment population size earlier in life.

## Absence of large clonal expansions

The absence of expanded clades in murine phylogenetic trees (Fig. 3c) indicates maintenance of clonal diversity, in contrast to the oligoclonality observed in older humans[5] (Extended Data Fig. 5). Population dynamics simulations confidently recapitulated observed phylogenies under a model of neutral growth without positive selection. No colonies ($n = 1,305$) displayed mutations in murine orthologues of human clonal haematopoiesis genes. Among all 88,053 SNVs, the relative rate of non-synonymous mutation acquisition also did not significantly depart from neutrality (Fig. 3d), with no new genes identified as being under selection (Supplementary Data 1).

Given that mutation entry, which furnishes a population with phenotypic variation and substrate for selection, is occurring at a higher rate in mice, we considered reasons for the lack of observable clonal expansions and absence of positive selection on non-synonymous mutations, both of which manifest ubiquitously in human haematopoiesis[5]. One possibility is that there are insufficient HSC and MPP divisions in the short lifespan of mice to facilitate detectable clonal expansions of cells with fitness-inferring mutations. Alternatively, as both population size and the frequency of self-renewing cell divisions (captured in $N/\lambda$) determine the rate of random drift and hence the drift threshold that selection must overcome[30], the fitness ($s$) of newly arising mutations may also be insufficient for their carrier subclones to exceed the genetic drift threshold in a mouse lifespan ($s = \lambda/N$ representing the drift threshold[30]).

In the first scenario, clones under positive selection (that is, with necessary driver mutations) will still be present but would be too small to detect using a phylogenetic approach that only readily identifies larger clones (greater than 5% clonal fraction). In the second scenario, the fitness landscape of any detectable clones would reflect the specific murine haematopoietic drift threshold. We explored both possibilities.

## Native and modulable murine clonal haematopoiesis

We examined murine blood (3–37 months) for very small, expanded clones and the presence of clonal haematopoiesis using targeted duplex-consensus sequencing of murine orthologues of 24 genes associated with human clonal haematopoiesis. Expanded clonal haematopoiesis clones clearly increased in prevalence with age (Fig. 4a,b). Average clone size was very small at 0.017% of nucleated blood cells (range 0.0036–0.27%) representing clonal fractions between 1 in 500 to 1 in 30,000 cells. Clonal expansions harboured mutations in *Dnmt3a* and *Tet2*, genes frequently mutated in human clonal haematopoiesis, but also *Bcor* and *Bcorl1*, observed in humans following bone marrow immune insult[7,8]. These data are consistent with a previous report identifying expanded clones in mice following bone marrow transplantation[11].

Increased clonal prevalence with age was observed across different laboratory strains, including the genetically heterogeneous HET3 strain and at similar clonal fractions (Fig. 4c,d), confirming that small clones with clonal haematopoiesis drivers are not specific to the C57BL/6J strain. Clones were present in biological replicates and persistent in mice sampled longitudinally over four months, although individual clonal dynamics varied (Fig. 4e). Variants displayed enrichment for non-synonymous mutations across these genes with positive selection evident for *Dnmt3a, Bcor* and *Bcorl1* (normalized ratio of non-synonymous to synonymous somatic mutations (dN/dS) > 1, $q < 0.1$) (Fig. 4f). These data indicate that such small clonal expansions in murine blood are under positive selection and not the result of genetic drift.

Laboratory mice are maintained in exceptionally clean and controlled conditions. By contrast, microbial exposures and systemic insults in humans shape the landscape of clonal haematopoiesis[31]. To examine whether the murine haematopoietic selective landscape can be similarly altered, we applied a series of infectious or myeloablative exposures.

We subjected mice to a normalized microbial experience (NME) in which laboratory mice experience several bacterial, viral and parasitic pathogens[32] through exposure to fomite (pet store) bedding. Aged NME-exposed mice displayed an increased burden of somatic clones, especially driven by *Trp53* (Extended Data Fig. 10a). Targeted infection with the single pathogen *Mycobacterium avium*, which has been shown to activate HSCs and lead to chronic inflammation[33], led to increased frequency of *Bcor, Tet2* and *Asxl1* mutant clones (Extended Data Fig. 10b). To observe the effect of myeloablation, aged mice were also treated with chemotherapeutic agents 5-fluorouracil and cisplatin. When treated with cisplatin, we observed globally increased somatic clonal burden (Extended Data Fig. 10c, $P = 0.027$) with increased *Trp53, Tet2* and *Asxl1* and statistically significant positive selection for *Trp53* (Extended Data Fig. 10c), analogous to humans[34,35]. Similarly, 5-fluorouracil exposure led to clones at magnitudes-greater proportions than age-matched controls (Extended Data Fig. 10d). These data illustrate that haematopoietic mutation accrual and selection are sufficient to drive native clonal haematopoiesis in mice, with modulable selection landscapes.

## Fitness landscape of murine clonal haematopoiesis

We wished to understand if the fitness landscape of clonal haematopoiesis drivers in mice influenced the small observed clone sizes (median 0.017%) compared to humans at equivalent lifespan ages. We estimated the fitness landscape of the driver mutations using an established continuous time branching evolutionary framework for HSC dynamics[36].

By analysing the distribution of neutral mutation VAFs (clones at low VAF bearing synonymous or intronic mutations), we can derive independent orthogonal estimates for the underlying effective population size ($N/\lambda$) and mutation rates ($\mu$). We estimated $N/\lambda$ at 16,500 HSC-years (CI 11,122–21,836; Fig. 5a), consistent with estimates from phylogenetic

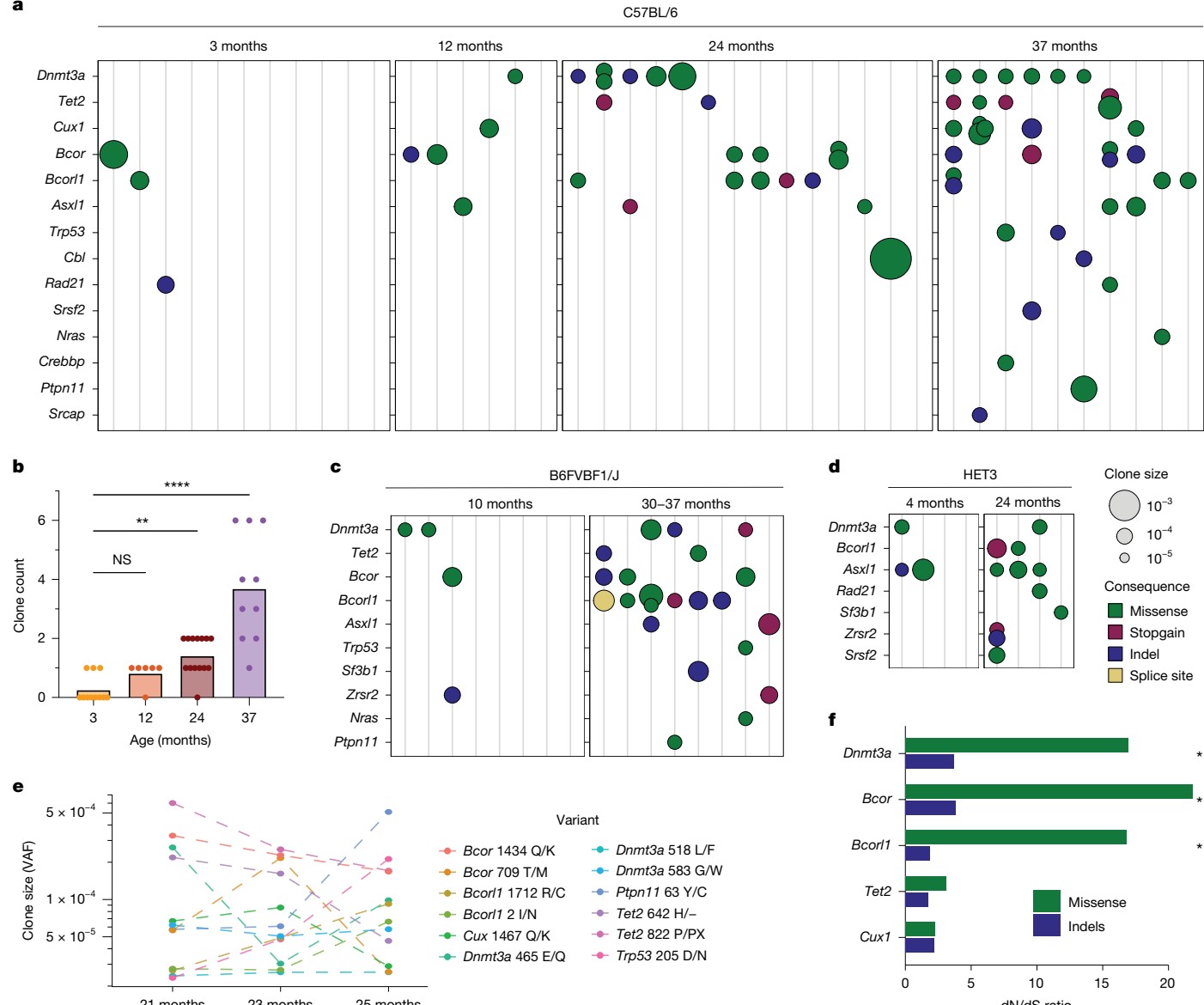

**Fig. 4 | Clonal haematopoiesis during normal ageing in mouse. a**, Dot-plot describing incidence of clonal haematopoiesis in mice at increasing age. Each vertical column represents a single mouse sample with detected clone size and consequence indicated by dot size and colour. Strain is C57BL/6J. **b**, Bar plot summarizing clone count per sample as illustrated in **a**. Bar height represents mean clone count. Differences in clone incidence were quantified by the Kruskal–Wallis test. ** denotes $P = 0.0067$, and **** denotes $P < 0.0001$, with correction for several hypothesis testing. NS, not significant. **c**,**d**, Murine

clonal haematopoiesis incidence in the laboratory strains B6FVBF1/J (F1 hybrid from crossing inbred C57BL/6J × FVB/NJ; **c**) and HET3 (a four-way cross between C57BL/6J, BALB/cByJ, C3H/HeJ and DBA/2J; **d**). **e**, Clone size changes in samples collected serially over 4 months. Clones are coloured by mutation. **f**, dN/dS ratios for targeted genes mutated in murine clonal haematopoiesis. Variants from all donors in **a** were used to determine gene-level dN/dS ratios. * represents dN/dS greater than 1 with $q$ value < 0.1.

trees ($N/\lambda$ 7,918 HSC-years, CI 2,277–20,309). Differences in these estimates are probably influenced by the following: (1) ABC allows for the inferred population growth, whereas the branching evolutionary framework assumes a stable population size; and (2) the branching evolutionary framework model uses the intronic/synonymous mutation rate as the background genome-wide mutation rate, unlike ABC. In the targeted gene panel, the synonymous/intronic mutation rate was estimated at $1.8 \times 10^{-4}$ base pairs (bp) per year (CI $1.2 \times 10^{-4}$–$2.7 \times 10^{-4}$) and the non-synonymous mutation rate at $3.4 \times 10^{-4}$ bp per year (CI $2.9 \times 10^{-4}$–$3.9 \times 10^{-4}$). The total mutation rate, when scaled to total genome size, corresponds to a global mutation rate of $11.77 \times 10^{-9}$ bp per year (CI $9.28 \times 10^{-9}$–$14.94 \times 10^{-9}$). Encouragingly, this is similar to the mutation rate directly observed from the phylogenetic trees ($8.29 \times 10^{-9}$ bp per year, CI $7.73 \times 10^{-9}$–$8.85 \times 10^{-9}$).

Finally, we estimated the distribution of fitness effects driven by non-synonymous mutations. Our analysis indicates that approximately 7% (CI 5–21%) have strong fitness effects (50–200% growth per year) (Fig. 5b). Considering that we infer mouse stem cells to be self-renewing roughly every 6 weeks (CI 2.3–12.5 weeks), an annual growth rate of 200% translates to a per-symmetrical-self-renewing division selective advantage of approximately 15% (5–30%), in line with reported selection coefficients of mutated genes associated with clonal haematopoiesis in humans[5,21,36]. Indeed, in the short-lived mouse, variants with weaker fitness (less than 50%) might have insufficient time to reach exponential, deterministic growth in the population, given that clones are not established until $t_{years} > \frac{1}{s}$ (ref. 37), although any background growth in population size could circumvent this, allowing for weaker variants to fix in the population. These data may also explain

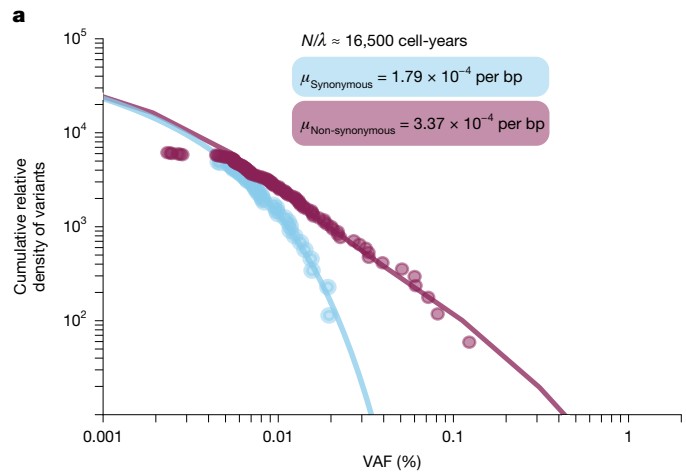

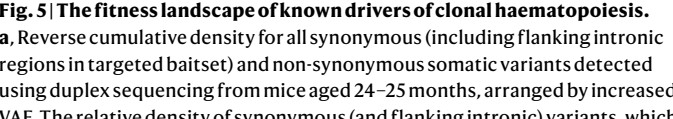

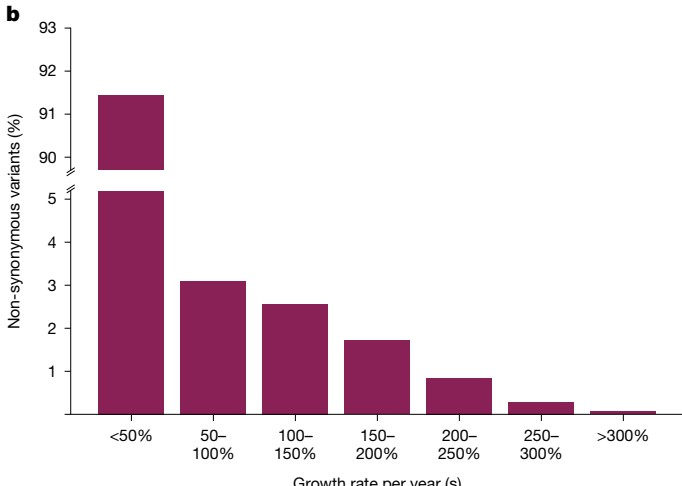

**Fig. 5 | The fitness landscape of known drivers of clonal haematopoiesis.**
**a**, Reverse cumulative density for all synonymous (including flanking intronic regions in targeted baitset) and non-synonymous somatic variants detected using duplex sequencing from mice aged 24–25 months, arranged by increased VAF. The relative density of synonymous (and flanking intronic) variants, which are assumed to have neutral fitness, yields an estimate for $N/\lambda$, the ratio of population size and symmetric cell division rate (per year). The synonymous and non-synonymous mutation rates ($\mu$, bp per year) can then be estimated using a maximum-likelihood approach. **b**, Distribution of fitness effects for non-synonymous mutations.

why some of the low-VAF clones identified by duplex consensus sequencing did not increase in clone size over time (Fig. 4e).

## Discussion

Here, we study the ontogeny, population dynamics and somatic evolution of HSCs in the most widely used mammalian model organism, the laboratory mouse. Our data indicate that HSC and MPP populations are established during embryogenesis, following which they independently self-renew throughout murine life. That MPPs are generated from HSCs in a transplant setting[12–16] may underscore their potential in an experimental setting versus in vivo function. Both MPPs and HSCs contribute to differentiated progenitors and peripheral blood production, with a possible slight bias of contribution from ancestral MPP lineages. These data align with lineage tracing data[2,3] and reports of embryonic MPPs[2,38]. HSCs and MPPs then grow in lockstep over life with indistinguishable clonal dynamics, reaching 25,000–100,000 cells, similar to the human HSC pool size (20,000–200,000 stem cells)[5,20] and reminiscent of the indicated conservation of stem cell numbers across mammalian species[39]. Considering the log-fold difference in body mass and consequent demands on blood production, this similarity may be surprising. In both organisms, but especially the mouse, the number of stem cells exceeds apparent lifetime need; the stem cell compartment of a single mouse can fully reconstitute the blood of approximately 50 transplant recipients[40]. A large stem cell pool may confer an advantage in the face of naturally occurring exposures to environmental pathogens[41] and tissue injury, through increased tolerance to stem cell losses and improved adaptation afforded by somatic cell diversity.

Somatic mutation rates scale inversely with mammalian lifespan in colonic epithelium—mice accumulate mutations 20 times faster than humans, aligned with the difference between their lifespans[9]. Our data show this pattern does not extend to blood; the murine HSC mutation rate is only 3-fold higher than human[5,6,20,21] despite a 35-fold shorter lifespan, indicating tissue-specific evolutionary constraints on mutation rates across species. Indeed, somatic mutation rates in germline cells are lower in mouse than in human[42] and under the influence of distinct factors such as effective population size and age of reproductive maturity[43]. In blood, a low somatic mutation rate may minimize entry of detrimental disease-causing mutations, which, when combined with a large stem cell pool, may also reduce the probability that such mutations become established in the pool. Alternatively, the low HSC mutation rate may simply be a feature of phylogenetic legacy[44].

Patterns of somatic evolution in humans provide one plausible mechanism by which ageing phenotypes occur: the presence of clonal expansions in older human blood driven by somatic mutations is associated with diseases of ageing[8]. However, we only observe small mutation-driven clonal expansions in murine blood by the end of life, indicating that any role age-associated haematopoietic oligoclonality has in human ageing is not shared by the laboratory mouse. The size of clonal expansions is constrained in mice partly because of infrequent HSC self-renewing divisions. Our data fit with mouse stem cells self-renewing every 6 weeks (1.8–13.2 weeks), in the range of previous estimates (every 4–24 weeks)[45–47]. While this is more frequent than human HSCs, which divide 1–2 times a year, for patterns of oligoclonality in humans to be observed in the shorter-lived mouse, stem cells would need to self-renew much more frequently. Long-lived species, such as non-human primates, may more closely resemble human haematopoietic population dynamics[48,49]. Alternatively, mouse strains with higher HSC turnover[50] or exposed to more sustained systemic insults may better mimic human patterns of clonal haematopoiesis. Murine haematopoietic clones expand on systemic exposures, recapitulating patterns observed in human studies[34,35] and following murine transplant[8,31]. Our data indicate a conserved selection landscape in mouse during homoeostatic haematopoiesis and under stress that is driven by mutations in some of the same genes as seen in humans and observable with highly sensitive sequencing. With the evolutionarily conserved constraints on stem cell dynamics in blood, together these drive a distinct pattern of somatic evolution over the murine lifespan. Our data provide a framework for the interpretation of future studies of HSC biology and ageing using the laboratory mouse.

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

## Methods

### Cohort

Wild-type C57BL/6 mice were bred at Baylor College of Medicine or received from the Aged Rodent Colony at the National Institute of Aging (Baltimore, MD). C57BL/6J:FVB/NJ $F_1$ hybrid mice were bred in the Niedernhofer laboratory at the University of Minnesota as previously described[52]. HET3 mice were bred at the Jackson Laboratories as described[53]. C57BL/6 were housed at the AAALAC-approved Center for Comparative Medicine in BSL-2 suites. Experimental procedures were approved by the Baylor College of Medicine or University of Minnesota Institutional Animal Care and Use Committees and performed following the Office of Laboratory Animal Welfare guidelines and PHS Policy on Use of Laboratory Animals. Mice were housed with the same sex in ventilated cages, under a 14/10-h light–dark cycle with temperature and humidity control and enrichment material and fed ad libitum with a standard chow diet. Female mice were chosen for internal consistency and because of their slightly longer lifespan. All mice were nulliparous.

### Haematopoietic progenitor purification

Whole bone marrow cells were isolated from murine hindlimbs and enriched for c-Kit+ haematopoietic progenitors prior to fluorescence-activated cell sorting using a BD Aria II and BD FAC-SDiva software (v.9.0). Whole bone marrow was incubated with anti-CD117 microbeads (Miltenyi Biotec) for 30 min at 4 °C following magnetic column enrichment (LS Columns, Miltenyi Biotec). Progenitor-enriched cells were stained with an antibody cocktail to identify specific progenitor populations using a recent consensus definition[51]. LSKs containing a mixture of stem and progenitor cells were defined as Lineage⁻c-Kit⁺Sca-1⁺ (Lineage⁻ refers to being negative for expression of a set of lineage-defining markers indicated below). HSCs were defined as LSK⁺FLT-3⁻CD48⁻CD150⁺; MPPs were defined as LSK⁺FLT-3⁻CD48⁻CD150⁻. MPP^GM was defined as LSK⁺FLT-3⁻CD48⁺CD150⁻, and MPP^Ly was defined as LSK⁺FLT-3⁺CD150⁻. The gating strategy is illustrated in Extended Data Fig. 1a. This immunophenotypic HSC population includes long-term stem cells with serial repopulating ability, whereas the MPP population is limited to short-term repopulation, as demonstrated in transplantation assays[12–16]. For sorting HSCs from newborn pups, the lineage marker Mac1 was excluded because it is known to be highly expressed on fetal HSCs[54]. Antibodies were c-Kit/APC, Sca-1/Pe-Cy7, FLT-3/PE, CD48/FITC, CD150/BV711, Lineage (CD4, CD8, Gr1, Mac1, Ter119)/eFlour-450 and purchased from BD Biosciences or eBioscience. All antibodies were used at 1:100 dilution for staining, except FLT-3/PE and CD150/BV71, which were used at 1:50 dilution. Flow cytometry data was analysed using FlowJo (v.10.8.1).

### Single-cell haematopoietic colony expansion in vitro

Cell sorting was performed on a BD Arial II in two stages. First, HSCs and MPPs were sorted into separate tubes containing ice-cold fetal bovine serum using the 'yield' sort purity setting to maximize positive cells. Second, the cell populations from stage one were single-cell index-sorted into individual wells of 96-well flat-bottom tissue culture plates containing 100 µl of Methocult M3434 medium (Stem Cell) supplemented with 1% penicillin–streptomycin (ThermoFisher). No cytokine supplements were added to the base methylcellulose medium. Cells were incubated at 37 °C and 5% $CO_2$ for 14 ± 2 days, followed by manual assessment of colony growth. Colonies (more than 200 cells) were transferred to a fresh 96-well plate, washed once with ice-cold PBS and then centrifuged at 800g for 10 min. Supernatant was removed to 10–15 µl prior to DNA extraction on the fresh pellet. The Arcturus Picopure DNA Extraction kit (ThermoFisher) was used to purify DNA from individual colonies according to the manufacturer's instructions; 62–88% of HSCs and MPPs produced colonies, indicating that we are sampling from representative populations in each individual

compartment (Extended Data Fig. 1b). Extracted DNA from each colony was topped off with 50 µl Buffer RLT (Qiagen) and stored at −80 °C.

### Laser capture microdissection

Matched colonic tissue from the three 30-month-old mice used in this study was dissected and snap-frozen at the time of bone marrow collection. Colon tissue sectioning and laser capture microdissection were performed as previously described[55]. In brief, previously snap-frozen colon tissue was fixed in PAXgene FIX (Qiagen) at room temperature for 24 h and subsequently transferred into a PAXgene Stabilizer for storage until further processing at −20 °C. The fixed tissue was then paraffin-embedded, cut into 10-µm sections and mounted on polyethylene naphthalate membrane slides. Staining of histology sections was done using haematoxylin and eosin as previously described[9], with scans of each section captured thereafter. Individual colonic crypts were identified, demarcated and isolated by laser capture microdissection using a Leica Microsystems LMD 7000 microscope (Extended Data Fig. 1d) followed by lysis using the Arcturus Picopure DNA Extraction kit (ThermoFisher).

### Whole-genome sequencing

For low-DNA-input WGS of haematopoietic colonies (from young and aged mice) and colonic crypts (from aged mice), enzymatic fragmentation-based library preparation was performed on 1–10 ng of colony DNA, as previously described[55]. WGS sequencing (2 × 150 bp) was performed at a median sequencing depth of 14× for haematopoietic colonies and 17× for colonic crypts on the NovaSeq platform. Reads were aligned to the GRCm38 mouse reference genome using bwa-mem. For whole-genome single-molecule (nanorate) sequencing, we used matched whole-blood genomic DNA collected from the three aged mice during tissue collection. Nanorate sequencing library preparation was performed as previously described[19], followed by sequencing to 146–153× coverage on the Illumina Novaseq platform.

### Somatic mutation identification and quality control in haematopoietic colonies

Single-nucleotide variants (SNVs) in each colony were identified using CaVEMan[56], including an unmatched normal mouse control sample that had previously undergone WGS (MDGRCm38is). Insertions and deletions were identified using cpgPindel[57]. Filters specific to low-input sequencing artefacts were applied[55]. As variant calling used an unmatched control, both somatic and germline variants were initially called. Germline variants and recurrent sequencing artefacts were then identified using pooled information across mouse-specific colonies and filters as follows: (1) Homopolymer run filter. To reduce artefacts due to mapping errors or introduced by polymerase slippage, SNVs and indels adjacent to a single nucleotide repeat of length 5 or more were excluded. (2) Strand bias filter. Variants supported by reads only in positive or negative directions are probably artefacts. For SNVs, a two-sided binomial test was used to assess if the proportion of forward reads among mutant allele-supporting reads differed from 0.5. Any variant with significantly uneven mutant read support (cutoff of $P < 0.001$) and with over 80% of unidirectional mutant reads was excluded. For each indel, if the Pindel call in the originally supporting colony lacked bidirectional support, the indel was excluded. (3) Beta-binomial filter. Variants were filtered on the basis of a beta-binomial distribution across all colonies, as previously described[9]. The beta-binomial distribution assesses the variance in mutant read support at all colonies for a given mutation. True somatic variants are expected to be present at high VAF (approximately 0.50) in some colonies and absent in others, yielding a high beta-binomial overdispersion parameter ($\rho$). By contrast, artefactual calls are likely to be present at low VAF across many colonies, which corresponds to low overdispersion. The maximum-likelihood estimate of the overdispersion parameter $\rho$ was calculated for each loci. For samples

with more than 25 colonies, SNVs with $\rho < 0.1$ and indels with $\rho < 0.15$ were discarded. For samples with fewer than 25 colonies, SNVs and indels with $\rho < 0.20$ were discarded. (4) VAF filters. Variants with VAF significantly lower than the expected VAF for clonal samples across all mutant genotyped colonies, as assessed with a binomial test with $P$ threshold less than 0.001, were discarded. Additionally, variants with VAF less than half the median VAF of variants that pass the beta-binomial filter were discarded. (5) Germline filter. All sites at which the aggregate VAF is not significantly less than 0.45 are assumed to be germline and discarded. The aggregate VAF is derived from the mutant read count across all colonies for a sample. The binomial test with a confidence threshold less than 0.001 was used to assess departure from germline VAF. (6) Indel proximity filter. SNVs were discarded if they occurred in ten bps of a neighbouring indel. (7) Missing site filter. Loci at which genotype information is unavailable because of poor sequencing coverage will interfere with accurate phylogeny construction. Variants that have no genotype or coverage less than 6× in over one-third of samples were discarded. (8) Clustered site filter. SNVs and indels within 10 bp of a neighbouring SNV or indel, respectively, were filtered. (9) Non-variable site filter. Sites genotyped as mutant or wild type in all colonies do not inform phylogeny relationships and are probably recurrent artefacts or germline variants, and thus were discarded.

Some colonies were excluded on the basis of low coverage or evidence of non-clonality or contamination. Visual inspection of filtered variant VAF distributions per colony was used to identify colonies with mean VAF < 0.4 or with evidence of non-clonality (Extended Data Fig. 1c).

## Mutation burden estimation
Total SNV burden from WGS of individual colonies was corrected for differing depths of sequencing using a per-sample asymptomatic regression fit[55] (Extended Data Fig. 1e). A linear mixed-effect model was used to estimate the rate of mutation acquisition with age, taking into account individual animals as a random effect as follows: burden ~ age + (0 + age|sampleID).

We filtered nanorate sequencing calls as previously described[19], with the following modifications: we excluded variants (1) mapped to the mitochondrial genome, (2) located within 15 bp of sequencing read ends or (3) observed in all duplex consensus reads, as these are probably germline events. Matched colony WGS data were used as a normal control. Mutation burdens were normalized to diploid genome size to determine the global SNV burdens.

## Phylogeny construction and quality control
Phylogenetic trees were constructed on the basis of shared mutations between colonies for each mouse, as extensively described previously[5,21]. The steps, in brief, were as follows: (1) Create genotype matrix. Every colony has high sequencing coverage (median 14×) distributed evenly across the genome, allowing the determination of a genotype for nearly every mutated site observed across colonies. Each locus was annotated as Present, Absent or Unknown in a read-depth-specific manner. The number of unattributable sites was low, allowing precise inferences of colony interrelatedness. (2) Infer phylogenetic tree from genotype matrix. We applied the maximum parsimony algorithm MPBoot to construct phylogenetic trees from the genotype matrix. Only SNVs were used to infer tree topology, but both SNVs and indels (if any) were assigned to inferred branches using treemut. Loci with unknown genotypes in at least one-third of colonies were annotated as missing sites and not used in phylogeny inference. (3) Normalize branch lengths for differing sequencing depth and sensitivity. Branch lengths at this stage are defined by the number of mutations supporting each branch (molecular time). However, each colony has slightly different sequencing coverage, which correlates with differences in mutation detection sensitivity. Thus, we normalized branch lengths on the basis of genome coverage to correct for sensitivity differences across colonies with varying depth, as described in ref. 21 (Extended Data Fig. 1e). (4) Annotate trees with phenotype and genotype information. Each terminal branch (tip) of a tree represents a specific colony. Thus, we annotated each branch of the tree with the sampled cell phenotype (HSC versus MPP).

Tree-level checks were used to identify any discordant branch assignments and assess the validity of tree topology. Any branches supported by variants with mean VAF < 0.4 likely contained contamination by non-clonal variants and indicated that the filtering strategy (see above) was insufficient. Similarly, the branch-level VAF distributions of every colony (tip) in the tree were manually inspected to confirm supporting variants were not present in unrelated portions of the tree (topology discordance). Finally, the trinucleotide spectra of individual somatic mutations were compared between those mutations located on shared branches (that is, mutations supported by more than two colonies) and mutations only observed once and thus present on terminal branches. Mutation spectra were highly similar, indicating that mutations not shared by more than one colony were not populated by a relative excess of artefacts (Extended Data Fig. 1f).

## Population size trajectories
We use the phylodyn package, which uses the density of coalescent events (bifurcations) in a phylogenetic tree to estimate the trajectory of $N(t)/\lambda(t)$ over time[5,20]. Ultrametric lifespan-scaled trees were used to infer chronological timing. Under a neutral model of population dynamics, the phylogeny of a sample is a realization of the coalescent process. In the coalescent process, the rate of coalescent events at time $t$ is proportional to the ratio of population size, $N(t)$, to the birth rate, $\lambda(t)$ (which in the context of stem cell dynamics is the symmetric cell division rate). The sequence of intercoalescent intervals across any time interval $[t_1, t_2]$ is informative about the value of the parameter ratio $N(t)/\lambda(t)$ across the same time interval. We note that only with a constant cell division rate $\lambda$ over time can the trajectory parameter be interpreted as a scalar multiple of the trajectory of population size $N(t)$. Phylodyn assumes isochronous sampling and a neutrally evolving population. We overlaid separate population size trajectories for HSCs and MPPs in Fig. 3a.

## Approximate Bayesian computation
We used inference from phylodynamic trajectories to inform the development of an HSC population dynamics model. Population size trajectories from phylodyn indicated two successive 'epochs' of exponential growth, with some variation in growth rate between epochs and a steady increase in population size over time (Fig. 3a). Given the constraint of tissue volumes, it may be implausible that the HSC population grows constantly. We reconcile this discrepancy by noting that there are very few late-in-life coalescences in our phylogenies and, as a consequence, the estimated phylodyn trajectory in late adulthood is associated with very wide credible intervals. We used a population growth model based on a linear birth–death process[58] (in which a population tends to grow exponentially, subject to stochastic fluctuations), together with a fixed upper bound $N$ on population size. The model assumed a constant birth rate $\lambda$ and constant death rate $v$, with the population trajectory growing at a rate $\lambda - v$ in an epoch. The shape of the trajectory of $N(t)/\lambda(t)$ depends on the cell division rate parameter $\lambda$, not only through the denominator in the ratio $N(t)/\lambda(t)$, but also on $\lambda - v$, through the tendency of the population size $N(t)$ to grow exponentially at a rate $\lambda - v$. In particular, if we increase the fixed upper limit $N$ and at the same time increase the cell division rate $\lambda$, so that their ratio remains constant, the shape of the trajectory of $N(t)/\lambda(t)$ will change as a consequence of the changes in the value of the parameter $\lambda$. This indicates that the parameters $\lambda$, $v$, (in each epoch) and $N$, are all identifiable and so can be estimated separately. The identifiability of $\lambda$, $v$ and $N$ are expanded on in Supplementary Note 4.

We applied Bayesian inference procedures[29] to estimate the parameters ($\lambda$, $v$ and $N$) of the bounded birth–death process. We used ABC. This method first generates simulations of population trajectories and (sample) phylogenetic trees across a lifespan. Each population simulation is run with specific values for the population dynamic parameters drawn from a previous distribution over biologically plausible ranges of parameter values. The ABC method includes a rejection step that retains only those parameter values that generated simulated phylogenies resembling the observed phylogeny (as measured by an appropriate Euclidean distance). The accepted simulations constitute a sample from the (approximate) posterior distribution. Population trajectories and sample phylogenies were simulated using the rsimpop R package. Approximate posterior distributions were computed using the R package abc. We specified uniform joint prior densities for $\lambda$, $v$ and $N$ that encompassed published estimates for $N$ (population size) and $\lambda$ (symmetric division rate)[16,45–47,59]: $N$ ranged from $10^2$ to $10^5$ cells, $\lambda$ ranged from 0.01 to 0.15 cell divisions per day, and $v$ ranged from 0 to $\lambda$, such that the growth rate ($\lambda - v$) is always positive (as observed in the phylodyn trajectories).

Our population dynamics model was a birth–death process incorporating two separate growth epochs. The first (early) epoch lasted until 10 weeks post-conception, and the second (later) epoch lasted from 10 weeks onwards and corresponded to murine adulthood. Inferences were weak for the early epoch; thus, the later epoch was used for parameter inferences. Posterior densities from the three older mice were computed using the 'rejection' method (Extended Data Fig. 6) and pooled to yield parameter estimates and credible intervals.

### Early life polytomy analysis

The polytomies were used to estimate lower and upper bounds for the mutation rate per symmetric division during embryogenesis. The method detailed in ref. 20 was used, whereby the number of edges with zero mutation counts at the top of the tree (up to the first 12 mutations) is inferred from the number and degree of polytomies assuming an underlying tree with binary bifurcations. The mutations per division are assumed to be Poisson distributed. A maximum-likelihood range is then calculated in two steps: first, using the 95% CI of the proportion of zero length edges, with this next leading to a maximum-likelihood estimate for the Poisson rate. Sample M7183 lacked sufficient early life diversity (fewer than 10 unique lineages in 12 mutations molecular time) and was excluded.

### Shared variants between blood and colonic crypts

Mutation genotype matrices (described above) were generated for colonic crypt samples at loci observed in truncal (shared) branches in the matched HSC tree. Every variant was annotated as present or absent for each colonic crypt. We applied two stages to crypt annotation. First, a crypt sample was marked positive if the given variant exceeded a per-sample minimum VAF threshold. The minimum VAF threshold was defined as half the median VAF for all pass-filter colonic crypt variants (as described above). Next, for each variant represented in at least one crypt, any remaining crypt with greater than 2 mutant allele read support was marked positive. This tiered definition allowed for shared variant capture despite differences in coverage among crypt samples. The proportion of a shared variant present among crypt samples was illustrated as a pie chart and annotated to the respective branch of the matched HSC tree (Extended Data Fig. 3).

### Mutational signature analysis

We used the Hierarchical Dirichlet Process (HDP) algorithm to extract mutation signatures across aged and young HSC and MPP colony samples, following the process detailed in ref. 60. Previous work in humans has applied mutation signature extraction to SNVs found only on terminal branches of phylogenetic trees: such terminal branches displayed mutation burdens in excess of 1,000 mutations, depending on the organ. Given the low mutation burden in mouse haematopoietic colonies (terminal branch lengths spanning 30–150 mutations) and thus reduced mutational information, we used all branches with length greater than or equal to 30 mutations as input. To circumvent any bias against shared variants, branches with fewer than 30 SNVs were collapsed to a single 'shared branch' sample. We generated mutation count matrices for each branch, using the 96 possible trinucleotide mutational contexts as input to the R package hdp. HDP was run (1) without priors (de novo), (2) with the reference catalogue of all 79 signatures derived from the PanCancer Analysis of Whole Genomes study (COSMIC v.3.3.1) as priors or (3) with the signatures previously defined as active in mouse colon[9], SBS1, SBS5, SBS18, as priors. Trinucleotide signature definitions were adjusted to mouse genome mutation opportunities before use as priors, and all prior signatures were weighted equally. Signature extraction parameters (1) and (2) produced profiles that did not resemble any existing signatures (cosine similarity < 0.9), probably because of relatively limited SNV burden in mouse colony data. Use of mouse colon signatures as prior information (3) yielded four signature components. Two signature components demonstrated high similarity to SBS1 and SBS5 (cosine > 0.9). The remaining two unknown components were deconvoluted to reattribute their composition to known signatures using the fit_signatures function from sigfit. This yielded three components with a reconstruction cosine similarity metric exceeding 0.99 for similarity to SBS1, SBS5 and SBS18, indicating these three signatures explain the majority of our data (Extended Data Fig. 4a). We surmise the final reattribution step was necessary because of the log-fold lower SNV burdens in mouse blood colonies (30–200 mutations) relative to other tissues examined in previous work (more than 1,000 mutations).

### Branch signature assignment and analyses

For each mouse, we pooled the assigned SNVs into a 'private' or 'shared' category depending on whether the variant maps to a shared branch. Signature attribution to signatures SBS1, SBS5 and SBS18 was then carried out for each of these per mouse category using sigfit::fit_to_signature with the default 'multinomial' model. The per-branch attributions were then carried out by (1) assigning a per-mutation signature membership probability and then (2) summing these signature membership probabilities over all SNVs assigned to a branch to obtain a branch-level signature attribution proportion. The per-mutation signature probability was calculated using

$$P(\text{mutation} \in \text{Sig})$$
$$= \frac{P(\text{mutation} \in \text{Sig})P_0(\text{mutation} \in \text{Sig})}{\Sigma_{\text{Sig}' \in \{\text{SBS1,SBS5,SBS18}\}}P(\text{mutation} \in \text{Sig}')P_0(\text{mutation} \in \text{Sig}')}$$

where the prior probability, $P_0(\text{mutation} \in \text{Sig})$, is given by the mean Sigfit attribution probability of the specified signature, Sig, for the category that the mutation belongs to.

A linear mixed-effect model was used to assess the relationship between age and the signature-specific substitution burden for each colony while accounting for repeated measures. The signature-specific burdens per colony were estimated using a linear mixed model (R package lme4) with age as a random effect and mouse ID as grouping variable:

$$\text{burden}_{\text{signature}} \sim \text{age} + (0 + \text{age}|\text{mouseID}).$$

### Hidden Markov tree approach

**Modelling the ancestral unobserved MPP and HSC states with a hidden Markov tree.** We defined three unobservable ('hidden') ancestral states, EMB, HSC and MPP, and used the observed outcomes (HSC or MPP tip states) to infer the transition probabilities between these identities and the most likely sequence of cell identity transitions during life. The transitions between states are modelled by a discrete

time Markov chain with one step in time representing one mutation in molecular time. We require the root of the tree, presumably the zygote, to start in the 'EMB' state and to stay in that state until ten mutations in molecular time. After ten mutations, the cell then has a non-zero probability of transitioning to another state given by the transition matrix $M$:

$$M = \begin{pmatrix} 1 - p_{\text{HSC->MPP}} & p_{\text{HSC->MPP}} & 0 \\ p_{\text{MPP->HSC}} & 1 - p_{\text{MPP->HSC}} & 0 \\ p_{\text{EMB->HSC}} & p_{\text{EMB->MPP}} & 1 - p_{\text{EMB->HSC}} - p_{\text{EMB->MPP}} \end{pmatrix}$$

This then implies the following transition probabilities for branch $u$ having length $l(u)$ (excluding any overlap with molecular time less than ten mutations), starting in state $i$ and ending in state $k$:

$$P_{i,k}(u) = (M^{l(u)})_{i,k}$$

Now, for a node that is in a specified state, the probability of descendent states is independent of the rest of the tree. This conditional independence property facilitates recursive calculation of a best path ('Viterbi path'), the likelihood of the Viterbi path and the full likelihood of the observed phenotypes given the model. The approach is essentially an inhomogeneous special case of the approach previously described[61].

**Upward algorithm for determining likelihood of the observed states given $M$ and a prior probability of root state π.** The probability of the observed data descendant from a node $u$ whose end of branch state is $i$ is given by

$$P_u(D_u|i) = \prod_{v \in \text{children}(u)} \left( \sum_{k=1}^{S} P_{i,k}(v) P_v(D_v|k) \right)$$

where $S$ is the number of hidden states ($S = 3$ in our use), and $D_u$ denotes the observed data descendant of $u$: that is, the observed tip phenotypes of the clade defined by $u$.

**Initialization of terminal branches.** The probability of observing a matching phenotype is assumed to be

$$P_u(\text{Observed Phenotype of } u = i|i) = 1 - \epsilon$$

The probability of observing a mismatching phenotype, $j \neq i$, is

$$P_u(\text{Observed Phenotype of } u = j|i) = 0.5\epsilon$$

The root probability $P_{\text{root}}(D_{\text{root}}|i)$ is calculated recursively from the above, and the model likelihood is given by

$$P = \sum_{i=1}^{S} \pi_i P_{\text{root}}(D_{\text{root}}|i)$$

Given the two-stage cell sorting approach described above, we assume nearly error-free phenotyping and set $\epsilon = 10^{-12}$.

**Determining the most likely sequence of hidden end-of-branch states.** This Viterbi-like algorithm can be run in conjunction with the upward algorithm. Here, instead of summing over all possible states, we keep track of the most likely descendant states for each possible state of the current node $u$.

The quantity $\delta_u(i)$ is the probability of the most likely sequence of descendant states given that node $u$ ends in state $i$:

$$\delta_u(i) = \prod_{v \in \text{children}(u)} \left( \max_k \{ \delta_v(k) P_{i,k}(v) \} \right)$$

Additionally, for each node, we store the most probable child states given that $u$ is in state $i$:

$$\Psi_{u,v}(i) = \text{argmax}_k \{ \delta_v(k) P_{i,k}(v) \}$$

The tip deltas are initialized using the emission probabilities:

$$\delta_u(i) = P_u(\text{Observed Phenotype of } u|i)$$

The above provides a recipe for recursively finding $\delta_{\text{root}}(i)$ and is combined with prior root probability π to give the most likely root state, $\max_k \{ \delta_{\text{root}}(i) \}$; in our case, we set the prior probability of 'EMB' to unity, so EMB is the starting state. The child node states are then directly populated using $\Psi$.

### Targeted duplex-consensus sequencing

Genomic DNA from freshly collected peripheral blood was purified using the Zymo Quick-DNA Miniprep Plus kit according to the manufacturer's instructions. 1,650 ng of high-molecular-weight DNA was ultrasonically sheared to an average 300 bp fragment size using a Covaris M220 and ligated to duplex identifier sequencing adaptors[62] using the Twinstrand Biosciences DuplexSeq library prep kit. A large input of gDNA was used to ensure that the number of input genomic equivalents (about 275,000–330,000 genomes) did not limit the achievable duplex sensitivity. A custom baitset of biotinylated probes was used to enrich sequences targeting mouse orthologues of common human clonal haematopoiesis driver genes over two overnight hybridization reactions. Our target panel spanned 61.8 kb and captured homologous regions from the entire coding region of the following genes: *Dnmt3a*, *Tet2*, *Asxl1*, *Trp53*, *Rad21*, *Cux1*, *Runx1*, *Bcor* and *Bcorl1* and specific exons with hotspot mutations (as observed in COSMIC) for the following genes: *Ppm1d*, *Sf3b1*, *Srsf2*, *U2af1*, *Zrsr2*, *Idh1*, *Idh2*, *Gnas*, *Gnb1*, *Cbl*, *Jak2*, *Ptpn11*, *Brcc3*, *Nras* and *Kras*. Targeted loci encompass more than 95% of human clonal haematopoiesis events[63] and are described in Supplementary Data 2. Libraries were sequenced on the NovaSeq platform to a raw depth between 1 and 3 million reads, corresponding to duplex-consensus depths between 30,000 and 50,000× that vary across targeted exons. Quality control of duplex sequencing is discussed in Supplementary Note 3.

### Variant identification in targeted gene duplex-consensus sequencing

Duplex-consensus and single-strand consensus reads were generated using the fgbio suite of tools according the fgbio Best Practices FASTQ to Consensus Pipeline Guidelines (https://github.com/fulcrumgenomics/fgbio/blob/main/docs/best-practice-consensus-pipeline.md). To build a duplex-consensus read, we required at least three reads in each supporting read family (that is, at least three sequenced polymerase chain reaction duplicates of matched top and bottom strands from an original DNA molecule). The 'DuplexSeq Fastq to VCF' (v.3.19.1) workflow hosted on DNANexus was also used to generate duplex-consensus reads. Next, VarDict[64] was used to identify all putative variants, followed by functional annotation using Ensembl Variant Effect Predictor[65]. Finally, numerous post-processing filters were applied to remove false positives and artefactual variants: (1) Quality flag filter. VarDict annotates all variants using a series of quality flags that assess mapping and read-level fidelity[64]. Any variant with a quality flag other than 'PASS' was discarded. (2) Read support filter. Duplex sequencing enables detection of somatic variants even from a single read[62]; however, variants supported by a consensus read (singletons) were found to be highly enriched for spurious calls. Thus, any variant supported by a single read was discarded. (3) Mismatches per read filter. Variants were excluded if the mean number of mismatches per supporting read exceeded 3.0. (4) End repair and A-tailing artefact filter. Library preparation enzymatic steps may introduce false-positive SNVs near read ends due to

misincorporation of adenine bases during A-tailing or mistemplating during blunting of fragmented 3′ ends. The fgbio FilterSomaticVcf tool was used to assess the probability that any variant within 20 bp of read ends was due to such enzymatic errors; probable end-repair artefacts were discarded. (5) Read position filter. Variants in positions less than or equal to 15 bp from the 5′ or 3′ end of a consensus read were observed to be enriched for spurious variants based on trinucleotide signature and were discarded. (6) Oxidative damage filter. Mechanical fragmentation (prior to duplex adaptor attachment) creates oxidative DNA damage, often in the form of 8-oxoguanine[66,67], which mispairs with thymine and is fixed after polymerase chain reaction amplification. Any variants fitting the previously described oxidative artefact signature (SBS45) were discarded. (7) Sequencing coverage filter. Variants at loci with duplex depth of less than or equal to 20,000× were considered undersequenced and discarded. (8) Strand bias filter. We used a Fisher's exact test to assess for forward or reverse strand bias between wild-type and mutant reads. Any variant enriched for unidirectional read support was discarded. (9) Recurrent variant filter. Variants present in 5% or more of samples per duplex-sequencing batch or in five or more independent samples were discarded. (10) Indel length filter. Long insertions or deletions could be attributed to poor mapping, erroneous fragment ligation or false-positive calls by VarDict. Any indels greater than or equal to 15 bp were excluded. (11) High-VAF filter. Germline variants display a VAF of 0.5 or 1.0. Any variants with VAF 0.4 or more were excluded as putative germline variants. (12) Impact filter. Clonal haematopoiesis is driven by functional coding sequence changes in driver genes. Thus, synonymous mutations were excluded during generation of the dot-plots in Fig. 4 and Extended Data Fig. 10c. This filter was not used for analyses that require synonymous variant information (dN/dS, fitness effect estimation). (13) Homologous position filter. Residues conserved with humans are likely to be functional in mice. Variants at loci without a matching reference allele at homologous position in humans were discarded. This filter primarily eliminated intronic variants and was not used for analyses incorporating synonymous variant information. Variants identified are detailed in Supplementary Data 2.

### Murine perturbation experiments

Perturbation experiments were initiated in aged (21-month) male and female mice unless otherwise described. Mice were randomly allocated to control or experimental groups. Investigators were not blinded to the group assignment during experiments. For *Mycobacterium avium* infection, mice were infected with $2 \times 10^6$ colony-forming units of *M. avium* delivered intravenously as previously described[68]. Infected and uninfected control mice were housed in a BSL-2 biohazard animal suite following infection. Mice were infected once every 8 weeks (twice in total) to ensure chronic infection. For cisplatin exposure, mice were exposed to 3 mg kg$^{-1}$ cisplatin delivered intraperitoneally every 4 weeks, as indicated. Dose spacing was selected to allow for sufficient recovery following myeloablation and blood counts were not altered in cisplatin-treated mice (Fig. 4c), indicating recovery of haematopoiesis. For 5-fluorouracil exposure, 150 mg kg$^{-1}$ 5-FU was delivered intraperitoneally every 4 weeks two times; this 5-FU dose has previously been shown to drive temporary activation of HSCs in mice[69,70]. Exposure to an NME of murine transmissible pathogens was performed as previously described[71]. In brief, immune-experienced 'pet store' mice were purchased from pet stores around Minneapolis, MN. Aged (24-month) C57BL/6J:FVB/NJF1 laboratory mice were either directly cohoused with pet store mice or housed on soiled (fomite) bedding collected from cages of pet store mice. Mice were exposed to continuous fomite bedding for 1 month, followed by 5 months recovery on specific pathogen-free bedding before tissue collection. All NME work was performed in the Dirty Mouse Colony Core Facility at the University of Minnesota, a BSL-3 facility. Age-matched C57BL/6J:FVB/NJF1 laboratory mice maintained in specific pathogen-free conditions were used as controls. For monitoring, peripheral blood (about 50 μl) was collected in EDTA-coated tubes and analysed on an OX-360 automated hemocytometer (Balio Diagnostics). For all aforementioned mouse cohorts, peripheral blood genomic DNA was purified and converted to duplex sequencing libraries as described above.

Differences in clone burden between control and treated cohorts was quantified using a Mann–Whitney test on cumulative VAFs per sample. Gene-level enrichment was measured using a Mann–Whitney test on the maximum VAF for a given gene per sample, normalized for coverage differences between samples. Gene-level dN/dS estimates were generated as described below.

### dN/dS analysis

dN/dS can be used to assess for selection within somatic mutations by comparing the observed dN/dS to that expected under neutral selection. We use the R package dNdScv[72] to estimate dN/dS ratios of somatic mutations derived from whole-genome and targeted-gene duplex-consensus sequencing. To incorporate mouse-specific differences in trinucleotide context composition and background mutation rates, we generated a murine reference CDS dataset using the buildref function and genome annotations in Ensembl (v.102). For the phylogenetic trees, we input all tree variants to the dndscv function. dN/dS output and all coding variants detected in trees are listed in Supplementary Data 1. To examine dN/dS in targeted duplex-consensus sequencing data, we pooled all variants observed in cross-sectionally sampled mice across ages and ran dndscv limited to exons only included on our targeted panel (Supplementary Data 2).

### Targeted capture of tree variants

We designed a custom targeted DNA baitset (Agilent SureSelect) targeting mutations on the phylogenetic trees of the aged mice and then queried genomic DNA purified from matched peripheral blood for tree-specific mutations using high-depth targeted sequencing. The baitset was designed to capture mutations on the phylogenetic trees of all three aged mice (MD7180, MD7181 and MD7182) and to cover mutations found in HSCs, MPPs and LSKs. The baitset was designed as follows: (1) All variants on shared branches that pass the SureDesign tool's 'moderately stringent filters'. (2) All variants on a random subset of private branches that pass SureDesign's 'most stringent filters'. Approximately 25% of the private branches of each mouse were randomly selected. (3) The exons and 3′ and 5′ UTRs for all clonal haematopoiesis driver genes used in our duplex sequencing panel (listed above). Target-enriched libraries were generated according to the manufacturer's protocol and sequenced using the Illumina Novaseq platform. Baits were sequenced to median depths of 2,616×, 2,549× and 2,628× for MD1780, MD7181 and MD7182, respectively.

To quantify the degree of HSC and MPP contribution to peripheral blood, we estimated the posterior distribution of true VAF for every mutation captured with our targeted baitset. This was done using the Gibbs sampling method previously developed[73]. Then, for each molecular time $t$ and for each branch that overlaps $t$, we estimate the VAF of a hypothetical mutation at time $t$. This is done by arranging our baitset variants in descending estimated VAF order at equally spaced intervals down the branch and then linearly interpolating the VAF at time $t$ based on the estimated VAF of the neighbouring mutations. The aggregate VAF at time $t$ for a tree or lineage is then calculated as the sum of the estimated VAFs of the overlapping branches at time $t$.

### Maximum-likelihood estimates of fitness effects

**Evolutionary framework.** To generate estimates of fitness effects, mutation rates and population size, we applied an evolutionary framework based on continuous time branching for HSCs, as previously reported[36]. The framework is based on a stochastic branching model of HSC dynamics, where variants with a variant-specific fitness effect, $s$, are acquired stochastically at a constant rate $\mu$. Synonymous and non-synonymous mutations detected with duplex sequencing

in untreated 24- to 25-month-old mice were used in the analysis. Synonymous and non-synonymous mutations were considered independently. Synonymous mutations are assumed to have no fitness effect and reflect behaviour under neutral drift, whereas non-synonymous mutations were hypothesized to reflect behaviour under a positive selective advantage. The density of variants declined at VAF $5 \times 10^{-5}$, so to only include VAF ranges supported by informative variants, only variants above this threshold were included in maximum-likelihood estimations described below.

How the distribution of VAFs predicted by our evolutionary framework changes with age ($t$), the variant's fitness effect ($s$), the variant's mutation rate ($\mu$), the population size of HSCs ($N$) and the time in years between successive symmetric cell differentiation divisions ($\tau$) is given by the following expression for the probability density as a function of $l = \log(\text{VAF})$:

$$\rho(l) = \frac{\theta}{(1-2e^l)}e - \frac{e^l}{\varphi(1-2e^l)} \text{ where } \theta = N\tau\mu \text{ and } \varphi = \frac{e^{st}-1}{2N\tau s}$$

The value of $\varphi = \frac{e^{st}-1}{2N\tau s}$ is the typical maximum VAF a variant can reach and this increases with fitness effect ($s$) and age ($t$). To reach VAFs greater than $\varphi$ requires a variant to both occur early in life and stochastically drift to high frequencies, which is unlikely. Therefore, the density of variants falls off exponentially for VAFs greater than $\varphi$. For neutral mutations ($s = 0$),

$$\varphi = \frac{t}{2N\tau}$$

Because the mouse age $t$ is known and the neutral $\varphi$ is measurable from the data, the ratio $\varphi/t$ allows us to infer $N\tau$ from the distribution of neutral mutation VAFs. Because the neutral $\theta$ is measurable from the data and $\theta = N\tau\mu$, we can also infer the neutral mutation rate ($\mu$).

Probability density histograms, as a function of log-transformed VAFs, were generated using Doane's method for log(VAF) bin size calculation. Densities were normalized by the product of bin sample size and width. Estimates for $N\tau$ and $\mu$ were inferred using a maximum-likelihood approach, minimizing the L2 norm between the cumulative log densities and the predicted densities. For synonymous mutations, maximum-likelihood estimates were optimized for $N\tau$ and $\mu$. For non-synonymous mutations, variants with VAFs below the observed maximum synonymous VAF ($1.99 \times 10^{-4}$) were used—these variants are in the 'neutral' range—and estimates were optimized for with the $N\tau$ estimated from synonymous mutations.

**Differential fitness effects.** We estimated the distribution of fitness effects across non-synonymous variants using our derived estimates of $N\tau$ and non-synonymous $\mu$. We parameterized the distribution of fitness effects using an exponential power distribution, which captures a strongly decreasing prevalence of mutations with high fitness:

$$\mu_{\text{non-neutral}}(s) \propto \exp\left[-\left(\frac{s}{d}\right)^\beta\right]$$

The shape of the distribution was fixed to $\beta = 3$ (ref. 74). Using the VAF density histograms from non-synonymous variants, we estimated the scale of the distribution and non-neutral mutation rate: $\int_{s=0}^{\infty}\mu_{\text{non-neutral}}(s)ds$. The maximum-likelihood fit predicted a scale of about $d = 2$ and the proportion of non-neutral non-synonymous mutations to be about 12% (Fig. 5b).

**Statistics and reproducibility**
For all presented data, the sample size $n$ represents the number of biologically independent samples. Individual data points are displayed and represent independent biological replicates. For whole-genome-sequence experiments, colonies from at least three independent biologically replicate mice were queried at each age timepoint. For duplex experiments, each individual mouse queried is an independent biological replicate. In perturbation experiments, displayed data are aggregated from at least two independent replicate treatment cohorts. Mice were randomly allocated to control or experimental groups. Investigators were not blinded to the group assignment during experiments. Replicate experiments were designed identically. All replicate experiments were successful. All statistical analysis was performed using GraphPad Prism (GraphPad Software, v.10) or R (v.4.3) unless otherwise specified. The statistical tests used are detailed in the figure legends. Measure of centre and error bars are described in the figure legends.

**Reporting summary**
Further information on research design is available in the Nature Portfolio Reporting Summary linked to this article.

## Data availability
WGS data has been deposited at the European Nucleotide Archive at accession numbers ERP138320, ERP152795 and ERP144323. Targeted duplex sequencing data has been deposited at NCBI BioProject PRJNA1033340.

## Code availability
SNVs and indels were detected using CaVEMan (v.1.14.0, https://github.com/cancerit/CaVEMan), cgpPindel (v.3.9.0, https://github.com/cancerit/cgpPindel) and VarDict (v.1.8.3, https://github.com/AstraZeneca-NGS/VarDictJava). Variants were annotated using VAGrENT (v.3.7.0, https://github.com/cancerit/VAGrENT) and Ensemble VEP (v.107-110.0, https://github.com/Ensembl/ensembl-vep). Phylogenies were constructed using MPBoot (v.1.1.0, https://github.com/diepthihoang/mpboot). Variants were assigned to phylogenies using Rtreemut (https://github.com/nangalialab/treemut). Population trajectories were inferred using phylodyn (https://github.com/mdkarcher/phylodyn). Bayesian inferences used the packages rsimpop (https://github.com/nangalialab/rsimpop) for simulations and abc (v.2.2.1, https://CRAN.R-project.org/package=abc) for ABC. Mutation signatures were inferred using hdp (https://github.com/nicolaroberts/hdp) and sigfit (v.2.2.0, https://github.com/kgori/sigfit). Duplex consensus reads were generated using the fgbio suite of tools (v.1.5.1–2.1.0, http://fulcrumgenomics.github.io/fgbio/). dN/dS ratios were calculated using dNdScv (v.0.1.0, https://github.com/im3sanger/dndscv). Population genetic analyses of clone sizes and parameter inferences were based on code available at https://github.com/blundelllab/ClonalHematopoiesis/. Other analyses were carried out using custom R scripts available at https://github.com/CDKapadia/somatic-mouse.

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

**Acknowledgements** C.D.K. is supported by grant no. F30DK131638 and is a visiting scholar at the Wellcome Sanger Institute. The Goodell lab is supported by grants from the National Institutes of Health, including grant nos. AG036695, CA183252, CA237291, DK092883, 1P01CA265748, F30HD111129 (S.W.) and the Milky Way Research Foundation. J.N. is supported by a Cancer Research UK Advanced Clinical Fellowship, and work in the Nangalia lab is supported by Wellcome, Cancer Research UK, Alborada Trust, Blood Cancer UK, Leukemia and Lymphoma Society, and the WBH Foundation. L.J.N. is supported by NIH grant nos. AG063543 and AG056278. The Niedernhofer lab is supported by grant nos. AG063543 and AG063543-02S1. The Harrison lab is supported by grant no. 5U01AG022308. D. Le, M.A.F. and K.Y.K. were supported by grant nos. R35 HL155672 (K.Y.K.), R01 AI141716 (K.Y.K.), F31 HL154661 (D. Le), F31 HL156500 (M.A.F.) and a minority graduate fellowship from the American Society of Hematology (M.A.F.). K.N. is supported by the Wellcome Trust and CRUK. Adobe Stock library images used in illustrations were covered under a Baylor College of Medicine education license. We are grateful for the assistance of R. D. O'Kelly and M. Pierson in conducting NME experiments and the CASM Support team at the Wellcome Sanger Institute. We thank E. Laurenti, S. Loughran and A. R. Green for constructive discussions and J. T. Gebert and H. L. Chan for the critical feedback.

**Author contributions** C.D.K., J.N. and M.A.G. designed the experiments. J.N. and M.A.G. supervised the project. C.D.K. performed cell sorting and in vitro culture with support from S.W., X.Z., R.A., A.M., A.G., E.M., S.K. and K.S. C.D.K. performed genomic, phylogenetic, signature and population dynamics analyses with support from N.W., K.J.D., D. Leongamornlert, J.D.L.F., E.M., P.J.C., M.A.G. and J.N. N.W. developed hidden Markov modelling, and K.J.D. performed population dynamic inferences. A.C. and K.N. prepared colonic crypt microdissections. S.K. and K.S. assisted with colony culture. C.D.K. performed mouse experiments with advice and assistance from M.J.Y., S.W., K.N., A.C., D. Le, M.A.F., R.A., A.M., A.G., D.H., K.Y.K. and L.J.N. C.W. and J.B. developed the population genetic analyses of clone sizes and parameter inferences in Fig. 5. C.D.K., J.N. and M.A.G. wrote and edited the manuscript. All authors reviewed and edited the manuscript.

**Competing interests** The authors declare no competing interests.

**Additional information**
**Correspondence and requests for materials** should be addressed to Margaret A. Goodell or Jyoti Nangalia.

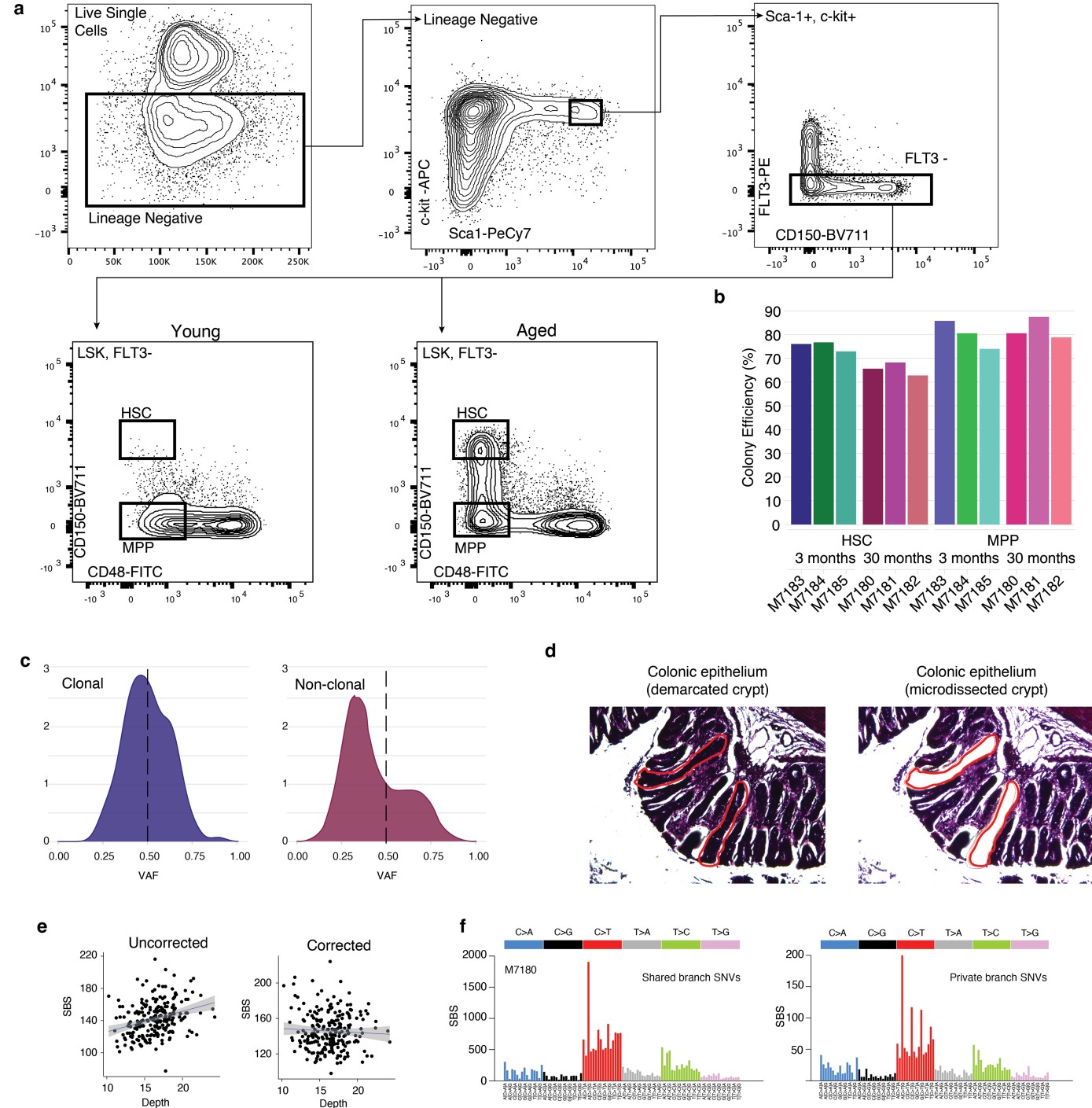

**Extended Data Fig. 1 | Cell isolation strategy and quality control. a)** Sorting strategy for single HSCs and MPPs from young and aged mice. Progenitor-enriched bone marrow was stained as described in the Methods, and then single cells were sorted into individual wells for in vitro expansion. **b)** Colony-forming efficiency of sorted HSCs and MPPs for each sample. Each bar represents the listed cell type and underlying sample ID. **c)** Variant allele fraction (VAF) distribution of all variants within a colony that pass filtering, shown for a representative clonal colony that passed sample QC (left) and a non-clonal colony that passed sample QC (right). After variant filtration, the VAF distribution of a

colony's variants is centred around 50% in clonal colonies, but less than 50% in non-clonal colonies. **d)** Representative image of two colonic crypts isolated by laser capture microdissection. **e)** Correlation between total single base substitution burden and depth, for all colonies from sample M7180, shown before (left) and after (right) sequencing depth correction. Shaded area denotes 95% confidence interval. **f)** Trinucleotide spectra from aggregated somatic mutations mapped to shared (truncal) or private branches of phylogenetic trees. Signatures are highly similar, suggesting artefacts are not relatively enriched in either portion of reconstructed trees.

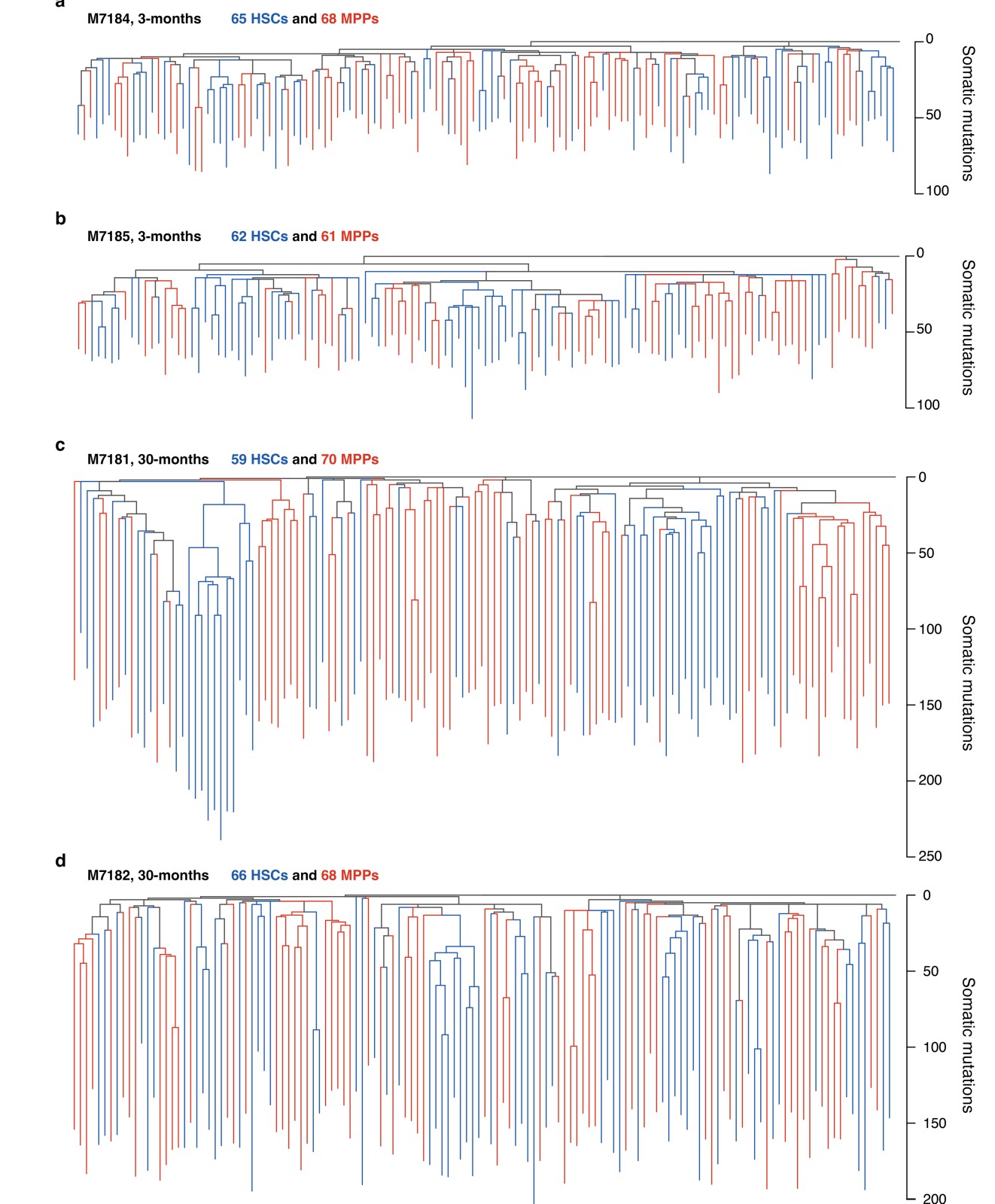

**Extended Data Fig. 2 | Additional phylogenetic trees from young and aged mice.** Phylogenies for **a-b**) 2 additional young (3-month) mice and **c-d**) 2 additional aged mice (30-month), presented as described in Fig. 2.

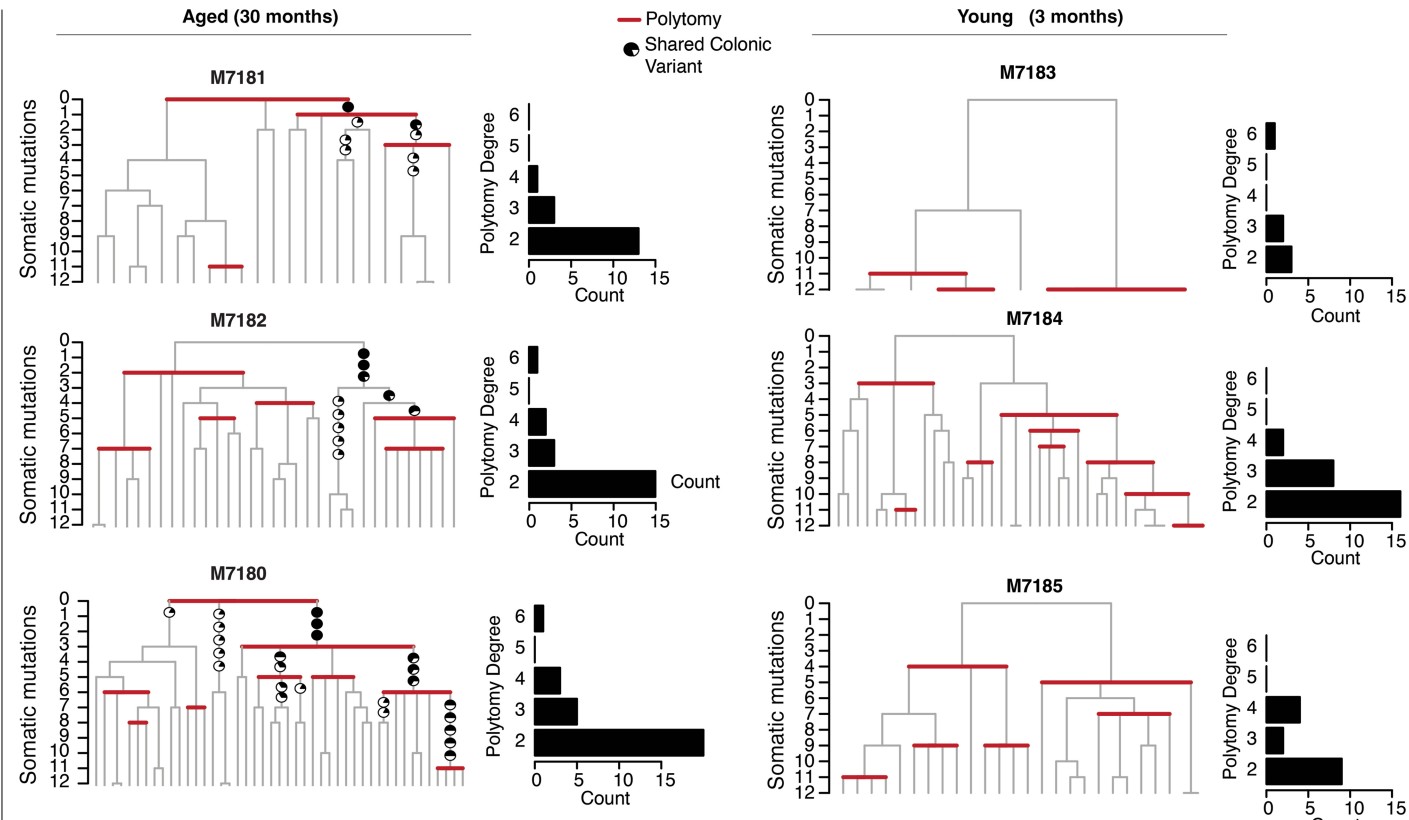

**Extended Data Fig. 3 | Early-in-life phylogenetic patterns and cross-tissue mutations.** Phylogenies from aged (left) and young (right) HSCs zoomed into the first 12 mutations molecular time. Polytomies in the branching structure, which represent cell division without mutation acquisition, are enriched among early-in-life cell divisions at the tops of the phylogenies. Variants shared with matched colonic crypts are layered onto the trees as pie charts. Pie chart fullness represents the proportion of colonic crypts in which the mutation present on the haematopoietic phylogeny was observed. Sample M7183 lacked sufficient early life diversity (<10 unique lineages within 12 mutations molecular time) and was excluded.

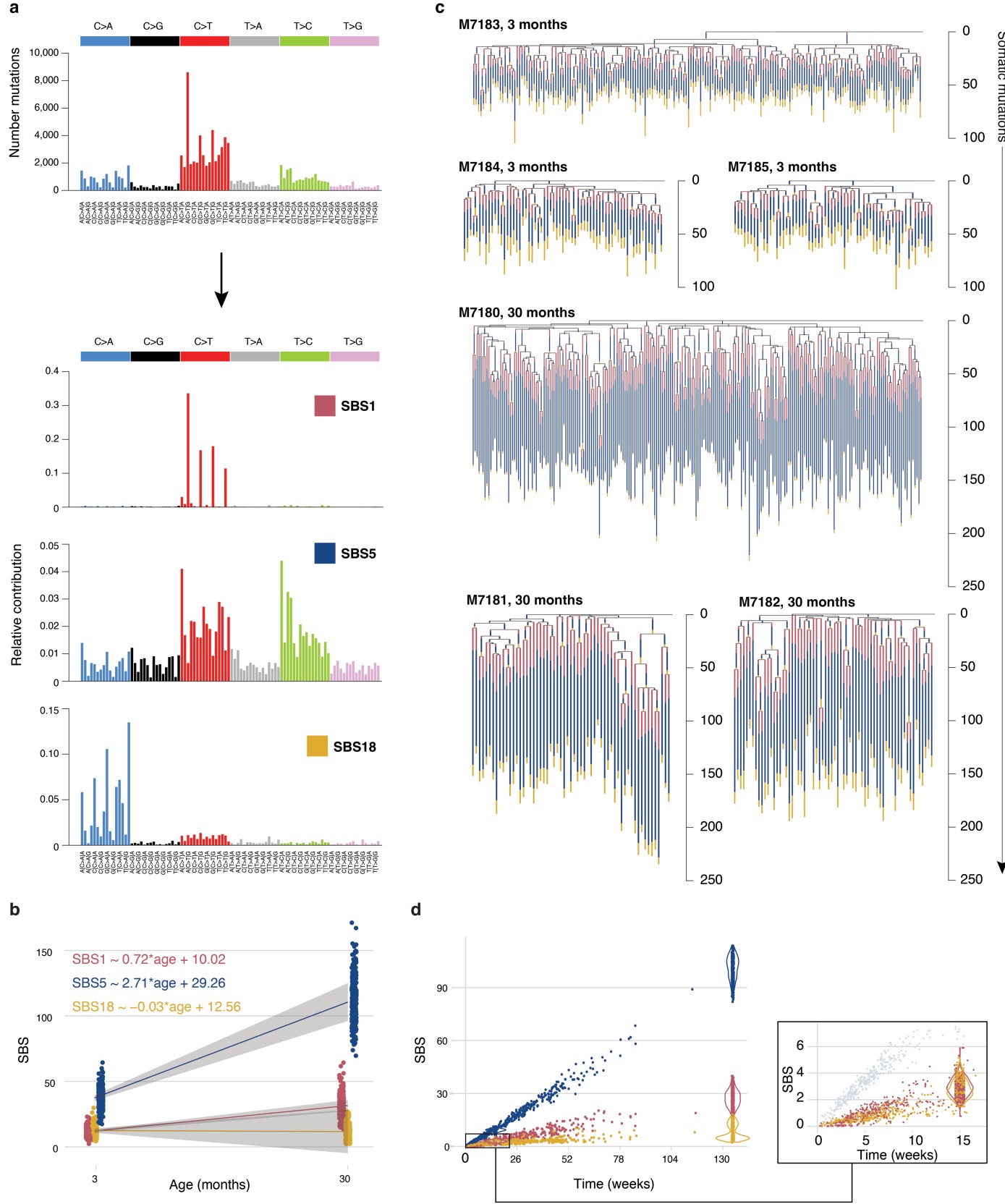

**Extended Data Fig. 4 | Mutational processes in murine stem cells. a)** Signature extraction overview. Trinucleotide spectra from all single-base substitutions (SBS) (top), were used for signature extraction as described in the Methods. Three signatures identified as SBS1, SBS5, and SBS18 best described the catalogue of mutations observed (cosine similarity=0.997). **b)** Linear mixed-effect regression of signature-specific mutation burdens observed in colonies. Shaded areas indicate the 95% confidence interval. **c)** Signature attribution in phylogenies. Individual branches of HSC phylogenies are overlaid with signature contribution proportions. SBSs assigned to each branch were fit to SBS1, SBS5 or SBS18. **d)** Signature-specific mutation accumulation in all branches across phylogenies. Early-life branchpoints, located at the top of a given phylogeny, and shown as an inset.

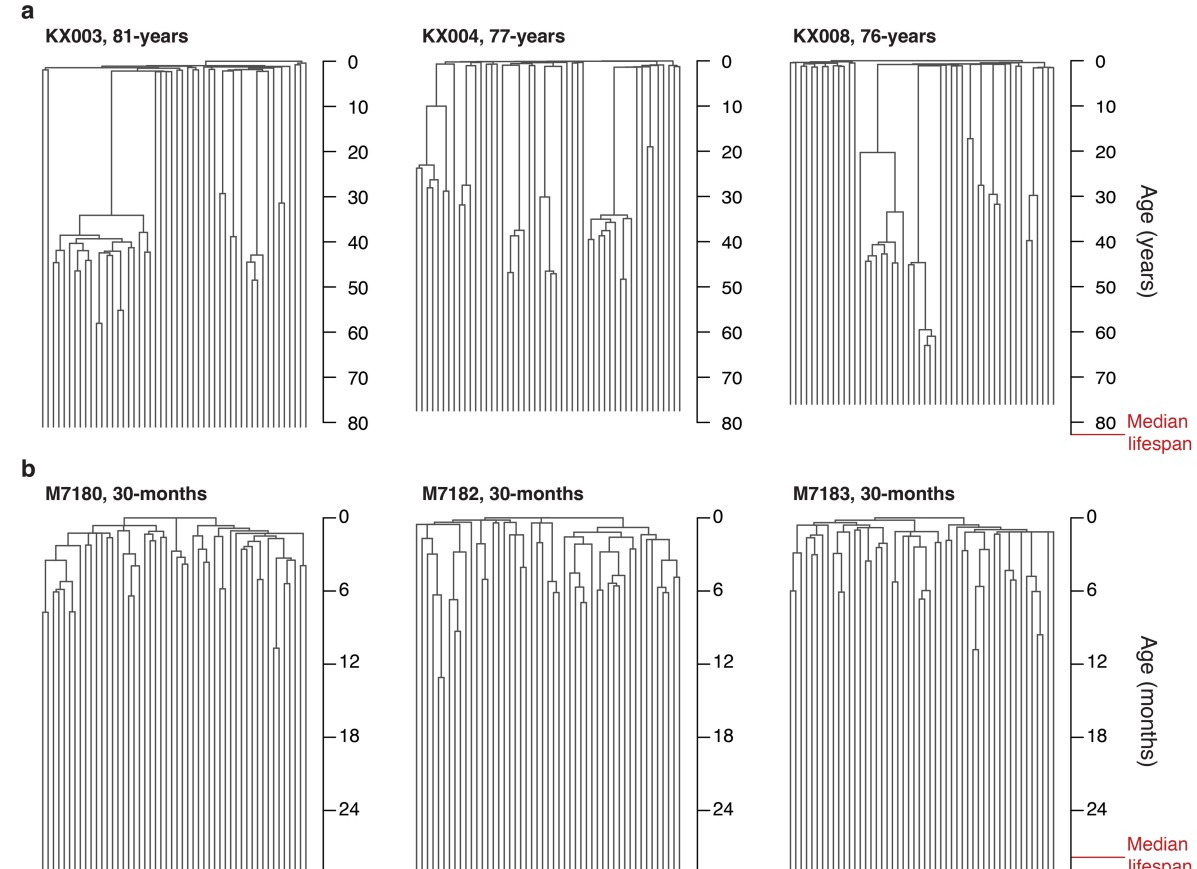

**Extended Data Fig. 5 | Phylogeny comparison between aged human and mouse. a**) Representative ultrametric phylogenies from the three oldest humans described in Mitchell et al.[5]. The published trees have been randomly downsampled to 100 colonies (tips). **b**) Aged mouse phylogenies, also downsampled to 100 colonies, to allow comparison of topological structure. The median lifespan for human and mouse species are labelled and were derived as described in Supplementary Note 1. Full murine phylogenetic trees are shown in Fig. 2a,b and Extended Data Fig. 2.

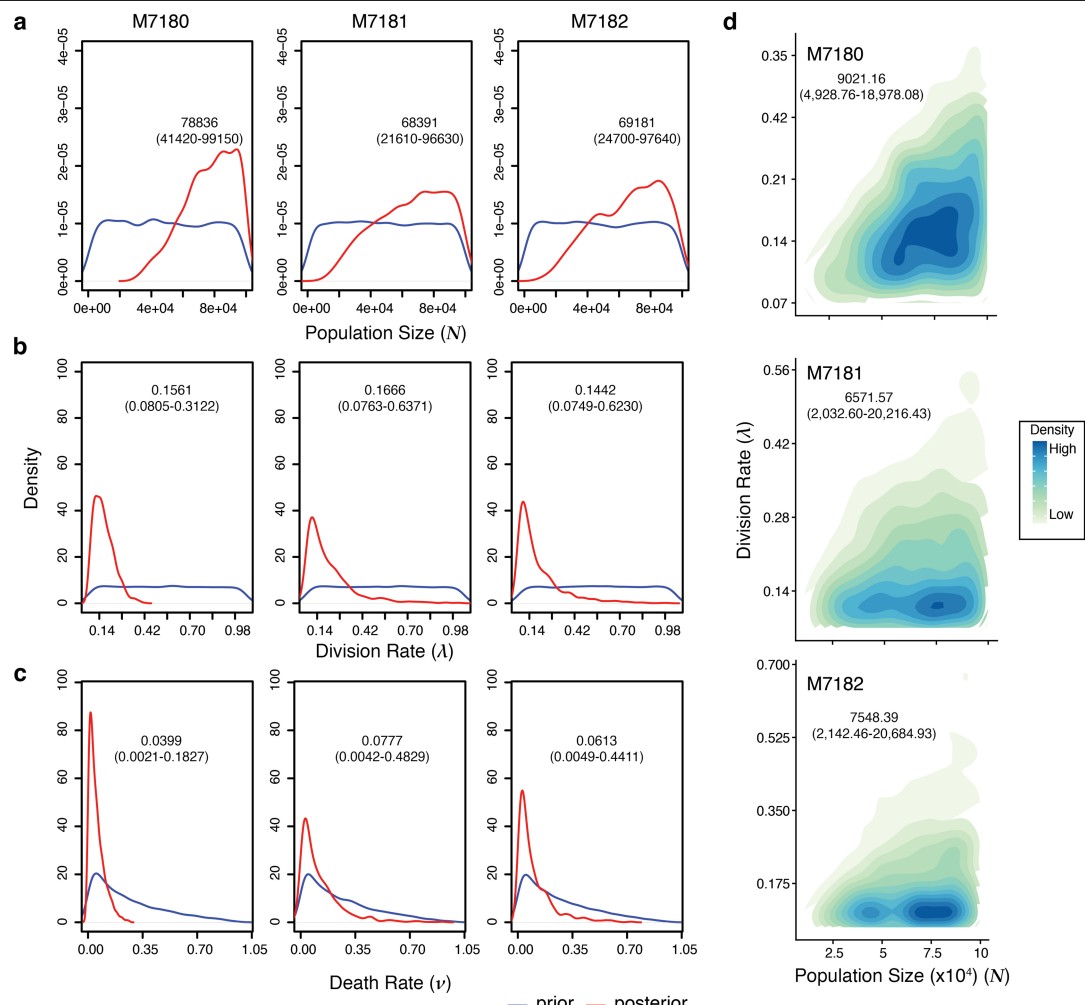

**Extended Data Fig. 6 | Approximate Bayesian inferences.** Results from approximate Bayesian computation (ABC) inference of **a**) population size (*N*), **b**) symmetric division rate per week (*λ*), and **c**) death rate per week (*ν*) for the three 30-month-old mice. Blue lines represent the prior density of parameters; red lines represent the posterior densities. Median posterior density estimates and 95% credibility intervals are displayed for each parameter per sample.

The prior density for the death rate was bounded to ensure the growth rate (*λ – ν*) remained positive, as observed in *phylodyn* trajectories in Fig. 3. **d**) Joint density distributions indicating optimal parameters of population size and division rates that explain observed phylogenetic trees. The estimated *N/λ*, in HSC-years, is shown with 95% credibility intervals. Data from the three aged mice are shown.

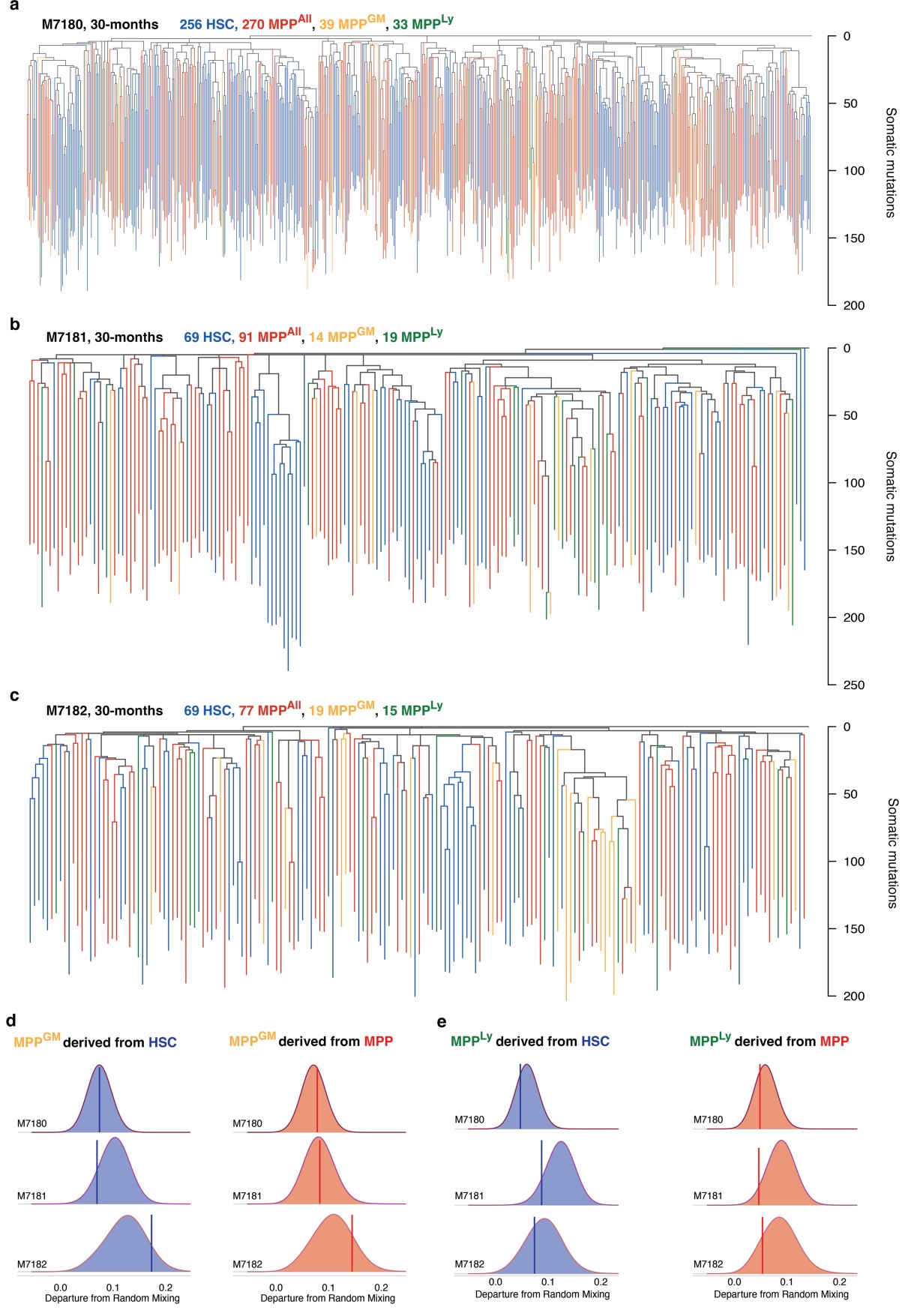

**a**

M7180, 30-months 256 HSC, 270 MPP^All, 39 MPP^GM, 33 MPP^Ly

**b**

M7181, 30-months 69 HSC, 91 MPP^All, 14 MPP^GM, 19 MPP^Ly

**c**

M7182, 30-months 69 HSC, 77 MPP^All, 19 MPP^GM, 15 MPP^Ly

**d**

MPP^GM derived from HSC MPP^GM derived from MPP

**e**

MPP^Ly derived from HSC MPP^Ly derived from MPP

**Extended Data Fig. 7** | See next page for caption.

**Extended Data Fig. 7 | Extended phylogenetic trees including early progenitors. a-c)** Extended phylogenies for three 30-month mice using the pattern of sharing of somatic mutations among HSCs (blue), MPPs (red), and the mixed LSK (Lineage-, Sca1+, c-kit+) haematopoietic progenitor compartment. The LSK compartment contains HSCs and MPP, and additionally contains the myeloid-biased MPP[GM] (orange) and lymphoid-biased MPP[Ly] populations (green). LSK subcompartments were identified at time of single cell sorting using a consensus definition[51]. Each tip represents a single colony. Branch lengths represent mutation numbers. **d-e)** Clade mixing metrics for MPP[GM] and MPP[Ly] colonies used to evaluate interrelatedness with HSC and MPP. HSC, MPP and MPP[GM] or MPP[Ly] were designated as being in the same clade if they shared a most recent common ancestor after 25 mutations, corresponding to early foetal development. Only clades with more than 3 colonies are considered. The vertical bar reflects the average clade mixing metric observed in the constructed phylogenies, while distributions reflect the average clade mixing metric expected random chance, estimated by reshuffling the tip states. If the observed value (vertical bar) significantly deviated from random chance (filled distribution), then there would be minimal overlap between the observed data and the random reshuffling distribution. The average clade mixing metric for MPP[GM] compared to HSCs (blue) and MPPs (red) is shown in d). The similar measure of interrelatedness of MPP[Ly] to HSCs and MPPs is shown in e). All one tailed significance values in d) and e) were $p > 0.05$ and were derived from the rank of the observed metric in the corresponding reshuffled distribution.

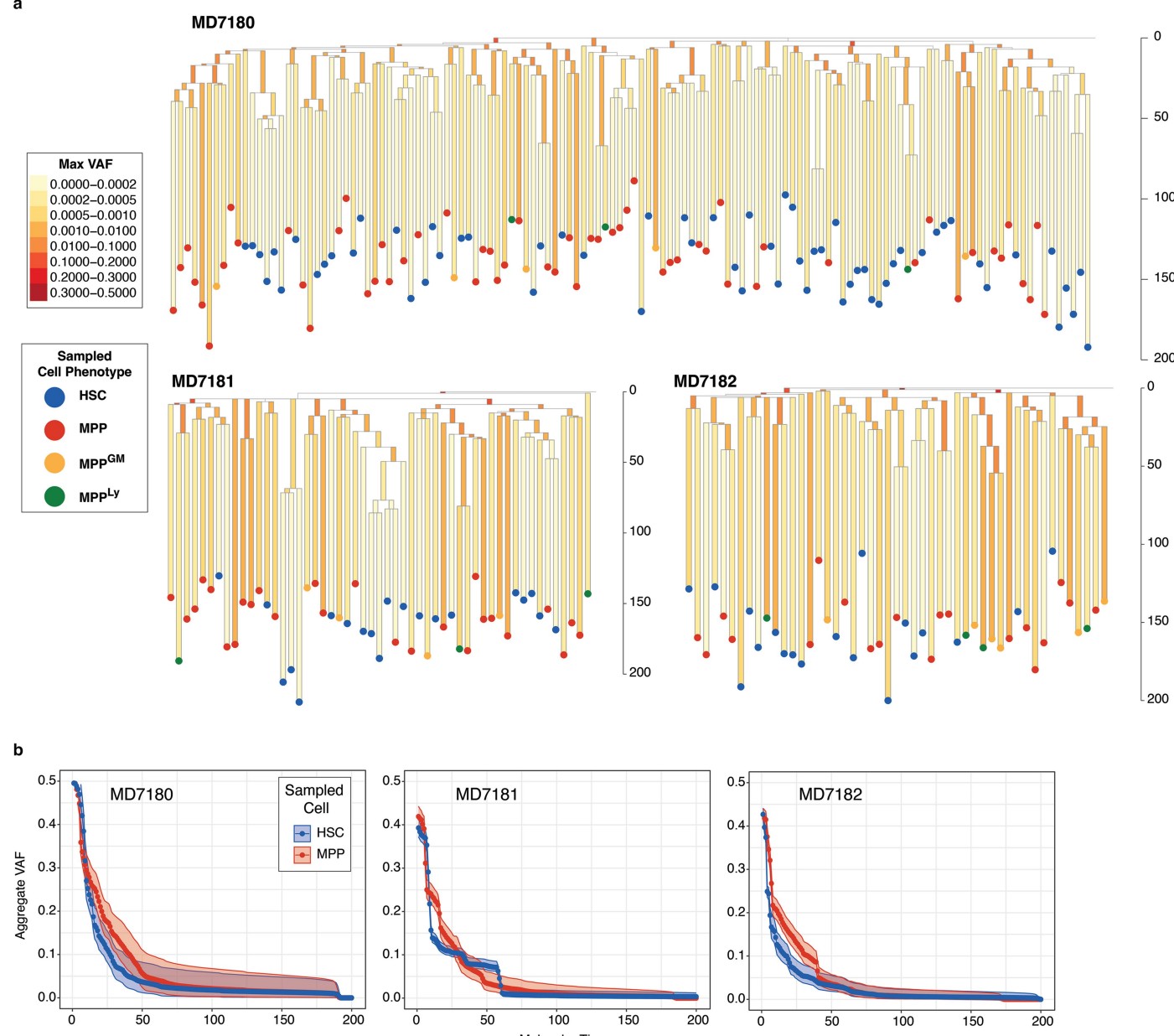

**Extended Data Fig. 8 | Mutation overlap between phylogenies and peripheral blood. a**) Phylogenies for three aged mice (as described in Extended Data Fig. 7a–c) constructed to only include private branches targeted with the peripheral blood baitset. Branch shading indicates the maximum VAF among branch-specific variants captured in peripheral blood. The sampled cell immunophenotype is indicated by dot colour at the bottom of each private branch. **b**) VAF trajectories of HSC and MPP variants shared in peripheral blood. The aggregate VAF across molecular time is calculated using Gibbs sampling (Methods). Earlier molecular time corresponds to further in the ancestral past. Shaded regions denote 95% confidence intervals of VAF estimates.

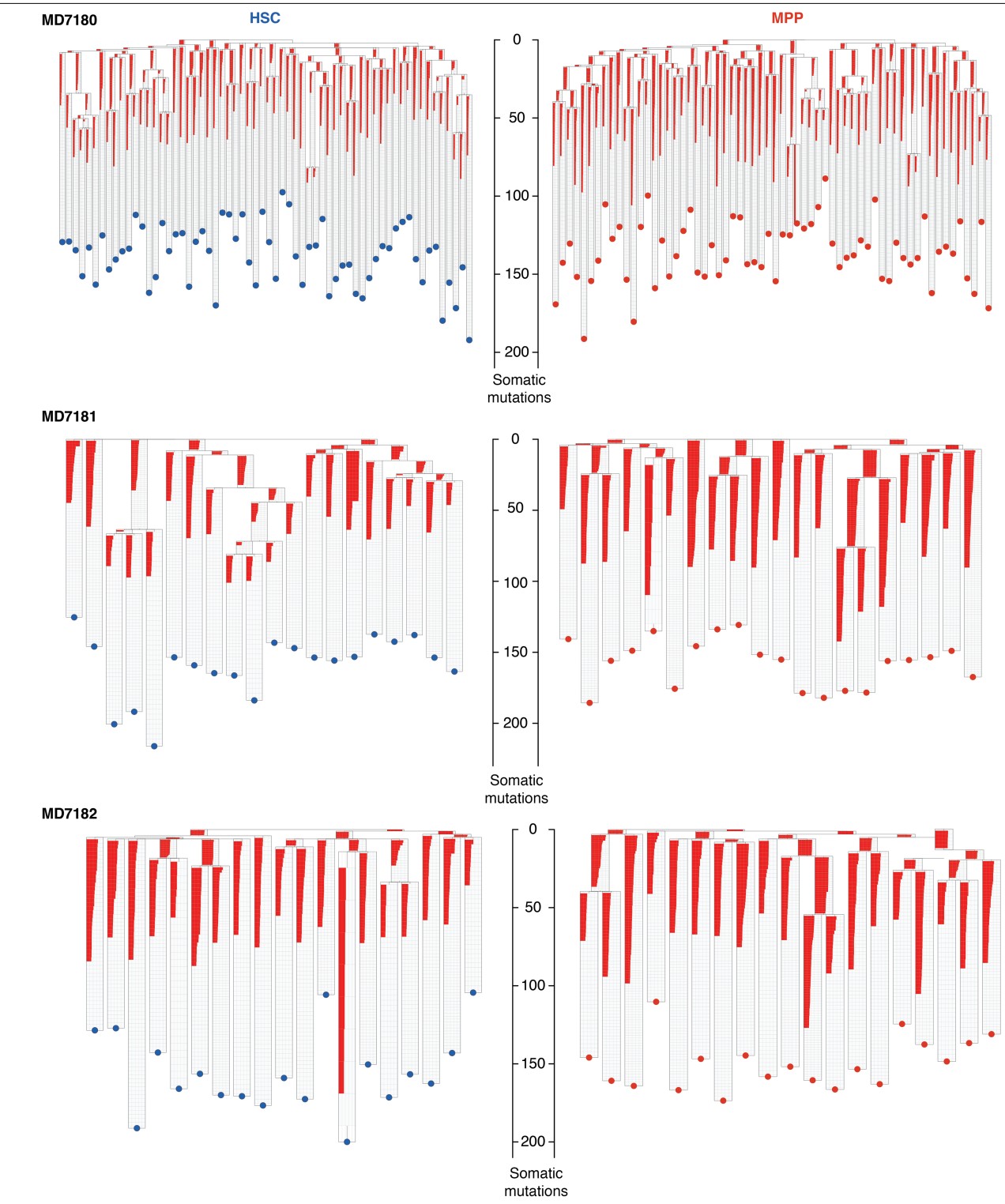

**Extended Data Fig. 9 | Peripheral blood VAF of variants shared with HSCs and MPPs.** Baitset mutation-specific HSC and MPP phylogenies are shown for each 30-month mouse. Each branch shows mutations that were detected in peripheral blood in descending VAF order. On each branch, a row denotes a single variant mapped to that specific branch. Red fill denotes the peripheral blood VAF for the variant. VAF is denoted on a log scale from 10-5 to 1; internal divisions are marked from left to right at VAF 0.0001, 0.001, 0.01, and 0.1. HSC trees are shown on the left with blue dots at terminal branches; MPP trees are shown on the right with red dots. Trees are downsampled to allow equivalent comparison between HSC and MPP branches. Only variants seen in peripheral blood with a depth >100X are shown.

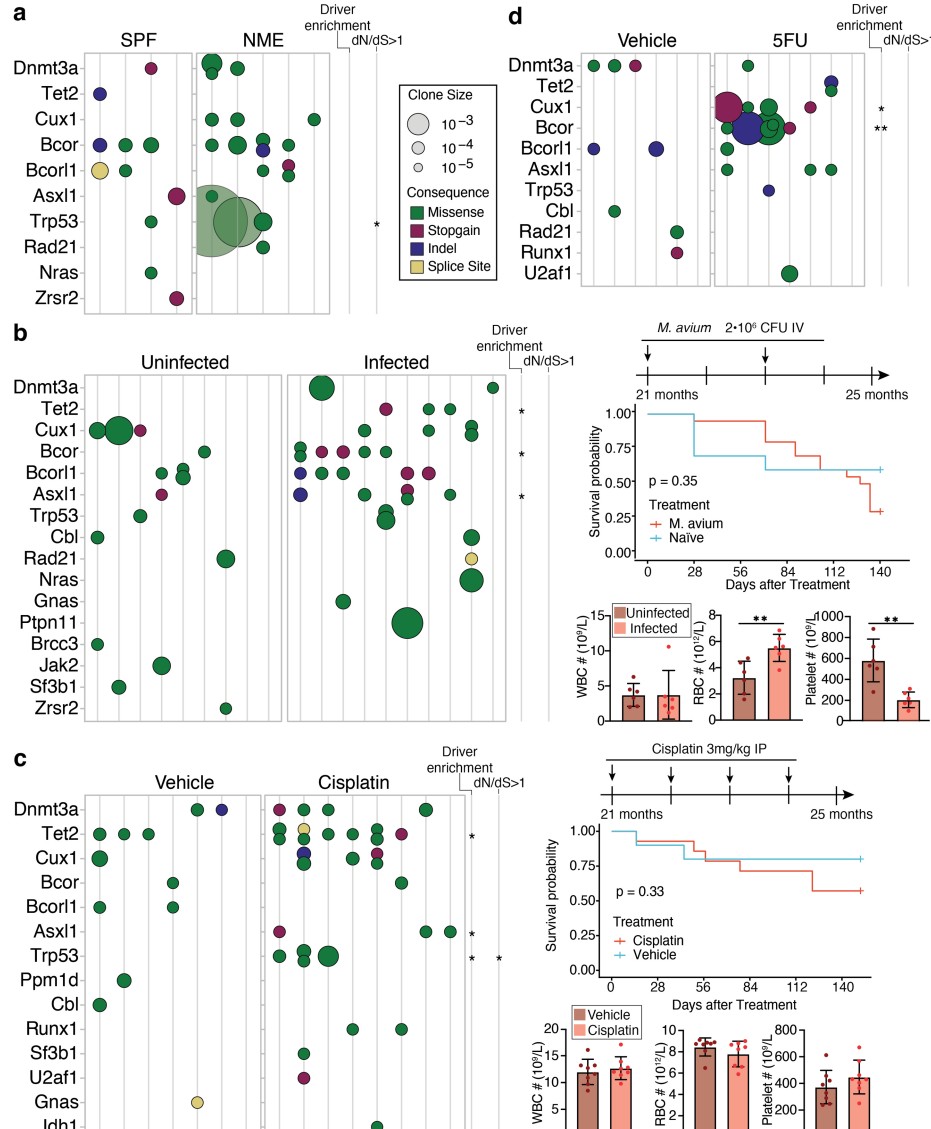

**Extended Data Fig. 10 | Haematopoietic perturbation modulates selection landscapes.** Clonal haematopoiesis prevalence in aged mice following **a**) normalized microbial experience (NME), **b**) *M. avium* infection, **c**) cisplatin treatment, and **d**) 5-FU myeloablation. At final sampling, aged mice were 30 months old for the NME experiments in panel a), and were 25 months old for the perturbation experiments in panels b), c), and d). Enrichment of clonal prevalence (measured using a two-sided Mann-Whitney test) and dN/dS ratios departing from parity (q < 0.1) following treatment are shown for each gene. Within b), significance values for Tet2, Bcor, and Asxl1 are 0.047, 0.046, 0.046, respectively. Within c) significance values for Tet2, Asxl1, and Trp53 are 0.025,

0.049, 0.037, respectively. Within d), significance values for Cux1 and Bcor are 0.022 and 0.007, respectively. Survival curves and experimental endpoint blood counts are displayed for b) and c), using log-rank and two-sided t tests, respectively. For blood count data, bar height denotes the mean among samples (individual dots), error bars denote SEM, and each data point is a biologically independent animal aggregated from 2 independent treatment cohorts. * denotes p-value < 0.05 and ** denotes p-value < 0.01, with no correction for multiple hypothesis testing. Treatment schedules are as displayed or described in Methods.

# Reporting Summary

## Statistics

For all statistical analyses, confirm that the following items are present in the figure legend, table legend, main text, or Methods section.

| n/a | Confirmed | |
|---|---|---|
| ☐ | ☒ | The exact sample size (*n*) for each experimental group/condition, given as a discrete number and unit of measurement |
| ☐ | ☒ | A statement on whether measurements were taken from distinct samples or whether the same sample was measured repeatedly |
| ☐ | ☒ | The statistical test(s) used AND whether they are one- or two-sided<br>*Only common tests should be described solely by name; describe more complex techniques in the Methods section.* |
| ☒ | ☐ | A description of all covariates tested |
| ☐ | ☒ | A description of any assumptions or corrections, such as tests of normality and adjustment for multiple comparisons |
| ☐ | ☒ | A full description of the statistical parameters including central tendency (e.g. means) or other basic estimates (e.g. regression coefficient) AND variation (e.g. standard deviation) or associated estimates of uncertainty (e.g. confidence intervals) |
| ☐ | ☒ | For null hypothesis testing, the test statistic (e.g. *F*, *t*, *r*) with confidence intervals, effect sizes, degrees of freedom and *P* value noted<br>*Give P values as exact values whenever suitable.* |
| ☐ | ☒ | For Bayesian analysis, information on the choice of priors and Markov chain Monte Carlo settings |
| ☐ | ☒ | For hierarchical and complex designs, identification of the appropriate level for tests and full reporting of outcomes |
| ☐ | ☒ | Estimates of effect sizes (e.g. Cohen's *d*, Pearson's *r*), indicating how they were calculated |

*Our web collection on statistics for biologists contains articles on many of the points above.*

## Software and code

Policy information about availability of computer code

| Data collection | Flow cytometry data was collected using BD FACSDiva version 9.0 and analyzed using FlowJo version 10.8.1. |
|---|---|
| Data analysis | SNVs and indels were detected using CaVEMan (version 1.14.0, https://github.com/cancerit/CaVEMan), cgpPindel (version 3.9.0, https://github.com/cancerit/cgpPindel), and VarDict (version 1.8.3, https://github.com/AstraZeneca-NGS/VarDictJava). Variants were annotated using VAGrENT (version 3.7.0, https://github.com/cancerit/VAGrENT), and Ensemble VEP (release 107-110.0, https://github.com/Ensembl/ensembl-vep). Phylogenies were constructed using MPBoot (version 1.1.0, https://github.com/diepthihoang/mpboot). Variants were assigned to phylogenies using Rtreemut (https://github.com/nangalialab/treemut). Population trajectories were inferred using phylodyn (https://github.com/mdkarcher/phylodyn). Bayesian inferences utilized the packages rsimpop (https://github.com/nangalialab/rsimpop) for simulations and abc (version 2.2.1, https://CRAN.R-project.org/package=abc) for approximate Bayesian Computation. Mutation signatures were inferred using the hdp (https://github.com/nicolaroberts/hdp) and sigfit (version 2.2.0, https://github.com/kgori/sigfit). Duplex consensus reads were generated using the fgbio suite of tools (version 1.5.1-2.1.0, http://fulcrumgenomics.github.io/fgbio/). dN/dS ratios were calculated using dNdScv (version 0.1.0, https://github.com/im3sanger/dndscv). Custom DNA baitsets were designed using the Agilent SureDesign tool (version 7.10.2). Population genetic analyses of clone sizes and parameter inferences were based on code available at https://github.com/blundelllab/ClonalHematopoiesis/. Other analyses were carried out using custom R scripts (R version 4.3) and will be are available at https://github.com/CDKapadia/somatic-mouse. |

For manuscripts utilizing custom algorithms or software that are central to the research but not yet described in published literature, software must be made available to editors and reviewers. We strongly encourage code deposition in a community repository (e.g. GitHub). See the Nature Portfolio guidelines for submitting code & software for further information.

## Data

Policy information about availability of data

All manuscripts must include a data availability statement. This statement should provide the following information, where applicable:

- Accession codes, unique identifiers, or web links for publicly available datasets
- A description of any restrictions on data availability
- For clinical datasets or third party data, please ensure that the statement adheres to our policy

> Whole genome sequencing data will be deposited at the European Nucleotide Archive at accession numbers ERP138320, ERP152795, and ERP144323. Targeted duplex sequencing data has been deposited at NCBI BioProject PRJNA1033340.

## Research involving human participants, their data, or biological material

Policy information about studies with human participants or human data. See also policy information about sex, gender (identity/presentation), and sexual orientation and race, ethnicity and racism.

| | |
|---|---|
| Reporting on sex and gender | No human data in this study. |
| Reporting on race, ethnicity, or other socially relevant groupings | No human data in this study. |
| Population characteristics | No human data in this study. |
| Recruitment | No human data in this study. |
| Ethics oversight | No human data in this study. |

Note that full information on the approval of the study protocol must also be provided in the manuscript.

# Field-specific reporting

Please select the one below that is the best fit for your research. If you are not sure, read the appropriate sections before making your selection.

☒ Life sciences ☐ Behavioural & social sciences ☐ Ecological, evolutionary & environmental sciences

For a reference copy of the document with all sections, see nature.com/documents/nr-reporting-summary-flat.pdf

# Life sciences study design

All studies must disclose on these points even when the disclosure is negative.

| | |
|---|---|
| Sample size | No power calculations were used to determine sample size. Sample size was determined based on feasible experimental size and subject survival. |
| Data exclusions | Whole-genome data was excluded due to quality control failures as described in the methods. Exact numbers of excluded data are described specifically.<br>-Some colonies were excluded based on low coverage or evidence of non-clonality or contamination. Visual inspection of filtered variant VAF distributions per colony was used to identify colonies with mean variant allele fraction (VAF)<0.4 or with evidence of non-clonality (Extended Data Fig. 1C).<br>-Following exclusion of 106 colonies due to low sequencing coverage or lack of clonality (Extended Data Fig.1C, methods), 1305 whole genomes (666 HSC, 639 MPP) were taken forward for somatic mutation identification and phylogenetic reconstruction. |
| Replication | Whole genome analysis was not replicated due to large per-subject datasets. For duplex experiments, each individual mouse queried is an independent biological replicate. In perturbation experiments, displayed data is aggregated from at least 2 independent replicate treatment cohorts. All subjects within experimental cohorts are included in the aggregated data. Replicate experiments were designed identically. All replicate experiments were successful. |
| Randomization | Mice were randomly allocated to control or experimental groups. |
| Blinding | Investigators were not blinded to the group assignment during experiments or during analysis because the same investigators performed all mouse experiments and subsequent bioinformatics analyses. |

# Reporting for specific materials, systems and methods

We require information from authors about some types of materials, experimental systems and methods used in many studies. Here, indicate whether each material, system or method listed is relevant to your study. If you are not sure if a list item applies to your research, read the appropriate section before selecting a response.

## Materials & experimental systems

| n/a | Involved in the study |
|---|---|
| ☐ | ☒ Antibodies |
| ☒ | ☐ Eukaryotic cell lines |
| ☒ | ☐ Palaeontology and archaeology |
| ☐ | ☒ Animals and other organisms |
| ☒ | ☐ Clinical data |
| ☒ | ☐ Dual use research of concern |
| ☒ | ☐ Plants |

## Methods

| n/a | Involved in the study |
|---|---|
| ☒ | ☐ ChIP-seq |
| ☐ | ☒ Flow cytometry |
| ☒ | ☐ MRI-based neuroimaging |

## Antibodies

| | |
|---|---|
| Antibodies used | • FACS: anti-CD150-Bright Violet 711 (Biolegend, catalog number 115941, clone TC15-12F12.2, 1:50)<br>• FACS: anti-CD48-FITC (eBioscience, catalog number 11-0481-82, clone HM48-1, 1:100)<br>• FACS: anti-CD117-APC (eBioscience, catalog number 17-1171-83, clone 2B8, 1:100)<br>• FACS: anti-FLT3-PE (eBioscience, catalog number 12-1351-82, clone A2F10, 1:50)<br>• FACS: anti-Sca1-PE-Cy7 (eBioscience, catalog number 25-5981-82, clone D7, 1:100)<br>• FACS: anti-Ter119-eFlour-450 (eBioscience, catalog number 48-5921-82, clone Ter119, 1:100)<br>• FACS: anti-Gr1-eFlour-450 (eBioscience, catalog number (48-5931-82, clone RB6-8C5, 1:100)<br>• FACS: anti-CD8a-eFlour-450 (Invitrogen, catalog number 48-0081-82, clone 53-6.7, 1:100)<br>• FACS: anti-Mac1-eFlour-450 (eBioscience, catalog number 48-0112-82, clone M1/70, 1:100)<br>• FACS: anti-CD4-eFlour-450 (eBioscience, catalog number 48-0041-82, clone GK1.9, 1:100)<br>• FACS: anti-B220-eFlour-450 (eBioscience, catalog number 48-0452-82, clone 48-0452-82., 1:100) |
| Validation | All antibodies are from commercial vendors and have been validated by the manufacturer with supporting publications found on the manufacturer's website for each antibody. See below for summary:<br><br>FACS: anti-CD150-Bright Violet 711 (Biolegend, catalog number 115941, clone TC15-12F12.2, 1:50)<br>Species: Mouse<br>Application: FACS<br><br>FACS: anti-CD48-FITC (eBioscience, catalog number 11-0481-82, clone HM48-1, 1:100)<br>Species: Mouse<br>Application: FACS<br><br>FACS: anti-CD117-APC (eBioscience, catalog number 17-1171-83, clone 2B8, 1:100)<br>Species: Mouse<br>Application: FACS<br><br>FACS: anti-FLT3-PE (eBioscience, catalog number 12-1351-82, clone A2F10, 1:50)<br>Species: Mouse<br>Application: FACS<br><br>FACS: anti-Sca1-PE-Cy7 (eBioscience, catalog number 25-5981-82, clone D7, 1:100)<br>Species: Mouse<br>Application: FACS<br><br>FACS: anti-Ter119-eFlour-450 (eBioscience, catalog number 48-5921-82, clone Ter119, 1:100)<br>Species: Mouse<br>Application: FACS<br><br>FACS: anti-Gr1-eFlour-450 (eBioscience, catalog number (48-5931-82, clone RB6-8C5, 1:100)<br>Species: Mouse<br>Application: FACS<br><br>FACS: anti-CD8a-eFlour-450 (Invitrogen, catalog number 48-0081-82, clone 53-6.7, 1:100)<br>Species: Mouse<br>Application: FACS<br><br>FACS: anti-Mac1-eFlour-450 (eBioscience, catalog number 48-0112-82, clone M1/70, 1:100)<br>Species: Mouse<br>Application: FACS<br><br>FACS: anti-CD4-eFlour-450 (eBioscience, catalog number 48-0041-82, clone GK1.9, 1:100)<br>Species: Mouse<br>Application: FACS |

FACS: anti-B220-eFlour-450 (eBioscience, catalog number 48-0452-82, clone 48-0452-82., 1:100)
Species: Mouse
Application: FACS

# Animals and other research organisms

Policy information about studies involving animals; ARRIVE guidelines recommended for reporting animal research, and Sex and Gender in Research

| Laboratory animals | Mouse strain, age, and sex is described in the manuscript text, legends, and methods. All mice were C57BL/6, C57BL/6J:FVB/NJ F1 hybrid mice , or HET3 mice. Mouse age ranged from 3 months to 36 months, as described in the manuscript. Wild-type C57BL/6 mice were bred at Baylor College of Medicine or received from the Aged Rodent Colony at the National Institute of Aging (Baltimore, MD). C57BL/6J:FVB/NJ F1 hybrid mice were bred in the Niedernhofer laboratory at the University of Minnesota. HET3 mice were bred at the Jackson Laboratories. C57BL/6 were housed at the AAALAC-approved Center for Comparative Medicine in BSL-2 suites. Mice were housed with the same sex in ventilated cages, under a 14/10-hour light-dark cycle with temperature and humidity control, enrichment material, and fed ad libitum with a standard chow diet. |
|---|---|
| Wild animals | No wild animals were used in this study. |
| Reporting on sex | Duplex sequencing experiment used a mix of male and female mice because clonal hematopoiesis occurs in both sexes. For whole genome colony data, all mice were female. |
| Field-collected samples | No field collected samples were used for this study. |
| Ethics oversight | Experimental procedures were approved by the Baylor College of Medicine or University of Minnesota Institutional Animal Care and Use Committees and performed following the Office of Laboratory Animal Welfare guidelines and PHS Policy on Use of Laboratory Animals. |

Note that full information on the approval of the study protocol must also be provided in the manuscript.

# Plants

| Seed stocks | No plants were used in this study. |
|---|---|
| Novel plant genotypes | No plants were used in this study. |
| Authentication | No plants were used in this study. |

# Flow Cytometry

## Plots

Confirm that:

☒ The axis labels state the marker and fluorochrome used (e.g. CD4-FITC).

☒ The axis scales are clearly visible. Include numbers along axes only for bottom left plot of group (a 'group' is an analysis of identical markers).

☒ All plots are contour plots with outliers or pseudocolor plots.

☒ A numerical value for number of cells or percentage (with statistics) is provided.

## Methodology

| Sample preparation | Whole bone marrow (WBM) cells were isolated from murine hindlimbs and enriched for c-Kit+ hematopoietic progenitors prior to fluorescence-activated cell sorting (FACS). WBM was incubated with anti-CD117 microbeads (Miltenyi) for 30 minutes at 4C following my magnetic column enrichment (LS Columns, Miltenyi). HSCs were defined as Lineage–ckit+Sca-1+FLT-3–CD48–CD150+; MPPs were defined as Lineage–ckit+Sca-1+FLT-3–CD48–CD150–. The gating strategy is illustrated in Extended Data Fig. 1A. |
|---|---|
| Instrument | FACS was completed on a BD FACSAriaII. |
| Software | Flow cytometry data was collected using BD FACSDiva version 9.0 and analyzed using FlowJo version 10.8.1. |

Cell population abundance | Cell population abundance is described in Figure 3b and Extended Data 1A.

Gating strategy | Gating strategy is describing in Extended Data 1A.

☒ Tick this box to confirm that a figure exemplifying the gating strategy is provided in the Supplementary Information.

