## [Peer Review file · Nature]

Clonal dynamics and somatic evolution of haematopoiesis in mouse

Corresponding Author: Dr Jyoti Nangalia

Version 0:

Reviewer comments:

Referee #1

(Remarks to the Author)

In “Clonal dynamics and somatic evolution of haematopoiesis in mouse” Kapadia and colleagues seek to understand how hematopoiesis arises and evolves over the course of the mouse lifespan. They leverage phylogenetic analyses to infer that mouse hematopoietic stem cells and hematopoietic progenitor cells are established during embryogenesis and renew independently of one another over the lifespan. Unlike humans, aged mice do not appear to develop large clonal hematopoiesis or have the loss of stem cell diversity that is characteristic of human hematopoiesis. The authors observe that small clones arise with aging but why such clones expand in humans and do not appreciably expand in mice is unclear. The authors use mathematical modeling to suggest that there are insufficient cell divisions within the comparatively short mouse lifespan to enable detectable clonal expansions which is a widely held opinion, but not one that has been rigorously evaluated before. Although I hold this group of investigators in the highest esteem, from my perspective the scientific observations presented here fall a bit short of the kind of ‘field setting’ observations that are of broad interest to this journal's diverse scientific readership.

1. The first major finding that mouse HSCs and progenitors can arise independently and contribute to hematopoiesis over the lifespan has been described in a series of papers using murine lineage tracing techniques (Rodriguez-Fraticelli, Nature 2018; Rodriguez-Fraticelli, Nature 2020; Patel, Nature 2022; Li, Cell 2023; Kobayashi, Cell Reports 2023). These lineage tracing tools enable not just phylogenetic reconstruction but have the added advantage of permitting phenotypic characterization (for example with single cell RNA sequencing). Kapadia and colleague's use of spontaneously arising somatic mutations as naturally arising barcodes is elegant but feels like an orthogonal replication of this reasonably well developed line of research rather than a distinct divergence. To increase novelty, the authors could perhaps perform a similar set of analyses in humans where the mouse lineage tracing tools such as those used by the Camargo lab are unavailable (eg there are no “DARLIN” humans).
2. The second major finding is that using error corrected sequencing (Twin Strand DuplexSeq) on a small panel of 9 putative clonal hematopoiesis driver genes and specific hotspots for 15 other genes the authors observe clonal mutations in the 10⁻³-10⁻⁵ allele frequency range. The gene distribution of these mutations shift in response to different perturbations (eg chemotherapy, infectious exposures). This part of the paper, essentially suggests that the mutations in CH driver genes do happen in mice and cause positive selection. There has been active debate in the CH field on this point and to my knowledge there is no convincing published report one way or the other. That being said, the DuplexSeq method has been actively taken up by the toxicology field more broadly with groups identifying that this method can identify teratogenic exposures in mice and rats (Valentine, PNAS 2020; LeBlanc, BMC Genomics 2022; Dodge, Arch Toxicol 2023;) In these prior papers, mutational patterns were observed to occur genome wide (often in random parts of the genome) rather than focusing on CH genes.
3. From a technical perspective, there have been concerns about the accuracy of TwinStrand technology – for example in Supplementary Figure S8 the biological replicate samples only appear to have a perhaps 50% concordance of identifying a mutation at the 10⁻⁴ to 10⁻⁵ clone size (and there is only one 10⁻³ clone so it is difficult to truly evaluate), this has motivated the development of other technologies like NanoSeq (Abascal, Nature 2021). I am not sure whether the duplex sequencing assay can reliably detect clones in the <0.1% range which is where many of the observed clones occur and whether the mutational spectrum of driver mutations observed in the limited sample size actually does recapitulate human CH.
4. The technical accuracy of the duplex sequencing aside, the set of observations presented in Fig 4 and 5 highlighting different murine perturbations largely replicate that which is already known in humans (or has been established with other murine clonal hematopoiesis studies). For me what is missing here is something that takes unique advantage of the mouse as a system to show something new or that cannot (easily) be done in humans. For example, if the same cisplatin

perturbation is performed across an array of different mouse genotypes can a genetic locus be identified that controls susceptibility to somatic mutation in the setting of cisplatin perturbation. Or if the authors showed that administration of a CDK4/6 inhibitor could prevent CH from arising in mice when measured with their TwinStrand assay (building on observations from He, Science Translational Med. 2017) then the work would point to a potentially novel therapeutic strategy for treating CH. (These are not specific suggestions, but just illustrations of experiments within this system that the authors have established that I would find 'field-setting'. I am confident that the authors will have better ideas than me of exciting applications.)

5. The authors raise an interesting possibility that the observations made about species size and lifespan by Cagan et al [Nature 2022] may be tissue specific (eg the colon is different from blood). One could consider expanding this aspect of the manuscript by profiling blood cells from several different species, which could refine our fundamental understanding about this evolutionary process.

6. I thought the observation that the rate of HSC → MPP transitions might change over time was interesting (described in lines 175-9). This observation I do not think has been investigated by prior lineage tracing work and could perhaps be expanded upon with different timepoints or different models (or possibly through the re-analysis of published data from lineage tracing mouse models)

7. From a clinical perspective, does this work have implications for the cell of origin debate for AML or CHIP in humans (eg whether it is derived from the stem cell or from the MPP)? Could the authors perhaps show this by sorting the cells prior to duplex sequencing.

A few more minor considerations:

8. The SBS substitution rate with nanorate sequencing appears to be ~2-4x greater than that of the colonies. Similarly the N/lambda estimates from WGS single cell colonies are also mismatched by a similar factor – how do the authors explain this difference?

9. How have the authors excluded the possibility of SBS18 C>A transversions potentially linked to oxidative damage as not simply representing an artifact in library preparation or DNA sequencing.

10. Are the estimates of mutation rate consistent with the estimates of rate of cell divisions (70k cells, divisions every 6 weeks)

11. I feel that the authors are overstating the notion that the field is stuck in a rigid notion of 'classical hematopoiesis models' as a strawman when work over the past 5 years has presented a more nuanced view and would consider revising both the introduction and discussion accordingly.

12. A very minor point in Fig 1D – the two-tone HSC (left) MPP (right) color scheme are visually pleasing but the meaning is a bit hard to decipher without connecting it to the colors in 1B. For Fig 1D, I would suggest considering using different marker shapes (eg dots and Xs or colored dots and open dots) for HSCs and MPPs or making the violin around MPPs shaded light and dark or alternatively just labeling on the x axis HSC and MPP (or H and M).

Referee #2

(Remarks to the Author)

The authors apply very powerful analysis based on WGS of colonies derived from murine HSCs and phenotypically-defined putative downstream MPPs to gain really significant and immediately impactful insights into steady state hematopoiesis and hematopoietic aging, not dependent like all prior studies on transplantation as a readout for HSC function or on complex transgenic models with low resolution. While the concepts and methods applied are not novel (as some of the authors have used the approaches very very productively over the past several years to understand human hematopoiesis at steady state, with aging, post-transplant and in disease states), this is the first paper investigating the murine model, which is the absolute mainstay of all biomedical research on hematopoiesis and blood diseases, so the really stark differences uncovered in very basic parameters in mice vs humans are of great interest and importance to try and understand whether and how to utilize these models in basic discovery and in therapeutic development. The major findings include: 1) mutation rate for HSCs in mice is more than two-fold that of human HSCs, but lower than expected from differences in lifespan and the large differences for instance in murine vs human colonic mucosa. 2) The concept of LT-HSCs, able to self-renew through multiple rounds of serial transplantation vs ST-HSCs or MPPs able to engraft long term in primary but not serial transplants, the crux of much murine research and paradigm crafting over the years, appear to have no meaning in steady state non transplanted hematopoiesis in mice. So much of what we think we know about HSCs has been based on this distinction, and clearly that needs a major reconsideration based on this paper. This helps explain why already quite a few concepts don't translate from mouse models to humans based on what appears to be quite an artificial distinction. 3) HSCs/MPPs in mice have quite different trajectories over the lifespan in mice vs humans, with murine cells continuing to increase in number over time, reaching total numbers equivalent to humans despite much smaller size and shorter lifespans. 4) With aging, the massive clonal expansions seen in humans (ie "CH" or age-related clonal hematopoiesis) are simply not seen in mice. Much smaller expansions, some mutations in the same genes as in humans, can be found if look hard enough (also reported previously by a Japanese group but this goes far beyond that in terms of sophistication of analysis and explanations), but only with application of major stressors such as chemotherapy or chronic infection do the clones become detectable, and almost never reach the size seen in humans-making mice a questionable model for CH in humans. Every part of this study is interesting and will be widely read and discussed by anyone interested in stem cells, development, normal or abnormal hematopoiesis, aging, and transplantation and gene therapies. There are a couple of additional experiments that would increase the significance and complete the story and place it in context, as detailed below. The paper is very well-written and most of the figures are clear and meaningful. Methodologies in general are completely and carefully explained.

1. The core data in the paper is all based on one mouse strain with relatively long lifespan and known slower HSC cycling. Even the data on clonal hematopoiesis via targeted sequencing added two strains that are a C57F1 hybrid or a pretty unique 4 strain mix. Would have been of interest to instead include a strain with properties more extreme in the other direction ie

- DBA, with shorter life span, faster HSC cycling. Given that C57 used for >>90% of murine HSC and transplant studies, makes sense that the major work was carried out with only that strain, but at least this drawback should have been noted and discussed as a caveat and of importance for future studies if only one strain was utilized.
2. Phenotypically the ratio of HSC to MPPs changes between the two time points studied, but the number of colonies used for WGS and SNV discovery and phylogenetic reconstructions and thus basically entire paper based on approximately equal numbers of HSC and MPPs single cell colonies utilized. Could this throw off any of the models? Was this accounted for in any way? Does it need to be?
 3. At least to this reader, the data showing little evidence for HSC to MPP transitions is the single most important point of paper, so when first introduced based on the phylogenetic trees, a bit more clarity would be helpful—a more extensive even if succinct explanation for instance of the approach resulting in Figure 2C and D somewhere other than in the legend and wading through the Supplemental Notes would be helpful. This is true for large chunks of the paper and realize it can't all be in the main text, but the converse of no explanation is confusing to the reader.
 4. It was disappointing that the authors did not look at the blood and blood lineages (in the more usual sense ie myeloid/lymphoid) particularly regarding the question of whether differentiation/mature cell output from HSCs/MPPs reflects what is happening at the HSC/MPP level. Presumably there is some reason the two populations exist and are so distinct and for a really important paper such as this one with huge implications for use of mice as models for human biology and disease, this seems a rather large missing piece of information. Sorted populations transplanted in mice show that HSCs or MPPs transplanted separately result in normal blood counts, survival etc but the authors have the opportunity to see what happens in steady state normal hematopoiesis, seems rather important to look at. One hopes there is DNA from blood collected at the time of HSC/MPP harvest/purification from those 6 animals available to perform these studies.
 5. It would have been of interest to perform transplants from at least some of the primary animals to see happened to the HSC and MPP trees in that setting and whether there was any difference from what has been observed using sorted HSCs and MPPs where transition from HSC to MPP must be occurring, but the authors have the chance to see how much of a bottleneck that process might entail.
 6. For being able to easily appreciate the marked differences between human and murine phylogenetic trees, it would be important to include human trees in this paper, rather than having to go back and find these in prior publications. Regenerating them with the mutational clock dimension translated into lifespan for both the mouse and human to allow direct comparisons would be awesome. The whole point of the paper is really summed up in being able to appreciate that comparison in terms of where coalescences occur for instance.
 7. To this reader, the section on explanations for mutation rate being higher in mouse than human and types of mutational signatures found is out of place—would consider moving to right after the initial mention of the trees (but before all on HSC vs MPPs), because now it breaks up that set of studies/concepts. Or if there isn't room, to me much of this could move to Supplemental Notes, since no really firm conclusions.
 8. Are HSCs that have mutations more likely to disappear in humans than mice as another possible explanation for the lower apparent mutation rate? Known that DNA damage/breaks/some types of mutations results in P53-mediated apoptosis in human cells versus repair in murine.
 9. The quite comprehensive analysis for clonal expansions (or lack) by both the initial WGS approach and targeted sequencing is very important, and the attempts to understand why are quite convincing. The section on application of various stressors perhaps not particularly definitive. More mice with fewer stressors might have been better, making any broad conclusions about types of mutations and expansions with only 4-5 mice per stressor isn't very convincing. Please specify what age the mice used were (legend just says "aged"). This section seems a topic mostly for another paper in terms of trying to figure out what stressor might be most relevant to using the mouse as a model for CH in humans.
 10. The authors state in the discussion that "...any role age associated hematopoietic oligoclonality plays in human aging is unlikely to be shared by the laboratory mouse". What does this imply for murine-based HSC research being extrapolated to humans? Are there alternative models? No mention made of data from NHPs showing stem cell numbers and cycling estimates close to humans (Shepherd et al Blood 2007; Koelle et al Blood 2017) and clonal expansions with NHP aging (Yu et al Blood 2018; Shin et al Blood 2023).

Minor:

11. In Extended Figure 1C, what is the VAF referring to? Composite of all mutations? A specific site. Overall the figure legends throughout leave out quite a lot to help someone understand what is being shown without referring to Supplemental text sections on methods, or inference based on one's own knowledge. Please fix throughout going through with someone not an author to see what they don't understand or can't figure out easily.
12. Figure 1 Legend the last sentence of the description of panel B I believe refers to panel C, please move to correct location.
13. On the phylogenetic trees, continually refer to "grey" lines but they sure look black to me, I kept trying to enlarge panels to see if there were both grey and black sections and I was missing something. Just call them black please throughout all the various parts of text referring to the trees, and in legends.
14. Line 304 refers to wrong panel of Figure 4, F not E
15. In Extended Figure 4, includes HSC and MPP together?
16. Figure 4 panel arrangement is annoying and confusing, completely out of normal top to bottom and left to right order. Also the legend for clone size and mutation type shouldn't be hidden down by panel F, since it applies to all the panels, correct? Needs to be shown with panel A.

Referee #3

(Remarks to the Author)

Kapadia et. al. carry out whole genome sequencing of more than a thousand single blood cells from three young mice and

three very old mice in order to examine somatic evolution of murine blood. Their observations are multitudinous. They determine mutation burdens, noting that mice have few mutations in their blood near the end of life relative to humans, which is relevant to a larger discussion on whether somatic mutation rates scale inversely with lifespan among species. They construct a phylogenetic tree for each mouse. The phylogenetic structures are used to infer that stem and multipotent progenitor cells (HSCs and MPPs respectively) are established early in life, and that the HSC and MPP populations independently self-renew and grow throughout life. The phylogenetic trees also suggest limited selection among the cells, in contrast to human blood. To study selection in greater resolution, they carry out deep targeted sequencing of the blood, focusing on genes orthologous to those genes recurrently mutated in human blood, and they find some very small clonal expansions apparently driven by mutations in these genes, and that mutation frequencies among these genes are increased by exposing the mice to pathogens and chemotherapies. Overall, although only few mice are included in the analysis, I appreciate that the genetic data are at exceptionally high resolution and that multiple aspects of somatic blood evolution are studied in some detail.

My main concern is that the novelty of the findings is limited, and prior literature not cited adequately. Chin et al. (Blood 2022) already demonstrated that there is a considerably greater mutation burden among single blood cells in humans compared to mice towards old age (specifically, Chin et al. report a 5-fold lower mutation in mice vs. humans, on a similar scale as the 10-fold difference reported here). Chin et al. also deeply sequenced mouse blood finding limited clonal hematopoiesis, although the present work deserves credit for even deeper sequencing, which is important given the rarity of clonal expansions. As for perturbations' effects on mutation frequencies in the blood, there are a great number of studies documenting increased clonal expansion after perturbations to hematopoiesis in a variety of settings (e.g. Hormaechea-Agulla Cell Stem Cell 2021, Heyde et al. Cell 2021, Meisel et al. Nature 2018 and many others); the advantage, which is substantial, of the present study is that the mice are 'normal' rather than being transplant recipients, however the concept is already extremely well established. Then there is the point that HSCs and MPPs are established during embryogenesis and that they independently self-renew through life. This was already demonstrated unambiguously in the mouse by Patel et al. in Nature 2022, whose experimental view of the blood in early life is arguably even more direct. This paper is cited but in a general way, without pointing out the vast overlap between that work and the findings presented as novel here.

In summary, several of the main findings of the present work seem to lack novelty, however I do enjoy the breadth and resolution of the data, which provide some of the clearest views of blood somatic evolution in mice to date. I think that the manuscript has notable weaknesses in clearly acknowledging previous work. It would be important to better delineate what is already known and what is new. Additionally, there are some weaknesses of the modelling analyses that would benefit from clearer acknowledgement and explanation (see further comments below).

Specific comments:

- Line 42 and the section beginning line 68: It is posed that it is unknown whether the somatic mutation burden in the blood at the end of life is similar between species. This gives an overstated impression of novelty of the somatic mutation burden measurements. Here, Chin et al.'s paper deserves a clearer acknowledgement. They have already measured that the somatic mutation burden of single HSCs is far larger in old mice compared to old humans. Their mice and humans are not quite so old as those of the present manuscript, which is in favor of the present manuscript, however Chin et al.'s results already unambiguously point towards substantially differing end of life mutation burdens.
- Line 69: "Comparison of HSCs from young and old mice revealed a constant rate of somatic mutation accumulation with age" is an overstatement. There are only two time points. A straight line can indeed be drawn between two points, but two points are not enough to claim that a straight line has been revealed.
- Line 98: "the mutation burden [of peripheral blood] was not different from that of hematopoietic colonies" could be phrased more carefully. Figure 1E suggests that there is a difference, albeit with limited statistical resolution. It shows that among the three peripheral blood measurements and the multiple hundred HSC colony measurements, the peripheral blood makes up the top three mutation burdens.
- Line 152: "Overall, our data suggest that many long-term HSC and MPP lineages are established independently and in parallel during development, and that MPPs do not always arise from HSCs, contrary to classical haematopoiesis models." It is important to note here that this summary point is not new. Several ideas in this section were already established by Patel et al. whose work deserves prominent acknowledgement.
- Line 156 to 186: To help readers understand the meaning of the inferences, I think it would be valuable to clearly explain that the transition rates governing the development of typical HSCs and MPPs are not equivalent to the transition rates governing the development of those specific HSCs and MPPs whose lineages are seen on the phylogenetic tree. This point is hinted at in the sentence "The apparent inconsistency in the results between young and old mice could perhaps be explained if the HSCs that produce the MPPs early in life are extinguished by old age." However I think that most readers will need more than this sentence to understand the difference. Related to this point, can you say the extent to which the relative numbers of sampled MPPs and HSCs (which differ from the relative numbers of MPPs and HSCs in the pool) influence the inferred transition rates? As the analysis stands, it's unclear to me how much the parameter estimates may be determined by sampling decisions rather than biology.
- Line 172 to 176: The p values presented here are impressive-looking, however I am unsure how seriously to take them. The likelihood ratio test is used on six independent data points (only three independent data points for the age-specific tests), but the likelihood ratio test's theoretical validity requires large sample sizes. I suggest either to note this potential caveat for the less statistically experienced readers or to leave aside the p values in order to focus on the Bayesian analysis.
- Line 248 to 251: Can you comment on the robustness of the estimates population size, and cell division and exit rates? In particular, how much do these estimates depend on your prior distributions? My sense is that these estimates could dramatically change with different priors. For example, what if the prior distribution on the population size were uniform between 10^2 and 10^7 rather than uniform between 10^2 and 10^5 as written? It's also worth noting that while inference of N/λ from phylogenies is standard, disentangling N and λ from phylogenies alone is notoriously problematic (see

Louca and Pennell Nature 2020 for example).

- Figure 3: presented estimates for lambda are 0.16, 0.17, and 0.14, but on line 769 it is stated that lambda has a uniform prior distribution between 0.01 and 0.15, indicating an error.
- Line 268: It is stated that there is an "absence of selection on non-synonymous mutations (using dN/dS)" in mouse blood, which by contrast manifests "ubiquitously over time in humans". Please can the authors clarify whether this mouse-human dN/dS comparison is merely a question of sample size or, if not clarify, state that the comparison is not clear? Figure 3D shows that dN/dS for old mice hovers around 1.05 (although the actual number is not stated) and with a confidence interval that just covers one. On the other hand, the cited study is based on a much larger number of human HSC and it calculates a genome-wide estimate for dN/dS of 1.06 with a narrower confidence interval that does not cover one.
- Line 307 to 335: The great literature on environmental exposure-altered clonal expansions in the blood (in both mouse and human) deserves more recognition.
- Line 349: This inference of selection uses the assumption that the stem cell population has constant size, however, Figure 3 claims a continuously growing stem cell pool which expands by several orders of magnitude throughout life. Can the inference be adapted to the setting of a changing population size?
- Line 371: Can you clarify, is s the clone growth rate? Also, by establishment, do you mean fixation or reaching exponential, deterministic growth? The latter option makes sense in the context of your sentence, but the former appears to be the topic of the cited paper.

Version 1:

Reviewer comments:

Referee #1

(Remarks to the Author)

I appreciate the authors thoughtful response to the technical concerns raised in the original review. I have several additional very minor queries:

1. Please include a short justification for why only female mice were used for the phylogeny inference experiment and clarify in the methods whether these mice were nulliparous or multiparous.
2. Do the authors feel the gene distribution of driver mutation by strain across C57BL/6 ; B6FVBF1/J; HET3 (Fig 4A,C,D) represents a different propensity to driver mutations across different strains or is random chance? This is a potentially interesting observation as it would suggest that mouse genetic background may play a role in shaping somatic evolution.
3. For Figure 5, I have concerns about how the gene level enrichment is calculated. Per the methods "Gene-level enrichment was measured using a Fisher's exact test on the number and mutant and wildtype reads, normalized for coverage differences between samples." I think it would be more standard (and more robust) to consider individual mice as independent replicates and compare cases vs controls with a given gene mutated as the input to the fisher exact test rather than considering each sequencing read as an independent experiment.
For example, for figure 5B - the figure appears to indicate that there is a significant driver enrichment (two asterix) for CUX1 - when there are 3/9 uninfected mice with mutations in this gene and 3/10 infected mice with mutations in this gene. A fisher exact test comparing 3/9 to 3/10 would yield a $p=1$. Similarly in 5C, there appear to be 3/7 mice with TET2 in the vehicle group and 6/8 mice with TET2 in the cisplatin group - fisher test comparing the number of mice would yield a $p=0.31$ for this comparison. For Fig 5D - if the CUX1 is significant for driver enrichment with 5FU treatment how is Bcor not significant? It is a bit hard to tell with the over plotting but it looks like a similar number of mice in the 5FU group have this mutation and there are no mice in the vehicle group with this mutation.
If the authors wish to account for the VAF size, there are also non-parametric tests that could be used to compare two groups of mice where the maximum VAF of a mutation in a given gene could be used and a mouse that does not have any mutation in a given driver gene could be included as 0.
4. For Fig 5 - please clarify in the methods or the figure legend that there is no correction for multiple hypothesis testing and also what the asterix means eg (*) vs (**).
5. For Figure 5B/ 5C /5D - there appear to be notable differences between the different control cohorts by gene - for example for TET2 there are 0/9 "uninfected" mice in 5B compared to 3/7 "vehicle" mice in 5C. For DNMT3A there are 0 mice in 5B Uninfected / 2 mice in 5C vehicle / and 3 mice in 5D "vehicle. Do the authors have thoughts about a source of this variation? If the vehicle mice and uninfected mice were compared using the gene level enrichment method described in the current draft(using a Fisher's exact test on the number and mutant and wildtype reads, normalized for coverage differences between samples), I worry that there could be gene level enrichment differences between these control groups which would not seem biologically plausible if they are the same age.

Referee #2

(Remarks to the Author)

Thank you for the comprehensive revision, I have no further specific comments.

Referee #3

(Remarks to the Author)

I thank the authors for their replies. The responses to my specific comments are mostly satisfactory, although I feel that at certain points in the manuscript the authors still do not acknowledge prior literature in a sufficiently transparent way. In particular, I strongly agree with Reviewer 1 that the finding that HSCs and MPPs give rise to independent peripheral blood populations in a lifelong manner is not novel: this fact is described in great detail by Patel et al. in an endogenous (non-transplantation) setting and I feel that the authors' acknowledgement of this directly relevant prior work with the sentence: "In recent years, a more nuanced and dynamic picture has emerged, with the identification of additional self-renewing progenitor compartments" is simply insufficient. I worry that this wording will be interpreted as a deliberate downplaying of Patel et al.'s insights. In fact, this sentence is the only place in the relevant results section where Patel is acknowledged, and the authors end this narrative again by saying that their data challenge classical hematopoiesis models, without notifying the reader that the model has already been challenged in an identical manner by prior work. The discussion I find more transparent.

Perhaps more glaringly, in their rebuttal the authors have refused to address the concern that Chin et al.'s finding that the end-of-life mutation burden in single mouse HSCs is relatively low in comparison with humans is not acknowledged. Their argument is that "given the above paper sequences 6 single-cell derived whole genomes from (presumably) one 24-month mouse, we strongly believe that the data, whilst the first description of the difference between mutation burden in equivalently aged mouse/human, is arguably preliminary given the limited sample set." I think this response is highly problematic. As the authors themselves and many others have shown, mutation accumulation in HSCs is predictable and scales linearly with time with relatively limited variance among cells. It is unsurprising that major biological effects can be discerned with 6 colonies from one mouse – the authors have no reason to dismiss this finding. Furthermore, the authors do not fairly acknowledge prior measurements of CH in mice. Their citation of Chin et al's work in the introduction "with preliminary data suggesting a lower rate of CH" is an understatement of the fact that Chin et al have unambiguously demonstrated that CH is more limited in mice compared to humans.

Overall, no technical issues with this paper remain, but the acknowledgement of prior work is still inadequate in the places outlined above. The most critical aspect raised in my original review – a relative lack of novelty which was also clearly articulated by Reviewer 1 – has not substantially changed in my assessment.

Version 2:

Reviewer comments:

Referee #1

(Remarks to the Author)

Thank you for your effort on this revision. I have no further specific comments.

20th September 2024

To the Editor

We thank the Referees for their overall enthusiasm about our work, and their valuable comments to improve it. We have addressed all of the comments with additional sequencing data, data re-analysis, and text changes. We provide a full point-by-point response below. All referee comments are shown in black text. Our responses are in blue text with any changes made to the manuscript highlighted in red text. Text included from the manuscript is highlighted, with amended text in red.

Below is a summary of the major additional areas of work:

1. We have sequenced an additional 242 HSC whole genomes from new mice of different ages. Our original cohort included 3 month and 30 month mice (total 6 animals). We now have additional genomes from a further 17 animals across the lifespan, at ages: 1 day (newborn), 6 months, 12 months, 18 months, and 24 months. This provides greater resolution into the rate of mutation acquisition across the murine lifespan, including how this differs during early development. (Referee 3)
2. In addition to interrogating clonal dynamics of HSC and MPPs, we have now sequenced 298 genomes of a mixed hematopoietic progenitor compartment (LSK cells: Lineage-, Sca1+, c-kit+) from the 30-month-old mice. This has allowed us to examine whether HSCs or MPPs differentially contribute to progenitor subtypes. (Referee 2)
3. Overall, we present 1845 whole genomes from C57Bl/6 mice. This is the largest whole genome dataset presented to date. With the additional 540 genomes, representing a 41% increase to the initial cohort, we can still confirm the absence of large clonal expansions by the end of life, driven either by human orthologues of CH or mouse-specific selection on genes.
4. To address the question of whether HSCs and MPPs differentially contribute to mature blood cells, we undertook targeted sequencing of peripheral blood for mutations present on the phylogenetic tree. This allowed us to study how MPPs and HSCs contribute to peripheral blood. (Referee 2)
5. In this study, we identify HSC population parameters such as population size and cell division rate. Referee 3 contended that it may not be possible to tease apart these values from the data. We present an in-depth theoretical framework confirming the identifiability of N/λ . We also measure the number of HSC and MPP *in vivo* in mice from birth to old age to corroborate our results.
6. In colonic epithelium, somatic mutation rate has been shown to inversely scale with the lifespan of the mammalian species (Cagan *et al.* Nature 2022). Given that we present contrasting findings from blood in the mouse, Referee 1 asked if further mammals could be included. In pilot experiments, we have been able to successfully measure the somatic mutation rate in the blood of naked mole rats - we can confirm that in this species, somatic mutation rate in blood is even lower than expected for lifespan, and lower than the mutation rate in colonic epithelium. These data support our studies and are presented below for the reviewer.

Referee #1 (Remarks to the Author):

In “Clonal dynamics and somatic evolution of haematopoiesis in mouse” Kapadia and colleagues seek to understand how hematopoiesis arises and evolves over the course of the mouse lifespan. They leverage phylogenetic analyses to infer that mouse hematopoietic stem cells and hematopoietic progenitor cells are established during embryogenesis and renew independently of one another over the lifespan. Unlike humans, aged mice do not appear to develop large clonal hematopoiesis or have the loss of stem cell diversity that is characteristic of human hematopoiesis. The authors observe that small clones arise with aging but why such clones expand in humans and do not appreciably expand in mice is unclear. The authors use mathematical modeling to suggest that there are insufficient cell divisions within the comparatively short mouse lifespan to enable detectable clonal expansions which is a widely held opinion, but not one that has been rigorously evaluated before. Although I hold this group of investigators in the highest esteem, from my perspective the scientific observations presented here fall a bit short of the kind of ‘field setting’ observations that are of broad interest to this journal's diverse scientific readership.

We thank the reviewer for their comments. We believe the findings are of broad interest to the diverse scientific readership. This study provides the first *in vivo* estimates of HSC population size and cell division rates in the laboratory mouse, the most widely used mammalian model, in an entirely genetically unmodified setting. We show the timing of HSC and multipotent progenitor establishment during embryogenesis, as well as the transition rates between the cell types, highlighting their differences and similarities.

We find clear evidence for clonal evolution in murine blood, and that a similar set of genes is under positive selection in the blood of mice as are in humans, both during natural ageing and following haematopoietic stress. However, unlike humans, these clones remain small during the short lifespan of the mouse. Our observations contrast with recent data suggesting that somatic mutation accumulation scales with lifespan across mammalian species. In blood, we show that infrequent HSC division rates, and a low somatic mutation rate, result in minimal clonal expansions during normal murine ageing - these clones simply do not have the opportunity to reach the sizes observed after six to eight decades of human life in the absence of perturbations. Our conclusions are backed up by two independent orthogonal mathematical and sequencing approaches.

Taken together, we believe that the breadth of insights presented in this study is field setting, of interest to readers across the fields of somatic evolution, stem cell biology and ageing, and of importance for investigators using the laboratory mouse as an investigative model.

1. The first major finding that mouse HSCs and progenitors can arise independently and contribute to hematopoiesis over the lifespan has been described in a series of papers using murine lineage tracing techniques (Rodriguez-Fraticelli, Nature 2018; Rodriguez-Fraticelli, Nature 2020; Patel, Nature 2022; Li, Cell 2023; Kobayashi, Cell Reports 2023). These lineage tracing tools enable not just phylogenetic reconstruction but have the added advantage of permitting phenotypic characterization (for example with single cell RNA sequencing). Kapadia and colleague's use of spontaneously arising somatic mutations as naturally arising barcodes is elegant but feels like an orthogonal replication of this reasonably well developed line of research rather than a distinct divergence. To increase novelty, the authors could perhaps perform a similar set of analyses in humans where the mouse lineage tracing tools such as those used by the Camargo lab are unavailable (eg there are no “DARLIN” humans).

The four papers listed above from the Camargo group, and the last reference from Kobayashi and colleagues, have quite distinct and focussed aims to those motivating our broader study on haematopoietic stem cell dynamics, clonal selection and somatic evolution over the lifespan of the mouse. The papers above purely focus on stem cell outputs (in terms of numbers of different types of blood cells) in murine haematopoiesis.

In Rodriguez-Fraticelli *et al*, Nature 2018, naive murine haematopoiesis is studied using a transposon genome integration barcoding model in mice up to 5 months of age, to study differentiation paths and blood cell type outputs of long (LT-HSC), short term HSCs (ST-HSC), and multipotent progenitors (MPP2/3/4). In Rodriguez-Fraticelli *et al*, Nature 2020, a lentiviral barcoding technology is used, again to study the blood cell type output of LT-HSCs transplanted into lethally irradiated mice and tracked for 4 months. Of note, ex-vivo manipulation of stem cells was undertaken, and the study was not of naive haematopoiesis. Neither of these studies included old mice to characterise haematopoiesis over the entire lifespan. In the third study (Patel *et al*, Nature 2020), the authors first describe the notion that some embryonically derived cells that are not LT-HSCs (called eMPPs) contribute to haematopoiesis, with Kobayashi *et al*, Cell Reports 2023, suggesting that such lineages are a minority of the population, persist up to 12 months of life, and contribute predominantly to lymphopoiesis. Lastly, in Li *et al*, again from the Camargo group, the DARLIN mouse line is created, with enhanced genomic barcode diversity, and this is used to study HSC migration.

There are several caveats of the approaches above: (i) the percentage of cells labelled is often low, and the ability to recapture generated genomic barcodes is low and often lost with time, (ii) the studies do not study unperturbed cells during naive haematopoiesis, (iii) they do not characterise mouse stem cell outputs across the lifespan, and do not include aged mice.

In our study, we observe that MPPs (defined as ST-HSC in the studies above, and sometimes referred to elsewhere as MPP1) and HSCs (defined as LT-HSC in the studies above), arise independently in both young and extremely old mice, with these populations independently contributing to blood production across the lifespan. We provide data that suggest that MPP and HSC are not taking on such distinct roles during normal murine haematopoiesis beyond some minor differences in contributions to mature blood cell populations. It is important to note that these cell-type definitions were established as a result of outcomes of experiments that imposed highly unnatural conditions, such as total body irradiation followed by bone marrow transplantation, that revealed functional differences between different cell populations. All of our observations are from entirely unperturbed haematopoiesis and healthy unmanipulated mice using naturally occurring somatic mutations as lineage tracking.

Furthermore, this element of our study is just one part. Important messages of our paper also include the mutation rate in haematopoiesis over the murine lifespan, key parameters of the stem cell compartment in mouse, the degree of positive selection on somatic mutations in HSCs, the pattern of somatic evolution and CH clones, and most importantly, how and *why* the pattern observed is different to longer lived humans. For these reasons, we are confident that our study is seminal for the field with broad implications for the fields of evolutionary biology, somatic evolution and ageing.

The Referee raises the suggestion that similar studies to that presented here should be undertaken in humans to increase novelty. In two recent studies, we have used whole genome sequencing and phylogenetic reconstruction to study the clonal dynamics of haematopoiesis in humans. Mitchell *et al*, Nature 2022 undertakes phylogenetic reconstruction over the human lifespan to demonstrate the emergence of oligoclonality after the 7th decade of human life.

Williams *et al*, Nature 2022 utilised a similar approach in blood cancer to time the acquisition of causative driver mutations. As readers may be unfamiliar with our work on phylogenetic analysis of haematopoiesis over the human lifespan (Mitchell *et al*, Nature 2022), we have now shown the phylogenetic trees of human (A) and mouse (B) side by side (for lifespan equivalent ages) as Extended Data Figure 5, and also as shown below (Figure R1.1) to highlight the marked differences in population dynamics and structure between humans and the laboratory mouse by their natural ends of life.

We have also referenced several of the key studies above at the start of the introduction (page 3, para 1) and discussion (page 15 para 3) to provide due acknowledgement.

The current study is the first application of the phylogenetic tree mapping technique to non-human hematopoiesis. The breadth of analyses presented, both phylogenetic analysis of whole genomes from single-cell derived colonies, and sensitive error-corrected duplex sequencing to identify small clones in naive and perturbed haematopoiesis, in addition to the additional work conducted for the revision, provides unprecedented detail into clonal evolution patterns and stem cell dynamics of haematopoiesis over the murine lifespan.

Figure R1.1 and Extended Data Figure 5: Phylogeny topology comparison between aged human and mouse. **A)** Representative ultrametric phylogenies from the three oldest humans described in Mitchell *et al*, Nature 2022. The published trees have been randomly downsampled to 100 colonies (tips). **B)** Aged mouse phylogenies, also downsampled to 100 colonies, to allow comparison of topological structure. The median lifespan for human and mouse species are labelled and were derived from published lifespan figures, as described in Supplementary Note 1. Full murine phylogenetic trees are shown in Figure 2A-B and Extended Data Figure 2.

2. The second major finding is that using error corrected sequencing (Twin Strand DuplexSeq) on a small panel of 9 putative clonal hematopoiesis driver genes and specific hotspots for 15 other genes the authors observe clonal mutations in the 10⁻³ -10⁻⁵ allele frequency range. The gene distribution of these mutations shift in response to different perturbations (eg chemotherapy, infectious exposures). This part of the paper, essentially suggests that the mutations in CH driver genes do happen in mice and cause positive selection. There has been active debate in the CH field on this point and to my knowledge there is no convincing published report one way or the other. That being said, the DuplexSeq method has been actively taken up by the toxicology field more broadly with groups identifying that this method can identify teratogenic exposures in mice and rats (Valentine, PNAS 2020; LeBlanc, BMC Genomics 2022; Dodge, Arch Toxicol 2023;) In these prior papers, mutational patterns were observed to occur genome wide (often in random parts of the genome) rather than focusing on CH genes.

In this study, we show categorically that CH clones, driven by murine orthologs of genes that drive CH in humans, do occur in the mouse, increase in size with age as well as external perturbations, and are under positive selection (dN/dS ratio >1). We also show that the clones are incredibly small, and we provide a plausible rationale as to why this is the case backed by two orthogonal mathematical modeling methods. Our conclusion from computational modelling of the phylogenetic trees using approximate Bayesian computation, as well as analysis using an evolutionary framework on the duplex sequencing clones themselves, independently confirm that the reason the clones are small by the end of life is because the HSC division rate in mouse is not sufficiently frequent to enable larger clonal expansions, particularly, given that the *per cell* fitness advantage appears to be similar in both mouse and human.

We also show that any type of large CH clone - those driven by mutations in genes that drive CH in humans, those driven by any mouse specific gene mutations under positive selection, or indeed 'driverless' CH expansions as observed in humans - are not present in aged murine haematopoiesis. In our revised manuscript, we have added an additional 540 whole genomes from individual HSC and mixed progenitors (LSK cells (Lineage-, Sca1+, c-kit+) which include HSCs and MPPs and other progenitors). After increasing the cohort size by over 40%, we are still able to confirm that no large clonal expansions are observed in the phylogenies towards the end of murine life. These data show that the oligoclonality typical of older human blood does not scale to the shorter-lived mouse. This also suggests that patterns of CH in elderly humans are a function of *time*, and not *ageing*. Taken together, our data provide detailed answers about CH presence, clonal dynamics and positive selection in the mouse, as well as the similarities and differences to human CH. We hope our data and comprehensive narrative definitively settle this long standing debate in the field.

In terms of the studies mentioned above, indeed the Twinstrand duplex sequencing approach can be used to target specific genomic regions, but not the whole genome. The study mentioned by LeBlanc, BMC Genomics 2022, utilises duplex sequencing with a 48kB capture region, and Valentine *et al*, PNAS 2020, also uses duplex sequencing with unclear information on their capture region. Both studies measure somatic mutagenesis in response to DNA damaging exposures. We did not use these targeted capture regions of the genome as we were not interested in measuring mutagenesis in response to chemical carcinogens in this study. In a separate study, where our aim was to measure DNA mutagenesis induced by clinical chemotherapy, we have used an alternative technology of error-corrected duplex sequencing (Abascal *et al*. Nature 2021) to characterise the mutational consequences of chemotherapy carcinogens in human cells at the whole genome level (Mitchell...J Nangalia*, MR Stratton* *et al*, <https://www.biorxiv.org/content/10.1101/2024.05.20.594942v1>). It is worth noting though that carcinogens can alter genomes by causing DNA mutations ("mutagens") but can also alter the

cell populations present through selection on cells with certain mutations (so called “selectagens”). Therefore, future studies that use the mouse haematopoietic system to study the consequences of carcinogens, would need to bear in mind the different clonal dynamics of mouse HSCs as reported in this study, to correctly interpret the selective consequences of the agent studied.

For the duplex sequencing element of our study, we specifically chose to characterise human CH gene orthologues. This is because we wished to ask if the same CH genes were under positive selection in mice but perhaps, missed in our phylogenetic approach which would not detect tiny clonal expansions. We also wished to ascertain if the mouse was a tractable model for studying human CH.

3. From a technical perspective, there have been concerns about the accuracy of TwinStrand technology – for example in Supplementary Figure S8 the biological replicate samples only appear to have a perhaps 50% concordance of identifying a mutation at the 10⁻⁴ to 10⁻⁵ clone size (and there is only one 10⁻³ clone so it is difficult to truly evaluate), this has motivated the development of other technologies like NanoSeq (Abascal, Nature 2021). I am not sure whether the duplex sequencing assay can reliably detect clones in the <0.1% range which is where many of the observed clones occur and whether the mutational spectrum of driver mutations observed in the limited sample size actually does recapitulate human CH.

To reassure the Reviewer regarding the validity of the duplex sequencing data, we have now performed three additional analyses. (i) The first estimates the sensitivity and confidence of our duplex sequencing calls using in-silico data. The conclusion from this (Figure R1.2-R1.4), is that even with error free sequencing, we would not expect fully concordant VAF results for clones of decreasing size. (ii) The second analysis confirms this simulated analysis by using real duplex-sequencing data and performs a mixing of mutant versus wildtype reads to confirm that the concordance of calling low VAF variants in different samples falls with reducing VAF (Figure R1.5). (iii) The third analysis confirms that where low concordance of very small clone sizes is observed in replicates, the presence of the clone can often be confirmed if filtering is loosened to call variants using single-strand consensus reads. Taken together, we hope this additional work reassures the Referee that these clones are genuine, and the pattern of concordance observed is as expected. **These collective added analyses surrounding the accuracy of duplex sequencing have been added to Supplementary Note 3. We refer the reader to these additional analyses in the manuscript on page 12 para 3 and page 35 para 1.**

(i) In silico investigation of the sensitivity and specificity of the duplex-sequencing results

We consider a simple model for SNVs of conditional base calling probabilities for the reference base (R), a mutant base (A) and the two other bases (B,C). For an individual read (or read family/bundle) we model the probability of observing the bases in the following manner:

$$\begin{aligned}
 P(\text{Base is A}) = & \\
 & P(\text{DNA Molecule is mutant A at site}) * \\
 & P(\text{Base called as mutant A} | \text{DNA Molecule is mutant A at site}) \\
 & \quad + P(\text{DNA Molecule is not mutant at site}) * \\
 & P(\text{Base called as mutant A} | \text{DNA Molecule is not mutant at site}).
 \end{aligned}$$

Now $P(\text{DNA Molecule is mutant at site}) = \frac{\text{Aberrant Cell Fraction}}{\text{ploidy}} = \text{VAF}$ where, for economy, we now use the term (true) VAF to characterise the clone. Moreover, we assume there is a base

calling error rate (“epsilon”) ϵ . It is assumed that this results in any one of the 3 incorrect bases to be called with equal probability of $\epsilon/3$:

$$P(\text{Base is Reference}) = (1 - VAF)(1 - \epsilon) + VAF \frac{\epsilon}{3}$$

$$P(\text{Base is A}) = VAF(1 - \epsilon) + (1 - VAF) \frac{\epsilon}{3}$$

$$P(\text{Base is B}) = VAF \frac{\epsilon}{3} + (1 - VAF) \frac{\epsilon}{3} = \frac{\epsilon}{3}$$

$$P(\text{Base is C}) = \frac{\epsilon}{3}$$

For a given baitset wide depth of sequencing, $depth$, we assume that a given site has depth that is Poisson distributed with mean $depth$. For a clone to be detected, it is only required that at least 2 mutant reads are observed. We assume we have a known clone, with $VAF=1e-4$ or $VAF=1e-3$, and with mutant allele A. The A clone is discovered if there are 2 or more mutant “A” reads, and no other mutant reads (“B” or “C”). With these criteria, we can plot the sensitivity for a given error rate, ϵ , shown below.

Figure R1.2: True clone discovery across error rates using multinomial modelling. Estimated sensitivity of detecting a variant at a given site with true VAF 1^{-4} (left) or 1^{-3} (right) across increasing error rates. A range of duplex depth at variant sites are shown.

The above plots show that using single strand consensus sequencing with error rate of $\sim 3-5$ at depth 60,000x provides a sensitivity of 30% for clone sizes of $VAF=1e-3$ or less. However, using duplex depth of 30,000x with an error rate of 1^{-6} to 1^{-7} (as described in Kennedy *et al*, *Nature Prot*, 2014) provides a sensitivity of $>75\%$ (for $VAF=1e-4$) and $>90\%$ (for $VAF=1e-3$).

If we assume the extreme (and implausible) case of error-free sequencing, then the clone detection sensitivity is purely governed by the binomial distribution with a probability of True VAF. Importantly, even if the sequencing was error-free, we would not expect there to be concordance of clone detectability in different samples. In the plots below we can see that for error-free

sequencing at depth 20,000X, we would have a concordance of around 60%. This aligns with the observed concordance seen in the biological replicate samples shown in Supplementary Figure S8.

Figure R1.3. True clone discovery with error-free sequencing. The estimated sensitivity for clone detection at increasing VAFs in the scenario of error-free variant detection. In the absence of an error rate, detection sensitivity can be described with a binomial distribution. A range of duplex depth at variant sites are shown.

Finally, we can estimate the probability of false positive clone detection at a given error rate ϵ . As shown below, when querying a range of feasible duplex-sequencing sensitivities and duplex-corrected sequencing depths, a false positive clone is far less likely than a false negative clone (missing a true event). As an illustrative example, for duplex depths 20,000X to 30,000X and the duplex error rate of $<8e-04$ (estimated error rate of $<1e-06$), the false positive rate is <0.01 . (Figure R1.4)

Figure R1.4 False positive variant detection. The estimated incidence of incorrectly detecting a variant at a given site is shown, using multinomial modeling of detection error rate and site-specific duplex depth.

(ii) Concordance of duplex sequencing data explored through mixing mutant and wildtype reads

The in-silico analyses described above suggest that a true variant clone may not be observed due to insufficient duplex read support, and sensitivity increases with additional duplex depth. In such a scenario, an expected variant would likely be detectable in single-strand consensus reads (Supplementary Figure S4), which require reduced read support and are present at far higher coverage (Supplementary Figure 5), though at the expense of sensitivity.

Therefore, we performed a mixing analysis using our duplex data to estimate the concordance of calling serially lower VAF clones in different samples, with the aim to evaluate 1) the changing concordance of duplex variant detection at differing VAFs, and 2) the degree 'missing-but-expected' clones that can be found in single-strand consensus data.

We selected clones with a large detectable clone size, then generated serial dilutions of input mutant file reads mixed with wild-type file reads to simulate diminishing read support and the subsequent detection of an expected variant in duplex reads. Mutant file reads were diluted by the following percentages: 50%, 20%, 10%, 5%, 2%. Five replicates of each random subsample dilution were used as technical replicates. Read dilution was done with raw, unmodified reads, that is, before any mapping or consensus building steps. Mutant reads were mixed with wildtype reads to the same overall read count as the original data, then analysed using the duplex consensus building and variant calling pipeline described herein. In cases where the expected variant was not detected in duplex consensus reads (either due to lack of read support, or failing to pass stringent filters), we examined matched single strand consensus reads for the variant, and often were able to detect the expected clone.

As shown below in Figure R1.5, we observe concordance among technical replicates when the mutant clone is relatively less diluted from the original data, with reduced variant detection in duplex reads as mutant read support is further diminished. Many missing variants can be rescued when examining single-strand consensus data. With increasing dilution, the variant eventually lacks sufficient read support to build both duplex or single-strand consensus reads, and is not detectable.

Figure R1.5. Dilution of mutant reads and subsequent variant call concordance. For five initially large clones, reads from the original input file (ie supporting the observed clone) were diluted at the indicated proportion with wildtype reads. Five replicates for each dilution factor are grouped. The original clone observed in these unmixed data are shown at the far left column. Clones are presented as described in Fig.4A. Transparency indicates a clone that was not detected with standard duplex filtering, but was detectable within single-strand consensus reads.

(iii) Biological duplicate clones in single-strand consensus reads

As the reviewer noted, in Supplementary Figure 8, there was about 50% concordance in clone calls between biological replicates. In section (i) above, we observe that even if variant detection is error free, one would expect variants sequenced to 20,000X to 30,000X to have an expected concordance of around 60-70% (Figure R1.3).

This is because reads from clones at low magnitude will, by chance, fail to generate a duplex consensus read. Such borderline detectable clones will likely be detectable within single-strand consensus reads, which carry nearly double greater read depth, though at the expense of duplex sensitivity (Supplementary Note 1, Supplementary Figure S4). We examined single-strand consensus reads from the biological duplicate samples and were able to “rescue” missing variants from the paired replicate sample, in half of cases. This confirms that much of the missing replicate clones were lost during duplex consensus building, for example when a clone has insufficient top or bottom strand support to create a duplex read.

We have updated the biological replicate plot in Supplementary Figure S8 to reflect the single-strand consensus clones as shown below.

Figure R1.6 and Supplementary Figure S8. Native CH in biological replicate samples. Shaded and unshaded pairs represent duplex libraries separately prepared from an identical initial blood sample. Clones are presented as described in Fig.4A. Transparency indicates a clone that was only detectable within single strand consensus reads but not duplex consensus reads.

Perhaps the most important observation that confirms the *bona fide* nature of the duplex clones is that they increase in frequency, as well as size, with murine age - artefacts would be equally distributed across mice of different ages.

We have referred to the reader to these additional analyses in the manuscript on page 12 para 3 and page 35 para 1.

The Referee mentions an alternative technology, nanorate-error sequencing (Nano-seq) (Abascal *et al.* Nature 2021). We did opt to use this method to assess genome-wide mutation burden in colonic crypts in this study (Fig.1). However, there were three reasons why we did not use Nano-seq for detecting low VAF CH clones (Fig. 5). (i) There are currently no published targeted gene Nano-seq protocols, only whole genome. (ii) Whole genome Nano-seq would not have had the depth of sequencing to measure the small CH clone sizes that we were interested in. Whole genome Nano-seq is more suited to measuring genome-wide mutation burden and mutation spectra as the number of cells surveyed by Nano-seq is of the order of few hundred to thousand (depending on depth of sequencing). (iii) The reliability of duplex sequencing with our careful filtering and quality control efforts, was our technology of choice, capable of detecting clone sizes at 1 in 10,000 to 1 in 100 cells. Overall, it is worth remembering that the observation from the duplex sequencing data of increased clone sizes and frequency with mouse age would not fit with the behaviour of sequencing artefacts, nor would a positive dN/dS ratio.

4. The technical accuracy of the duplex sequencing aside, the set of observations presented in Fig 4 and 5 highlighting different murine perturbations largely replicate that which is already known in humans (or has been established with other murine clonal hematopoiesis studies). For me what is missing here is something that takes unique advantage of the mouse as a system to show something new or that cannot (easily) be done in humans. For example, if the same cisplatin perturbation is performed across an array of different mouse genotypes can a genetic locus be identified that controls susceptibility to somatic mutation in the setting of cisplatin perturbation. Or if the authors showed that administration of a CDK4/6 inhibitor could prevent CH from arising in mice when measured with their TwinStrand assay (building on observations from He, Science Translational Med. 2017) then the work would point to a potentially novel therapeutic strategy for treating CH. (These are not specific suggestions, but just illustrations of experiments within this system that the authors have established that I would find 'field-setting'. I am confident that the authors will have better ideas than me of exciting applications.)

The purpose of this study was not to use the mouse to find therapeutic opportunities for CH. The key goal of our study was to understand how a short lifespan influences patterns of somatic evolution in blood to better understand clonal evolution and ageing in humans. The metrics of CH we quantify here such as selection coefficients of driver genes, cell division rates, stem cell population size over lifespan, mutation rates, etc, lay the principles for the planning of future experiments targeting CH therapies. For example, if using the mouse as a testing system, one would need duplex error-corrected sequencing, one would need aged mice, and one might consider perturbations to enhance clone sizes in order to avoid the need for deep sequencing, etc.

5. The authors raise an interesting possibility that the observations made about species size and lifespan by Cagan *et al* [Nature 2022] may be tissue specific (eg., the colon is different from blood). One could consider expanding this aspect of the manuscript by profiling blood cells from several different species, which could refine our fundamental understanding about this evolutionary process.

[Redacted text]

[Redacted text and figure]

6. I thought the observation that the rate of HSC → MPP transitions might change over time was interesting (described in lines 175-9). This observation I do not think has been investigated by prior lineage tracing work and could perhaps be expanded upon with different timepoints or

different models (or possibly through the re-analysis of published data from lineage tracing mouse models)

Thank you for this suggestion. The key observation from our phylogenies is that MPPs are phylogenetically more closely related to other MPPs, while HSCs are closely related to other HSCs. This clustering of MPPs and HSCs on all phylogenetic trees would not be the expected pattern expected if all MPPs were generated from HSCs throughout murine life.

However, we do observe *slightly* more ‘mixing’ of HSC and MPP lineages in the younger mouse tree. This may be because we were closer in time to capturing when these populations were being laid out. Alternatively, there may be different rates of transitions early in life compared to later.

While it was challenging to add more timepoints to examine HSC to MPP transitions, we were able to add 242 additional HSC genomes to better ascertain mutation burden across life. Because the individual phylogenetic trees are very small (<20 colonies per animal), these additional data are not of sufficient resolution to make additional inferences using the permutation test based intermixing analysis (see below Figure R1.8 - even 20 samples equally split between HSC and MPP have only an estimated 2% power to detect departures from equal mixing), or the hidden Markov tree model.

We have also provided an additional 298 genomes from the mixed “LSK” progenitor population to ascertain if there is an HSC versus MPP bias in LSK cell production. We also include new flow cytometry data on HSC and MPP numbers over several additional time-points in life, deep targeted sequencing of HSC and MPP lineages in peripheral blood to ascertain their downstream activities, and >200 whole genomes from hematopoietic colonies of naked mole rats.

Overall, the study now includes 1845 whole genomes of haematopoietic stem progenitors (>2000 whole genomes from mice and naked mole rats), and we feel that further work to add depth of insights are better reserved for future studies. We hope that with the breadth of insights presented, the study provides sufficient novelty and advance to the field.

Figure R1.8. Power to detect MPP/HSC shared history 100 populations were simulated using *rsimpop* with HSC and MPP separately laid out and with a modest rate transition from HSC to MPP. Each population simulation was randomly subsampled to specified number samples (20 to 200).

7. From a clinical perspective, does this work have implications for the cell of origin debate for AML or CHIP in humans (eg whether it is derived from the stem cell or from the MPP)? Could the authors perhaps show this by sorting the cells prior to duplex sequencing.

This is an interesting idea- thank you. However, it is technically challenging to sort HSC versus MPP prior to duplex sequencing because the numbers of cells (and subsequent purified genomic DNA) that we would obtain would be insufficient to reach the depth of sequencing required for duplex sensitivity (ie at least 20-30,000x duplex corrected coverage reads). The high input of genomic DNA required during library preparation to reach these corrected depths is only reachable on populations with more starting cells. In addition, we wished to address clone sizes in the peripheral blood rather than stem cell compartments in this part of the study.

Thus far, CH clones and driver mutations have been found across HSC compartments in humans, save for drivers specific to lymphoid selection. Furthermore, even if we did find some differences between HSC and MPP in mice, any such findings would not necessarily be readily translatable to humans. Furthermore, much of the observed differences between murine HSC and MPP have been determined through functional differences observed after total body irradiation and transplantation, making relevance to normal human haematopoiesis unclear.

Nevertheless, our work has important implications. We show that the pattern of age-associated CH and oligoclonality in humans does not scale to mammals with shorter lifespans. Actual time, rather than normalised lifespan and ageing, is the key factor driving the pattern of clonal evolution in blood. This then suggests that CH may merely be a by-product of late 20th century humans who enjoy particularly long life spans. During the industrial era, only 150 years previously, human lifespan was around half of what it is now, and consequently, age associated CH would not have been a clinical issue for most individuals. Recent lengthening of lifespan has allowed weak, to moderately weak driver mutations, to have sufficient time to drive clonal expansions that may precede haematological and non-haematological disease late in life. Our work also highlights the challenges of using mice as a model for studying CH, given the need to use duplex sequencing to identify relatively small clones. Future studies should consider these crucial factors when using murine models for studies of natural CH.

A few more minor consideration:

8. The SBS substitution rate with nanorate sequencing appears to be ~2-4x greater than that of the colonies. Similarly the N/lambda estimates from WGS single cell colonies are also mismatched by a similar factor – how do the authors explain this difference?

We expected the SBS substitution rate to be higher with Nano-seq for 3 reasons (i) the samples used for Nano-seq were whole blood and therefore, comprised both lymphoid and myeloid cells. In humans, the SBS mutation rate in lymphoid cells has been shown to be substantially higher than that in myeloid cells (Machado *et al*, *Nature* 2022). (ii) whole blood Nano-seq would have included DNA from mature cells and in humans, these too have been shown to have a few more mutations than immature HSCs (Abascal *et al*, *Nature*, 2021). (iii) methodologically, whole genome mutation burden using Nano-seq is a derived figure that captures mutations across a limited footprint of the genome, and then scales this figure up to the whole murine reference genome size. On the other hand, standard whole genome sequencing of the colonies detects and counts all the mutations across (nearly) the entire genome, with correction for depth of sequencing. Therefore, 'blind spots' or 'masked regions' in the genome would not contribute to mutation burden in colonies, but would be included in the 'scaling up' of the mutation burden using Nano-seq. Even with the increased mutation burden via Nano-seq, the mutation rate does not

scale inversely with lifespan, as observed to occur with colonic epithelium. We would strongly advise using mutation burdens generated by the whole genome sequencing of clonal samples given that these are measuring mutations in a single cell of origin, unlike Nano-seq that measure mutations across all cell types (myeloid and lymphoid) in the blood.

To explain this better in the manuscript, we have adjusted the text as follows (page 5 para 2): "We did note a non-significant trend towards higher mutation burden estimates from whole blood than HSC colonies - this is likely due to whole blood including lymphoid cells which have higher mutation burdens. Despite whole blood having a mixture of mature cell types and the different sequencing technologies used, these data confirm that somatic mutation rates in blood do not inversely scale with lifespan to the same degree as observed in colon."

With respect to the differences in N/λ , while both the approximate Bayesian computation (ABC) method and branching framework are based on population genetic models, they are supported by differing input parameters and assumptions. The ABC derivations are based on whole genome measurements of mutation burden, while the evolutionary branching framework estimates are based on duplex sequencing-detected mutations within a very narrow targeted region of the genome comprising 100kb. The observations that two orthogonal methods (whole genome vs. targeted duplex sequencing) orthogonally yield similar N/λ estimates is remarkable to us, and increases confidence in our estimates. The differences in N/λ can be explained by the following:

1) In the ABC method, the HSC population is assumed to *grow* (approximately exponentially) according to a birth-death process until it reaches an upper bound on population size. In contrast, the branching framework assumes mutations only occur during an epoch of *non-growing, stable* HSC population size. In both cases, the parameter N is the stable HSC population size.

2) The two alternative estimates are based on mutations from sampling drastically different footprints on the genome: the whole-genome for the ABC method, while a narrow 20-gene panel for the branching framework. While the two alternative estimates of the ratio N/λ (7,918 HSC-years from the ABC method, and 16,500 HSC-years from Blundell et al method), lie within a factor of about 2 of each other, their 95% posterior credible intervals (CIs) exhibit considerable overlap: 2,277.5-20,309.6 from the ABC method, compared to 11,122-21,836 HSC-years from the branching framework. In our view, the closeness of these results from independent methods are striking considering that the population size parameters are not defined in quite the same way in these orthogonal approaches.

As a brief aside, while considering this question from the reviewers and checking our work, we noted that the initial version of the manuscript (mistakenly) reported the 90% CIs for population parameter estimates. In our revised manuscript, we report the more standard 95% CIs and have updated the text and figure text appropriately and provide the corrected CIs in the text above.

3) The branching framework assumes the intronic/synonymous mutation rate is the background for identifying clonal expansions, while the ABC method is agnostic of mutation functional relevance. It is unclear how accurately the intronic/synonymous mutation rate captures the genome-wide background mutation rate, and this may also account for differences in the results between the two orthogonal methods.

We have added a brief note to the manuscript to discuss the potential origin of the differences between these parameter estimations: Differences in the estimates for N/λ from ABC versus the branching evolutionary framework are likely influenced by (i) the ABC method takes into account

population growth observed in the phylogenetic trees, whereas the branching evolutionary framework assumes a stable HSC population size, and (ii) the branching evolutionary framework model relies on using the intronic/synonymous mutation rate as the background for identifying clonal expansions, which may not reflect the genome-wide background mutation rate (page 14, para 3).

9. How have the authors excluded the possibility of SBS18 C>A transversions potentially linked to oxidative damage as not simply representing an artifact in library preparation or DNA sequencing.

We carefully considered both possibilities, and as the Referee suggests, we were expecting them to be artefacts originally. However, there are several observations that confirm the observed SBS18 C>A transversions are bona fide somatic mutations occurring *in vivo*.

(i) Mutations are within internal branches of the tree (ie shared among colonies) and respect the phylogeny. This pattern of sharedness of these mutations across the colonies reflect their acquisition in a common ancestor. If the same artefactual mutation occurs across several colonies that underwent independent library preparation and sequencing, then we would expect such artefacts to occur randomly across colonies, and not only in more closely related colonies in a manner that fully respected the phylogenetic relationships between the colonies. Therefore, the most parsimonious explanation is that the mutation occurred once in a common ancestor of the colonies and that it was subsequently inherited by daughter cells.

(ii) Such mutations are also found in other tissues and species. SBS18 has been observed in very early haematopoiesis in humans (Spencer Chapman *et al*, Nature 2021), in placental tissue in humans (Coorens *et al*, Nature, 2021), and in mouse colonic epithelium (Cagan *et al*, Nature 2022). In this study, we also observe these mutations early in life within mouse haematopoietic stem cells. This raises the possibility of early-in-life exposure to mutagenesis in this pattern, perhaps due to oxidative stress during early development.

(iii) The SBS18 mutations are clonal, akin to other genuine mutations in the phylogenetic trees. Artefacts of sequencing or during library preparation tend to be subclonal within a sample. We show this in the figure below (Figure R1.9), which shows equivalent VAF distributions for variants attributed to either SBS1 or SBS18 that are included in the phylogenetic trees.

Figure R1.9: Colony VAF for SBS1 and SBS18 mutations. The mean colony VAF for all variants attributed to SBS1 and SBS18 across all colonies are shown as density histograms.

(iv) Using the same technology, sequencing approach and library preparation, SBS18 is not found during adult haematopoiesis across a variety of scenarios (Mitchell et al, Nature 2022, Williams et al Nature 2022, Machado et al Nature Comms 2023). Therefore, their presence near the tops of the trees in this study confirms their acquisition early in life during murine development. It is possible that this may reflect rapid cell division, as recently, we have also observed SBS18 within very rapid clonal expansions (eg chronic myeloid leukaemia phylogenies, unpublished data).

10. Are the estimates of mutation rate consistent with the estimates of rate of cell divisions (70k cells, divisions every 6 weeks)

Thank you for this thoughtful question. We consider the predicted mutation rates and cell division rate to be consistent with one another. On average, we observe about 45 mutations (95% CI 38-44) per year. In a single year, with a predicted division rate of every 6 weeks (95% CI 1.8-13.2 weeks), a single HSC would have divided 8 times (95% CI 4-22 times). We estimate that every cell division, at least near the start of life, generates 1.80 mutations (CI 1.5-2.2), resulting in ~15 mutations/year. The remaining ~25 mutations per year reflect the slightly increased rate of time dependent endogenous DNA damage, or reduced repair, in mice versus humans. Overall, the higher mutation rate in mice is still compatible with the increased HSC division rate.

Separately, we note that mutation rate does not necessarily scale with cell division rate. For example, neurons (post-mitotic) have a higher mutation rate than spermatogonial stem cells (Abascal *et al.* 2021, Moore *et al.* 2021). So a linear relationship between cell division rate and mutation burden is not our expectation, based on the literature.

11. I feel that the authors are overstating the notion that the field is stuck in a rigid notion of 'classical hematopoiesis models' as a strawman when work over the past 5 years has presented a more nuanced view and would consider revising both the introduction and discussion accordingly.

This is a fair comment and we have revised the introduction, results and discussion in several places to acknowledge this. Examples are shown below -

This process relies on a hierarchy of progenitors that successively amplify cellular output towards fully differentiated blood cells. All are believed to ultimately derive from a pool of rare haematopoietic stem cells (HSCs), a pool maintained in a relatively protected state to support blood production throughout life, with recent studies suggesting heterogeneity within these long-term self-renewing haematopoietic cells. (page 3 para 1)

Classical models of the haematopoietic differentiation hierarchy propose that MPPs derive from HSCs. In recent years, a more nuanced and dynamic picture has emerged, with the identification of additional self-renewing progenitor compartments. (page 7 para 1)

Classical models of blood production depict HSCs at the very top of the haematopoietic differentiation hierarchy, beneath which all blood cell types emanate. Recent studies suggest additional heterogeneity at the top of this haematopoietic hierarchy and more nuanced and dynamic haematopoietic self-renewing dynamics. (page 15 para 3)

12. A very minor point in Fig 1D – the two-tone HSC (left) MPP (right) color scheme are visually pleasing but the meaning is a bit hard to decipher without connecting it to the colors in 1B. For Fig 1D, I would suggest considering using different marker shapes (eg dots and Xs or colored dots and open dots) for HSCs and MPPs or making the violin around MPPs shaded light and dark or alternatively just labeling on the x axis HSC and MPP (or H and M).

Thank you for this helpful suggestion. We have now made these changes (dots for HSCs and squares for MPP, and in addition, we have labelled each group as “H” and “M” underneath). The improved new Figure 1D is below (Figure R1.10). We have updated the Figure legend accordingly.

Figure R1.10 and revised Figure 1D. Comparison of SBS burden between HSC- and MPP-derived colonies from the same mice. SBS burden from HSCs shown as circles and burden from MPPs are shown as squares. H, HSC; M, MPP, shown above donor ID.

Referee #2 (Remarks to the Author):

The authors apply very powerful analysis based on WGS of colonies derived from murine HSCs and phenotypically-defined putative downstream MPPs to gain really significant and immediately impactful insights into steady state hematopoiesis and hematopoietic aging, not dependent like all prior studies on transplantation as a readout for HSC function or on complex transgenic models with low resolution. While the concepts and methods applied are not novel (as some of the authors have used the approaches very very productively over the past several years to understand human hematopoiesis at steady state, with aging, post-transplant and in disease states), this is the first paper investigating the murine model, which is the absolute mainstay of all biomedical research on hematopoiesis and blood diseases, so the really stark differences uncovered in very basic parameters in mice vs humans are of great interest and importance to try and understand whether and how to utilize these models in basic discovery and in therapeutic development. The major findings include: 1) mutation rate for HSCs in mice is more than two-fold that of human HSCs, but lower than expected from differences in lifespan and the large differences for instance in murine vs human colonic mucosa. 2) The concept of LT-HSCs, able to self-renew through multiple rounds of serial transplantation vs ST-HSCs or MPPs able to engraft long term in primary but not serial transplants, the crux of much murine research and paradigm crafting over the years, appear to have no meaning in steady state non transplanted hematopoiesis in mice. So much of what we think we know about HSCs has been based on this distinction, and clearly that needs a major reconsideration based on this paper. This helps explain why already quite a few concepts don't translate from mouse models to humans based on what appears to be quite an artificial distinction. 3) HSCs/MPPs in mice have quite different trajectories over the lifespan in mice vs humans, with murine cells continuing to increase in number over time, reaching total numbers equivalent to humans despite much smaller size and shorter lifespans. 4) With aging, the massive clonal expansions seen in humans (ie "CH" or age-related clonal hematopoiesis) are simply not seen in mice. Much smaller expansions, some mutations in the same genes as in humans, can be found if look hard enough (also reported previously by a Japanese group but this goes far beyond that in terms of sophistication of analysis and explanations), but only with application of major stressors such as chemotherapy or chronic infection do the clones become detectable, and almost never reach the size seen in humans-making mice a questionable model for CH in humans. Every part of this study is interesting and will be widely read and discussed by anyone interested in stem cells, development, normal or abnormal hematopoiesis, aging, and transplantation and gene therapies. There are a couple of additional experiments that would increase the significance and complete the story and place it in context, as detailed below. The paper is very well-written and most of the figures are clear and meaningful. Methodologies in general are completely and carefully explained.

We thank the Reviewer for their highly positive comments on our study.

1. The core data in the paper is all based on one mouse strain with relatively long lifespan and known slower HSC cycling. Even the data on clonal hematopoiesis via targeted sequencing added two strains that are a C57F1 hybrid or a pretty unique 4 strain mix. Would have been of interest to instead include a strain with properties more extreme in the other direction ie DBA, with shorter life span, faster HSC cycling. Given that C57 used for >>90% of murine HSC and transplant studies, makes sense that the major work was carried out with only that strain, but at least this drawback should have been noted and discussed as a caveat and of importance for future studies if only one strain was utilized.

We agree that it would be interesting to undertake similar studies in murine strains where HSC numbers of division rates are dramatically altered. We sought out investigators working with such

strains, but none had sufficiently aged animals available. Given that we have focused on quite old mice, we would have to wait another ~2 years to complete such experiments, so we are not able to perform these for this paper. We are pursuing these experiments for future analyses. We have acknowledged this limitation in the discussion as follows: **It is possible that mouse strains thought to have higher HSC turnover (Chen, Astle, and Harrison 2000), or maintained for longer periods in more “wild”-like microbial environments, would exhibit higher levels of native CH, and additional studies to characterise such strains and environments would be of interest. (page 17 para 2)**

2. Phenotypically the ratio of HSC to MPPs changes between the two time points studied, but the number of colonies used for WGS and SNV discovery and phylogenetic reconstructions and thus basically entire paper based on approximately equal numbers of HSC and MPPs single cell colonies utilized. Could this throw off any of the models? Was this accounted for in any way? Does it need to be?

Thank you for this important and helpful question. We agree it is important to confirm our cell sampling decisions (i.e. purifying as many HSCs and MPPs as feasible) would not skew any models that take as their input as the relative HSC and MPP counts. We sampled roughly equal numbers of HSCs and MPPs, but as noted, the ratio in bone marrow of these progenitor populations at these ages are not equivalent.

We have first added new data to the paper on mice from additional ages, both for somatic mutation burden quantification (and refinement to the rate of mutation accrual), as well as for estimating the number of HSCs and MPP1 at various times over the murine lifespan (Figure R2.1, revised Figure 3B). This confirmed that phenotypically-defined HSCs and MPP compartments increase over life. (HSCs, as a proportion of LSK progenitors, increase over life, as do the proportion of LSK cells within the whole bone marrow. Whilst MPPs remain at a similar proportion of LSKs, the overall LSK proportion increases over life). These data are in line with the growing population of MPP and HSC observed both from the population estimates using approximate Bayesian computation and from the population trajectories observed using the programme *phylodyn*, providing orthogonal validation.

Figure R2.1 and revised Figure 3B: Haematopoietic stem and progenitor cell (HSPC) prevalence during murine ageing. The relative abundance of total HSPCs (left) and individual HSPC subpopulations (right) are compared. MPP^{LY} are lymphoid-biased progenitors, MPP^{GM} are myeloid-biased progenitors, based on current immunophenotypic definitions.

With respect to our HSC/MPP transitions, we appreciate that we have sampled roughly equal numbers of HSCs and MPPs for the hidden Markov tree modelling. However, qualitatively, the clustering of HSC tips and MPP tips on the phylogenetic trees will hold irrespective of the sampling regime. This banding of clades of HSCs and MPPs across all phylogenies in this study is, perhaps, the most important observation for divergence of these two populations early in life.

Nevertheless, we wished to check if our inferences for rejecting an “HSC-first” ontogeny would be affected by sampling regime. For the old animals, we downsampled HSC and MPP numbers separately by a factor of 0.33, 0.5 and 0.66 and found that the HSC-first likelihood ratio test is still highly significant (Table R2.1).

Label	Cell Type totals across 3 animals		HSC Ratio	Total Cell Count	Log Likelihood		p_LRT_HSCFIRST
	HSC	MPP			hscfirst	full	
HSC(116)/MPP(322)	116	322	26.5%	438	-241.64374	-213.78404	8.36E-14
HSC(177)/MPP(322)	177	322	35.5%	499	-293.16328	-265.01198	6.21E-14
HSC(233)/MPP(322)	233	322	42.0%	555	-339.75516	-305.97841	2.05E-16
HSC(353)/MPP(322)	353	322	52.3%	675	-418.77899	-377.79256	1.38E-19
HSC(353)/MPP(212)	353	212	62.5%	565	-337.50523	-304.32629	3.76E-16
HSC(353)/MPP(161)	353	161	68.7%	514	-289.49436	-265.83296	6.02E-12
HSC(353)/MPP(106)	353	106	76.9%	459	-232.35209	-215.63024	7.34E-09

Table R2.1 HSC and MPP sample size adjustments and hidden Markov tree based ontogeny inference. The maximum likelihood of two nested models is considered “full” where 4 parameters are estimated representing the transition probabilities of EMB->HSC, EMB->MPP, HSC->MPP, MPP->HSC, and “hscfirst” where the transition probability of EMB->MPP is fixed at 0, and the other 3 parameters are estimated. For details see Supplementary Note 2.

Next, we wished to more formally address this. We have previously developed an HSC population simulator *Rsimpop* (used in Williams *et al*, Nature 2022), and we first updated this package to support the simulation of continuous differentiation between tissue compartments (embryonic cells, HSCs and MPPs). This allows the user to specify rates for migration or differentiation between any number of cellular compartments. Once the cell compartment specific population limits are reached, the death rates in the cell compartment balance any cell production. We then used this population simulator to simulate 3 phylogenetic trees (equal population size of HSC and MPP, 10x more HSCs than MPPs, and 10x more MPPs than HSCs).

We next checked if our hidden Markov tree model, which we used to explore if it was likely that HSCs and MPPs were being laid out independently, was sensitive to different background HSC/MPP population sizes, when HSCs and MPPs are sampled in roughly equal numbers. One example shown below (Figure R2.2) reassures that despite unbalanced population sizes, the hidden Markov tree model can broadly capture the ground truth of transitions.

Figure R2.2 Simulation of unequal HSC and MPP population sizes and subsequent hidden Markov transition modelling Inferred Viterbi trees alongside the ground truth simulated trees (0:2 green is HSC and 0:3 purple is MPP). Informally we have a decent correspondence despite sampling bias.

Lastly, we considered if sampling decisions could influence our estimates of population size (N) or cell division rate. Both the *phylodyn* trajectories and approximate Bayesian computation (ABC) analyses rely on phylogeny topology (where we include branch lengths in our broad use of the term topology here) only, and thus are entirely 'blind' to if the underlying colonies are from HSCs or MPPs. The *phylodyn* analyses consider the HSC and MPP trajectories as entirely independent events. Given the similarity (or lack of drastic difference) in tree topology between HSC and MPPs, and that the *phylodyn* central estimates follow close trajectories, we can expect cell division and cell death rates to be similar between both populations. Additionally, the ABC analysis only uses the phylogeny topology as 'ground truth' but is totally agnostic to if specific branches are derived from HSCs or MPPs. Thus, these supporting models are driven by tree topology only and are 'blind' to sampling decisions or ratios between HSC or MPP. The selection of more colonies (either HSC or MPP) only serves to increase the resolution in topological structure of the phylogeny and the corresponding breadth of credibility intervals.

In summary, these additional analyses reassure us that the sampling approach used does not bias our main conclusion that HSC and MPP are essentially laid out independently early in life. In future work it would be beneficial to perform a large-scale ABC analysis using the simulation approach that accounts for continuous differentiation between blood cell types.

3. At least to this reader, the data showing little evidence for HSC to MPP transitions is the single most important point of paper, so when first introduced based on the phylogenetic trees, a bit more clarity would be helpful—a more extensive even if succinct explanation for instance of the approach resulting in Figure 2C and D somewhere other than in the legend and wading through

the Supplemental Notes would be helpful. This is true for large chunks of the paper and realize it can't all be in the main text, but the converse of no explanation is confusing to the reader.

We take this point on board and sincerely apologise for this. We have added further explanations within the main text throughout that describe the underlying approaches in readable language, which we hope will facilitate interpretation of Figures 2C and 2D. Additionally, we have improved the legends to include greater detail about the underlying method. These changes are reproduced below:

Main text description of Figure 2C: The clear separation of MPPs and HSCs clades suggests that most HSCs are derived from HSC self-renewing divisions, and most MPPs are derived from MPP self-renewing divisions, with each population independently self-renewing in parallel throughout life. If HSCs and MPPs were more closely related, as might be the case if MPPs were recently generated from HSCs, then one would expect these two cell types to be intermixed across the phylogenetic tree, with individual clades (cells derived from a common ancestor) containing branches of both types. Instead, we observed that clades contained more cells of the same type than would be expected by chance (Fig.2C). This phylogenetic separation of HSC and MPPs provides strong evidence that these two populations independently contribute to blood production in the mouse.

Figure 2C Legend: To determine the degree of phylogenetic relatedness between HSC and MPP, we measured the amount of HSC-MPP mixing within clades. If an MPP had a recent HSC ancestor, clades should contain both cell types. We thus compared the “observed” versus “expected-by-chance” clade mixing behaviour. The mixing metric for a clade is the absolute difference between the proportion of HSCs in a clade and the expected value under equal sampling 0.5; this metric is then averaged for all clades in a phylogeny. The vertical bar reflects the observed average clade mixing metric within the constructed phylogenies. The filled distributions reflect average clade mixing metrics that would be expected by random chance (or more frequent intermixing of HSCs and MPPs, and were generated by reshuffling the tip cell identities within the tree. HSC or MPP colonies are designated as being in the same clade if they share a most recent common ancestor after 25 mutations, corresponding to early foetal development. Only clades with more than 3 colonies are considered.

The main text descriptions related to Figure 2D together with legend updates are depicted on page 8 and 19. We have further improved the readability of several sections of the manuscript as highlighted in red.

4. It was disappointing that the authors did not look at the blood and blood lineages (in the more usual sense ie myeloid/lymphoid) particularly regarding the question of whether differentiation/mature cell output from HSCs/MPPs reflects what is happening at the HSC/MPP level. Presumably there is some reason the two populations exist and are so distinct and for a really important paper such as this one with huge implications for use of mice as models for human biology and disease, this seems a rather large missing piece of information. Sorted populations transplanted in mice show that HSCs or MPPs transplanted separately result in normal blood counts, survival etc but the authors have the opportunity to see what happens in steady state normal hematopoiesis, seems rather important to look at. One hopes there is DNA from blood collected at the time of HSC/MPP harvest/purification from those 6 animals available to perform these studies.

Thank you for this suggestion. Fortunately, we had sorted LSK progenitors and stored DNA from whole blood from the 3 aged animals. We have now undertaken two separate approaches to address this question:

- 1) additional single-cell genomes from mixed hematopoietic progenitors (LSK cells) compartment of the same mice, and
- 2) attempting to re-capture specific HSC and MPP variants from the peripheral blood using targeted sequencing.

These data are presented below, have been added to the main text on page 10-11, the methods on page 37-38, and included in the manuscript as Extended Data Figure 7, Extended Data Figure 8, and Extended Data Figure 9.

LSK-derived single-cell genomes

We reasoned that because the LSK population is a population of mixed progenitors with cells that ostensibly derive from HSCs and MPPs, we may be able to identify at least some progeny of the HSCs and MPPs within this population. To this end, we sorted and grew single cell derived haematopoietic colonies from a total of 298 LSK cells (Lineage-, Sca1+, c-kit+) from the three 30-month old mice, to build into the existing phylogenetic tree of HSC and MPPs. This was done with our standard HSC flow cytometry staining panel and index sorting so we could identify what type of LSK cell bore which genome. This would allow us to ask if HSCs and MPPs differentially contributed to downstream progenitors such as MPP^{GM} (granulocyte/macrophage biased MPPs) and MPP^{Ly} (Lymphoid biased MPPs).

The three expanded phylogenetic trees are shown below (Figure R2.3). There was no discernible bias in generation of LSK cell types from MPP versus HSC from visual inspection of the trees.

For the Referee's convenience, we show adjacent an example of an HSC ancestor giving rise to MPP^{GM} and MPP^{Ly} (left), and the same offspring types stemming from an MPP ancestor.

We next evaluated more formally if the myeloid-biased progenitors, MPP^{GM}, and lymphoid-biased progenitors MPP^{Ly} were preferentially derived from either HSCs or MPPs. We separately evaluated the interrelatedness of MPP^{GM} and MPP^{Ly} to HSCs, and then to MPPs. In this approach, we considered clades that contained MPP^{GM} (or MPP^{Ly}) and evaluated if they were more phylogenetically linked to HSCs or MPPs than expected by chance. We observed that MPP^{GM} cells did not preferentially derive from either HSCs or MPPs beyond random chance (Figure R2.4). MPP^{Ly} cells showed a similar pattern. These data suggest both HSCs and MPPs produce downstream LSK progenitors at seemingly similar proportions. However, we note the caveat that these data are limited by a relatively low number of sampled MPP^{GM} and MPP^{Ly}, since we were purifying the LSK compartment as a whole and growing single colonies independent of cellular immunophenotype. These data are presented below and included in the manuscript as Extended Data 7.

Figure R2.3 legend and Extended Data Figure 7A-C: Extended phylogenetic trees of the HSC, MPP, and LSK progenitor compartments. Extended phylogenies were created using the pattern of sharing of somatic mutations among HSCs (blue), MPP^{All}s (red), and the mixed LSK (Lineage-, Sca1+, c-kit+) hematopoietic progenitor compartment. The LSK compartment contains HSCs and MPP^{All}s, and additionally contains the myeloid-biased MPP^{GM} (orange) and lymphoid-biased MPP^{LY} populations. LSK subcompartments were identified at time of single cell sorting using a consensus definition (Challen *et al* 2021). Each tip represents a single colony. Branch lengths represent mutation numbers.

Figure R2.4 and Extended Data Figure 7D-E: Clade mixing metrics for MPP^{GM} and MPP^{LY} colonies used to evaluate interrelatedness with HSC and MPP. HSC, MPP and MPP^{GM} or MPP^{LY} were designated as being in the same clade if they share a most recent common ancestor after 25 mutations, corresponding to early foetal development. Only clades with more than 3 colonies are considered. The vertical bar reflects the average clade mixing metric observed in the constructed phylogenies, while distributions reflect the average clade mixing metric expected random chance, estimated by reshuffling the tip states. If the observed value (vertical bar) significantly deviated from random chance (filled distribution), then there would be minimal overlap between the observed data and the random reshuffling distribution. The average clade mixing metric for MPP^{GM} compared to HSCs (blue) and MPPs (red) is shown in **D**). The similar measure of interrelatedness of MPP^{LY} to HSCs and MPPs is shown in **E**).

Recapture of HSC and MPP variants from the peripheral blood

In our second approach, we wished to ask if differentiated peripheral blood (PB) cells were equally descended from HSCs and MPPs, or whether peripheral blood was preferentially derived from one progenitor over the other.

To this end, we designed a targeted DNA bait set (Agilent SureSelect custom capture) for mutations on the phylogenetic trees of the aged mice, and then queried gDNA purified from PB for tree-specific mutations using high-depth targeted sequencing.

The baitset was designed to capture mutations on the phylogenetic trees of all 3 aged mice (MD7180, MD7181 and MD7182), and to cover mutations found in HSCs, MPPs and LSKs. The baitset comprised of:

1. All variants on shared branches that pass SureDesign's "moderately stringent filters".
2. All variants on a random subset of private branches that pass SureDesign's "most stringent filters". Approximately 25% of the private branches of each mouse were randomly selected.
3. The exons and 3' and 5' UTRs for all CH genes used in our duplex sequencing panel (*Dnmt3a*, *Tet2*, *Asx11*, *Trp53*, *Rad21*, *Cux1*, *Runx1*, *Bcor*, *Bcor11*, *Ppm1d*, *Sf3b1*, *Srsf2*, *U2af1*, *Zrsr2*, *Idh1*, *Idh2*, *Gnas*, *Gnb1*, *Cbl*, *Jak2*, *Ptpn11*, *Brcc3*, *Nras*, *Kras*).

Target-enriched libraries were generated from PB DNA from the 3 mice and sequenced with Illumina Novaseq platform. We captured 15,457 somatic mutations in peripheral blood that matched across the trees of the three aged mice. Baits were sequenced to a depth of approximately 2600x, with median depths of 2616X, 2549X and 2628X for MD1780, MD7181 and MD7182 respectively.

As an illustrative example, Figure R2.5 shows phylogeny for sample MD7180 is decorated with the specific mutation and branches that were targeted and captured in matched peripheral blood.

As designed in our baitset, we capture both mutations shared between several colonies (internal branches) as well as private to single cells (private branches).

MD7180: Targeted recapture mutations from **Shared** and **Private** Branches

Figure R2.5. Targeted recapture of progenitor mutations in peripheral blood. Matched peripheral blood underwent targeted sequencing for loci detected in phylogenetic trees. In the MD7180 extended phylogeny (built from HSCs, MPPs, and LSKs), mutations recaptured (and thus shared) with peripheral blood are mapped to their respective branch and shown coloured. Red branches denote recaptured mutation shared between multiple colonies, thus located within internal branches of the phylogeny. Blue branches are recaptured mutations detected in single colonies only (private branches). Short branches, especially at the top of the phylogeny, have been magnified to facilitate visualisation.

Importantly, we recaptured mutations in the peripheral blood that were acquired in both ancestral HSCs and ancestral MPPs, suggesting that both these cell types actively contribute to mature blood production. Mutations private to single progenitors (that we sampled) were subclonal, occurring below 0.1% VAF in peripheral blood (Figure R.2.6).

The VAF of a tree variant in peripheral blood can be used to measure how much a single lineage contributed to the peripheral blood. For example, if a single cell or lineage contributed avidly to differentiated progeny, then its mutations would be seen at high proportion (VAF) in peripheral blood.

We next examined the maximum VAF for peripheral blood mutations matching single branches in the phylogeny. We *qualitatively* observed relatively higher VAFs for private MPP mutations compared to HSCs, suggesting that an occasional MPP lineage had a greater than expected representation in PB (Figure R.2.6). Thus, we explored this difference further.

Figure R.2.6 and Extended Data 8A: Mutation overlap between phylogenies and peripheral blood. Phylogenies for three aged mice (as described in Figure R2.3 legend) constructed to only include private branches targeted with the peripheral blood baitset. Branch shading indicates the maximum VAF among shared variants captured in peripheral blood. The sampled cell immunophenotype is indicated by dot colour at the bottom of each private branch.

To quantify the degree of HSC and MPP contribution to peripheral blood, we estimated the posterior distribution of true VAF for every (shared) mutation captured with our targeted baitset.

We estimated the true peripheral blood VAFs of the baitset variants using the hierarchical constraints imposed by their phylogenetic relationships by using the Gibbs sampling method developed in Spencer Chapman *et al* (DOI:[10.21203/rs.3.rs-2868644/v1](https://doi.org/10.21203/rs.3.rs-2868644/v1)). Then, for each time t , and for each branch that overlaps t , we estimate the VAF of a hypothetical mutation at time t . This is done by arranging our baitset variants in descending estimated VAF order at equally spaced intervals down the branch and then linearly interpolating the VAF at time t based on the estimated VAF of the neighbouring mutations. The aggregate VAF at time t for a tree or lineage is then calculated as the sum of the estimated VAFs of the overlapping branches at time t .

We examined the aggregate VAF across baitset mutation-specific phylogenies (Figure R.2.6) to evaluate the proportion of PB produced by the ancestral lineages of the sampled MPPs and HSCs. While both HSC and MPP ancestral lineages give rise to peripheral blood, we observe a slight bias towards increased representation of ancestral MPP lineages compared to HSCs, though this difference is subtle (Figure R2.7).

Figure R2.7 legend and Extended Data Figure 8B: VAF trajectories of HSC and MPP variants shared in peripheral blood. The aggregate VAF across molecular time is calculated using the Gibbs sampling method. Early molecular time corresponds to further in the ancestral past. Shaded regions denote 95% confidence intervals of the VAF estimates.

Finally, we re-examined our baitset-specific trees to determine if the slightly greater MPP contribution was due to specific, single, branches, or observable across all branches in the phylogenies. Given the divergence in MPP contribution to peripheral blood occurred early in ancestral life, we expected the latter.

We inspected the VAF distribution for each branch harbouring any baitset-detected mutations. Qualitatively, we observe greater mutation overlap with peripheral blood in MPP phylogenies. That is, we observe mutations shared between the MPP tree and peripheral blood encompass a greater proportion of tree variants, with sharing further down individual branches, than seen in the baitset-shared HSC trees (Figure R2.8).

Collectively, we have made considerable effort to understand if HSCs and MPPs are differentially contributing to downstream blood cells. Our approaches have not found major differences, and perhaps there are some increased progeny of ancestral MPPs represented in the peripheral blood. This may be due to 1) increased proliferation of MPP descendants or 2) differences in population sizes of the two compartments earlier in life; however we cannot distinguish these possibilities from our data. More importantly, we show that HSCs do contribute meaningfully to peripheral blood, unlike some previous studies have suggested (Sun *et al.* 2014). The self-renewing differences between MPP and HSC appear more relevant in the transplantation assays that were originally used to define these populations over the past two decades. In absence of assumptions from earlier studies and markers to identify HSC versus MPP subpopulations in the first place, we are not convinced that we would have picked these populations out as different based on the data we report here on mutation burden, mutational signatures, population dynamics and progeny *in vivo*.

These findings have been included in the manuscript on page 10-11.

Extended Data Figure 9

Figure R2.8 and Extended Data Figure 9: Peripheral blood VAF of variants shared with HSCs and MPPs. Baitset mutation-specific HSC and MPP phylogenies are shown for each 30-month mouse. Each branch shows mutations that were detected in peripheral blood in descending VAF order. On each branch, a row denotes a single variant mapped to that specific branch. Red fill denotes the observed PB VAF for the variant. VAF is denoted on a log scale from 10^{-5} to 1 ; internal divisions are marked from left to right at VAF 0.0001, 0.001, 0.01, and 0.1. HSC trees are shown on left with blue dots at terminal branches; MPP trees are shown on right with red dots. Trees are downsampled to allow equivalent comparison between HSC and MPP branches. Only variants seen in peripheral blood with a depth > 100X are shown.

5. It would have been of interest to perform transplants from at least some of the primary animals to see happened to the HSC and MPP trees in that setting and whether there was any difference from what has been observed using sorted HSCs and MPPs where transition from HSC to MPP must be occurring, but the authors have the chance to see how much of a bottleneck that process might entail.

We agree this would have been of interest but unfortunately, we did not perform transplants at the time the animals were sacrificed. Of note, a recent paper has looked at this in humans <https://www.sciencedirect.com/science/article/pii/S0006497121057311>, <https://www.researchsquare.com/article/rs-2868644/v1> and compares the evolutionary dynamics of clones within a donor, and in parallel, tracks the same clones within the corresponding allogeneic recipient. Selection is observed to happen at different stages, both at the point of clonal outgrowth within the recipient bone marrow, but also in terms of which donor HSCs successfully reach the recipient bone marrow in the first place. This study also quantifies the number of engrafting HSCs.

6. For being able to easily appreciate the marked differences between human and murine phylogenetic trees, it would be important to include human trees in this paper, rather than having to go back and find these in prior publications. Regenerating them with the mutational clock dimension translated into lifespan for both the mouse and human to allow direct comparisons would be awesome. The whole point of the paper is really summed up in being able to appreciate that comparison in terms of where coalescences occur for instance.

Thank you for this thoughtful suggestion. We have added a new Extended Data Figure 5 to the revised manuscript, in which we show the aged mouse trees alongside the aged human phylogenies from Mitchell et. al, Nature 2022 to convey the drastic topological differences between these aged HSCs' phylogenies. The figure is shown above as Figure F1.1 and also replicated below.

Of note, the human and mouse phylogenies were assembled using differing numbers of input whole genomes, so to allow a direct comparison, the provided trees have been randomly downsampled to 100 colonies each.

Rebuttal F1.1 and Extended Data Figure 5 Phylogeny topology comparison between aged human and mouse. **A)** Representative ultrametric phylogenies from the three oldest humans described in Mitchell et al, Nature 2022. The published trees have been randomly downsampled to 100 colonies (tips). **B)** Aged mouse phylogenies, also downsampled to 100 colonies, to allow comparison of topological structure. The median lifespan for human and mouse species are labeled and were derived as described in Supplementary Note 1. Full murine phylogenetic trees are shown in Figure 2A-B and Extended Data Figure 2)

7. To this reader, the section on explanations for mutation rate being higher in mouse than human and types of mutational signatures found is out of place-would consider moving to right after the initial mention of the trees (but before all on HSC vs MPPs), because now it breaks up that set of studies/concepts. Or if there isn't room, to me much of this could move to Supplemental Notes, since no really firm conclusions.

Thank you for this suggestion. We have done as the Referee suggests, moving the section to just after introducing the trees. (page 5)

8. Are HSCs that have mutations more likely to disappear in humans than mice as another possible explanation for the lower apparent mutation rate? Known that DNA damage/breaks/some types of mutations results in P53-mediated apoptosis in human cells versus repair in murine.

Thank you for this thought-provoking question. We note the mutation rate in human HSCs is half that of the mouse (~20 versus 40 mutations per year). If the rate of somatic mutation accrual in blood scales with lifespan, as demonstrated for colonic epithelial cells (Cagan *et al*, Nature 2022), then we might expect murine HSCs to have a mutation rate of ~500 mutations per year.

If we understand the Referee correctly, they are raising the possibility that murine and human HSCs may tolerate mutations differently in terms of predilection for subsequent DNA repair and apoptosis. For example, hypothetically, if human HSCs with more mutations resulted in apoptosis, then this may cause the observed human haematopoietic mutation rate to be lower. However, in our opinion, there are two lines of evidence that make this unlikely. First, end of life mutation burden is 10-fold higher in humans than mice, suggesting that human HSCs can tolerate more mutations. Secondly, during increased mutagenesis in humans (eg exposure to mutagenic chemotherapy), mutation burdens in HSCs rise to 8000-10000 mutations (during mid-life) (Mitchell...Nangalia,Stratton, <https://www.biorxiv.org/content/10.1101/2024.05.20.594942v1>). These data suggest that genome-wide mutation burden tolerance can increase by several fold and still be observed in HSC outputs.

However, the Referee's point could equally apply to mice, wherein a mouse HSC might have a "limit" on the number of accrued mutations before apoptosis (versus repair). We believe that such a mutation burden limit would appear as a plateau in mutation burden during ageing. Given our initial manuscript only evaluated two timepoints (3 months and 30 months), we could not evaluate for the existence of such a plateau during intermediate timepoints.

In our revised manuscript, we have added 242 additional colonies from mice at additional ages across the mouse lifespan: one day old pups (n= 3 animals, n=40 colonies), 12 months (n= 3 animals, n=32 colonies), 18 months (n= 5 animals, n=105 colonies), 24 months (n=5 animals, n=58 colonies), and 30 months (n=1 (control) animal, n=7 colonies). With this additional granularity, we still observe a linear rate of mutation accrual in life and the absence of a plateau or limit on HSC mutation burden.

Similarly, we observe, with flow cytometry quantification of progenitor population prevalence, we confirm an increase in the proportion of HSCs throughout life, suggesting there is no plateau or limit in HSC persistence.

We have updated Figures 1C and 3B in our revised manuscript to reflect these changes, as well as updating values for mutation rates in the manuscript. These updated figures are shown below. Overall, this suggests that there is no imposed limit during the mouse lifespan on mutation burden accrual (eg due to cell death).

Rebuttal Figure 2.9 and revised Figure 1C: Burden of individual single base substitutions (SBS) observed in HSCs ($n=908$) from each donor. Points are coloured as in panel B. Line shows linear mixed-effect regression of mutation burden observed in colonies and shaded areas indicate the 95% confidence interval.

Rebuttal Figure R2.1 and revised Figure 3B: Haematopoietic stem and progenitor cell (HSPC) prevalence during murine ageing. The relative abundance of total HSPCs (left) and individual HSPC subpopulations (right) are compared. MPP^{LY} are lymphoid-biased progenitors, MPP^{GM} are myeloid-biased progenitors, based on current immunophenotypic definitions.

9. The quite comprehensive analysis for clonal expansions (or lack) by both the initial WGS approach and targeted sequencing is very important, and the attempts to understand why are quite convincing. The section on application of various stressors perhaps not particularly definitive. More mice with fewer stressors might have been better, making any broad conclusions about types of mutations and expansions with only 4-5 mice per stressor isn't very convincing. Please specify what age the mice used were (legend just says "aged"). This section seems a topic mostly for another paper in terms of trying to figure out what stressor might be most relevant to using the mouse as a model for CH in humans.

We appreciate the Referee's acknowledgement of the efforts undertaken to characterise clonal expansions in the mouse, as well as the narrative for why we observe differences to humans. We have now added an additional 540 whole genomes, increasing the cohort size by over 40%, and the results remain unchanged.

We do feel that the section on stressors is important to retain. Without it, we would never have known if the lack of clones observed were due to the ultra-clean environment of laboratory mice. We used a variety of stressors to mimic the different selection environments likely to be experienced over the course of human lifespan. The increase in clone sizes observed post-perturbation was convincing (positive dN/dS), however, we concede that with more animals, we could have had greater resolution into the genes under positive selection on a *per* perturbation level - we have not been able to do so with the small numbers involved.

We have added the age of the mice used as suggested by the Referee on page 20 as follows: At final sampling, aged mice were 30 months old for the NME experiments in panel A), and were 25 months old for the perturbation experiments in panels B), C), and D).

10. The authors state in the discussion that "...any role age associated hematopoietic oligoclonality plays in human aging is unlikely to be shared by the laboratory mouse". What does this imply for murine-based HSC research being extrapolated to humans? Are there alternative models? No mention made of data from NHPs showing stem cell numbers and cycling estimates close to humans (Shepherd et al Blood 2007; Koelle et al Blood 2017) and clonal expansions with NHP aging (Yu et al Blood 2018; Shin et al Blood 2023).

We hypothesise that if there is a biological "function" for CH in humans, given the small size of the mouse clones, that biological impact is unlikely to be shared with the mouse in the natural state. We were not trying to imply that murine HSC work is not relevant to humans. In fact, many findings about hematopoiesis, the general quiescence of HSCs, etc, have been substantiated across species. However, the dramatically different population landscape of haematopoiesis in the old mouse versus old human, should be taken into account when conducting future studies of CH or hematopoietic ageing using a mouse model.

We have added the following to page 16-17: The dramatically different population structure and clonal evolutionary pattern of haematopoiesis in the old mouse versus old human, together with the small size of clones (necessitating sensitive detection methods), are crucial factors to be considered when using murine models for future studies of natural CH or haematopoietic ageing. These differences may be related to lifespan, as longer-lived non-human primates are reported to more closely reflect the haematopoietic population dynamics and somatic evolution patterns of humans. (Shepherd et al Blood 2007; Koelle et al Blood 2017, Yu et al Blood 2018; Shin et al Blood 2023)

Minor:

11. In Extended Figure 1C, what is the VAF referring to? Composite of all mutations? A specific site. Overall the figure legends throughout leave out quite a lot to help someone understand what is being shown without referring to Supplemental text sections on methods, or inference based on one's own knowledge. Please fix throughout going through with someone not an author to see what they don't understand or can't figure out easily.

Thank you for this critique that our figure legends can be improved and made more informative to improve readability. We have taken this comment onboard and made considerable effort in the revised manuscript to improve explanations and readability for all figures, using the main text where possible, as well as legends.

For Extended Figure 1C, 'VAF' refers to the VAF of each passing-filter variant within the colony. The distributions shown are for all post-filtering mutations detected in the colony. We have endeavoured to clarify this description by updating the Figure Legend as follows: **C) Variant allele fraction (VAF) distribution of all variants within a colony that pass filtering, shown for a representative clonal colony that passed sample QC (left) and a non-clonal colony that passed sample QC (right). After variant filtration, the VAF distribution of a colony's variants is centred around 50% in clonal colonies, but less than 50% in non-clonal colonies.**

For the remainder of the figure legends, as the referee suggested, we iteratively sought input from two non-author scientists (with respective backgrounds in molecular virology and cancer metabolism) and have expanded several figure legends to include greater details. We have acknowledged their support in the acknowledgements section of the manuscript.

12. Figure 1 Legend the last sentence of the description of panel B I believe refers to panel C, please move to correct location.

Thank you for catching this mistake; this has now been corrected in the revised manuscript.

13. On the phylogenetic trees, continually refer to "grey" lines but they sure look black to me, I kept trying to enlarge panels to see if there were both grey and black sections and I was missing something. Just call them black please throughout all the various parts of text referring to the trees, and in legends.

Thank you for highlighting this. We now refer to all the dark grey lines on the phylogenies as 'black' in the text and figure legends.

14. Line 304 refers to wrong panel of Figure 4, F not E

Thank you, this has been corrected.

15. In Extended Figure 4, includes HSC and MPP together?

Yes, we used both HSC and MPP colonies for the signature extraction process (Extended Figure 4A) and subsequent per-colony and per-branch signature-based mutation measurements (Extended Figure 4B and 4D). We have clarified that we used HSC and MPP together in the Methods section describing the signature extraction process. Additionally, we state that we are using "all" colonies in the figure legend for Extended Figure 4.

16. Figure 4 panel arrangement is annoying and confusing, completely out of normal top to bottom and left to right order. Also the legend for clone size and mutation type shouldn't be hidden down by panel F, since it applies to all the panels, correct? Needs to be shown with panel A.

Thank you for noting this discrepancy, and we agree the legend for panel A should be closer to the panel itself. We have reordered the panels within Figure 4 such that the panels follow normal left-to-right top-down convention and continue to be referenced in order within the text. We anticipate the new location of the figure panel will better facilitate the reader's interpretation of these data. The revised Figure 4 panel arrangement is shown below.

Figure 4

Referee #3 (Remarks to the Author):

Kapadia et. al. carry out whole genome sequencing of more than a thousand single blood cells from three young mice and three very old mice in order to examine somatic evolution of murine blood. Their observations are multitudinous. They determine mutation burdens, noting that mice have few mutations in their blood near the end of life relative to humans, which is relevant to a larger discussion on whether somatic mutation rates scale inversely with lifespan among species. They construct a phylogenetic tree for each mouse. The phylogenetic structures are used to infer that stem and multipotent progenitor cells (HSCs and MPPs respectively) are established early in life, and that the HSC and MPP populations independently self-renew and grow throughout life. The phylogenetic trees also suggest limited selection among the cells, in contrast to human blood. To study selection in greater resolution, they carry out deep targeted sequencing of the blood, focusing on genes orthologous to those genes recurrently mutated in human blood, and they find some very small clonal expansions apparently driven by mutations in these genes, and that mutation frequencies among these genes are increased by exposing the mice to pathogens and chemotherapies. Overall, although only few mice are included in the analysis, I appreciate that the genetic data are at exceptionally high resolution and that multiple aspects of somatic blood evolution are studied in some detail.

We thank the Reviewer for their positive comments on the study.

My main concern is that the novelty of the findings is limited, and prior literature not cited adequately. Chin et. al. (Blood 2022) already demonstrated that there is a considerably greater mutation burden among single blood cells in humans compared to mice towards old age (specifically, Chin et al. report a 5-fold lower mutation in mice vs. humans, on a similar scale as the 10-fold difference reported here). Chin et. al. also deeply sequenced mouse blood finding limited clonal hematopoiesis, although the present work deserves credit for even deeper sequencing, which is important given the rarity of clonal expansions. As for perturbations' effects on mutation frequencies in the blood, there are a great number of studies documenting increased clonal expansion after perturbations to hematopoiesis in a variety of settings (e.g. Hormaechea-Agulla Cell Stem Cell 2021, Heyde et al. Cell 2021, Meisel et al. Nature 2018 and many others); the advantage, which is substantial, of the present study is that the mice are 'normal' rather than being transplant recipients, however the concept is already extremely well established. Then there is the point that HSCs and MPPs are established during embryogenesis and that they independently self-renew through life. This was already demonstrated unambiguously in the mouse by Patel et. al. in Nature 2022, whose experimental view of the blood in early life is arguably even more direct. This paper is cited but in a general way, without pointing out the vast overlap between that work and the findings presented as novel here.

In summary, several of the main findings of the present work seem to lack novelty, however I do enjoy the breadth and resolution of the data, which provide some of clearest views of blood somatic evolution in mice to date. I think that the manuscript has notable weaknesses in clearly acknowledging previous work. It would be important to better delineate what is already known and what is new. Additionally, there are some weaknesses of the modelling analyses that would benefit from clearer acknowledgement and explanation (see further comments below).

We are pleased to read that the Reviewer finds the study's description of somatic evolution in the mouse to be of high resolution and breadth, providing "some of the clearest views of somatic evolution in the mouse". We wish to fully address all the points raised.

1. We apologise for inappropriately citing Chin *et al.* To clarify, Chin *et al.* is a small but very interesting study, published as a letter in *Blood* in 2022, in which the authors sequenced 6 single cell derived colonies from mice (24 months old). To our knowledge, this is the first time genome-wide somatic mutation burden has been measured in the mouse. However, we would first like to highlight that there are some caveats to this study. (i) The mouse dataset in this study is small - either six single colonies from one 24 month old mouse, or one single colony from 6 different 24 month old mice (or a combination, it is unclear to us from reading the paper). (ii) the mutation burden is difficult to assess from Figure 1b, being somewhere ~150-200 mutations at 24 months, and the 5x reduction from humans is based on a single time-point comparison (iii) the data do not allow assessment of the rate of somatic mutation accumulation over the murine lifespan as only one age (or possibly one mouse) was sequenced. (iv) the sequencing of only 6 murine whole genomes does not allow for the assessment of the presence of global positive selection in mouse HSCs, or the presence of mouse-specific CH driven by different genes to that driving CH in humans. The statement made in the study that CH is only present at low levels in mouse relies on CH in mice being driven by the same genes as in humans. (v) Murine CH was only detectable following a transplant and not able to observe native CH clone, limiting the models generalizability for the study of CH as an experimental endpoint. (vi) Lastly, no sequencing files were made available to reproduce the data. Overall, the data are intriguing, and are an important contribution to the field, but relatively incomplete, and hence published as a *Letter*.

Secondly, the data we observe from the sequencing of a larger number of single-cell derived colonies from mice aged 1 day through to 30 months (n= 908 colonies from n= 23 mice), suggest that the mouse mutation rate is only 2-fold higher than humans, and not 8.5x higher as reported by Chin *et al.*, *Blood* 2022.

We had originally cited this study in the introduction on page 4 para 3 but we agree that the wording we used fails to highlight this study's novel findings and we sincerely apologise for this.

We have adjusted the Introduction (page 3, para 5) as follows: The most commonly used strain ... broadly recapitulates many phenotypic features of human ageing, with preliminary data suggesting a lower rate of CH (Chin *et al.*).

We have also cited the paper in the relevant part of the results: Clonal expansions were recurrently driven by mutations in *Dnmt3a* and *Tet2*, genes frequently mutated in human CH⁴², but also *Bcor* and *Bcorl1*, observed in humans following bone marrow immune insult⁴³. These data are consistent with a previous report identifying rare, expanded clones in mice²².

Similarly, we apologise for not sufficiently acknowledging the literature on how environmental perturbations influence expansion of HSCs, a topic we have written about extensively previously (Kapadia & Goodell 2024, King...Goodell 2020). Importantly, environmental perturbations that truly drive clone expansion in humans are largely correlative. In mice, observations of environmental exposures have all been made by transplanting mice with large proportions of mutant cells (on the order of 10%) in mixtures with WT cells, and then treating the animals with a bolus of exposure (e.g. chemotherapy or infection). These transplant models are an excellent strategy to study these kinds of effects, and we ourselves have used them extensively (Hsu, *Cell Stem Cell* 2018; Chen, *Cell Stem Cell* 2023). However, dynamics of stem cell expansion likely differ when the starting populations are small. Nevertheless, we have endeavoured to acknowledge this work, and have cited three recent reviews which cover a number of these models. We have added the following statement to our discussion (page 17): Native murine clones do expand upon systemic exposures and recapitulate patterns previously observed in

correlative humans studies [Coombs, Bolton] and in exposures administered following murine transplantation [Hsu, Hormaechea-Agulla, Heyde, Meisel] this extensive literature is reviewed in depth in King et al., Florez et al., and Kapadia et al.

We have also endeavored to better cite the work of Patel on the relationship and timing of MPP and HSC generation. We posit our data support the concept but are very distinct in nature - see our response to Referee 1's first comment. We also extend the findings substantially through the new studies on LSK progenitor cells and matched sampling of the peripheral blood (See **new Extended Data Figures 7, 8, 9** and associated response to Reviewer 2 #4).

In our revised manuscript, we more appropriately cite the work of Patel et al. and others that have characterised nuances in the development and persistence of HSCs and MPPs. Some examples are shown below:

This process relies on a hierarchy of progenitors that successively amplify cellular output towards fully differentiated blood cells. All are **believed to ultimately derive** from a pool of rare haematopoietic stem cells (HSCs), a pool maintained in a relatively protected state to support blood production throughout life, **with recent studies suggesting heterogeneity within these long-term self-renewing haematopoietic cells.** (page 3 para 1)

Classical models of the haematopoietic differentiation hierarchy propose that MPPs derive from HSCs. **In recent years, a more nuanced and dynamic picture has emerged, with the identification of additional self-renewing progenitor compartments.** (page 7 para 1)

Classical models of blood production depict HSCs at the very top of the haematopoietic differentiation hierarchy, beneath which all blood cell types emanate. **Recent studies suggest additional heterogeneity at the top of this haematopoietic hierarchy and nuanced self-renewing dynamics.** (page 15 para 3)

In terms of potential weaknesses raised for the modelling analyses, please see our responses to the specific comments raised below.

Specific comments:

- Line 42 and the section beginning line 68: It is posed that it is unknown whether the somatic mutation burden in the blood at the end of life is similar between species. This gives an overstated impression of novelty of the somatic mutation burden measurements. Here, Chin et. al.'s paper deserves a clearer acknowledgement. They have already measured that the somatic mutation burden of single HSCs is far larger in old mice compared to old humans. Their mice and humans are not quite so old as those of the present manuscript, which is in favor of the present manuscript, however Chin et. al.'s results already unambiguously point towards substantially differing end of life mutation burdens.

We have adjusted the Introduction (page 3, para 5) as follows: **The most commonly used strain ... broadly recapitulates many phenotypic features of human ageing, with preliminary data suggesting a lower rate of CH (Chin et al.).**

Given the above paper sequences 6 single-cell derived whole genomes from (presumably) one 24-month mouse, we strongly believe that the data, whilst the first description of the difference between mutation burdens in equivalently aged mouse/human, is arguably preliminary given the limited sample set.

- Line 69: “Comparison of HSCs from young and old mice revealed a constant rate of somatic mutation accumulation with age” is an overstatement. There are only two time points. A straight line can indeed be drawn between two points, but two points are not enough to claim that a straight line has been revealed.

We apologise for this overstatement and fully agree. To address this issue, we have now grown HSC colonies from mice at additional ages across the mouse lifespan: one day old pups (n= 3 animals, n=40 colonies), 12 months (n= 3 animals, n=32 colonies), 18 months (n= 5 animals, n=105 colonies), 24 months (n=5 animals, n=58 colonies), 30 months (n=1 animal, n=7 colonies). Haematopoietic colonies were grown in the same manner as for the original cohort (comprising 3 months and 30 months old), together with the same protocol for library preparation, sequencing platform and somatic mutation calling. Given that the original cohort comprised many colonies from individual mice, whereas, for measuring mutation burden at additional timepoints, we opted for fewer colonies (range 9-24) across 3-5 mice at additional ages, our mutation calling strategy was optimised for the larger variation in number of colonies per mouse with respect to filtering of mutations. To assist with any potential batch effects, we added an additional 30 month old mouse in this second round of sequencing.

The new rate of somatic mutation accumulation is shown below.

Rebuttal Figure 2.9 and revised Figure 1C: Burden of individual single base substitutions (SBS) observed in HSCs (n=908) from each donor. Points are coloured as in panel B. Line shows linear mixed-effect regression of mutation burden observed in colonies and shaded areas indicate the 95% confidence interval.

This confirms that the mutation rate is indeed linear, and unchanged from what was reported previously. As expected, the intercept of the linear model is 48.6 mutations (CI 46.0-51.2), reflecting the excess mutations acquired in early development. We were pleased to see that this intercept was in line with the number of mutations that were present at birth in one day old pups (median 36.48, range 23.9-65.96). Indeed, this phenomenon is also seen in humans (both a positive intercept and excess number of mutations in cord blood). The rate of mutation accrual is 44.4 mutations per year (CI 40.8 - 46.8).

We additionally provide more granular timepoint data about the proportions of HSC and MPP subsets, as well as the LSK progenitor population. We have revised our Figure 3B to reflect added data from these additional intermediate timepoints.

Rebuttal Figure R2.1 and revised Figure 3B: Haematopoietic stem and progenitor cell (HSPC) prevalence during murine ageing. The relative abundance of total HSPCs (left) and individual HSPC subpopulations (right) are compared. MPP^{LY} are lymphoid-biased progenitors, MPP^{GM} are myeloid-biased progenitors, based on current immunophenotypic definitions.

We have updated our revised manuscript to reflect these data from intermediate timepoints, updating the mutation rate number described on page 4 and pages 14-15.

• Line 98: “the mutation burden [of peripheral blood] was not different from that of haematopoietic colonies” could be phrased more carefully. Figure 1E suggests that there is a difference, albeit with limited statistical resolution. It shows that among the three peripheral blood measurements and the multiple hundred HSC colony measurements, the peripheral blood makes up the top three mutation burdens.

Thank you for noting this overstatement. We have corrected this paragraph to: **The mutation burden was not statistically different from that of haematopoietic colonies (Fig.1E).** (page 5 para 2)

By way of explanation, we expected the SBS substitution rate to be higher with Nanoseq for 3 reasons (i) the samples used for Nanoseq were whole blood and therefore, comprised both lymphoid and myeloid cells. In humans, the SBS mutation rate in lymphoid cells has been shown to be substantially higher than that in myeloid cells. (ii) whole blood Nanoseq would have included DNA from mature cells and in humans, these too have been shown to have a few more mutations than immature HSCs. (iii) whole genome mutation burden using Nanoseq is a derived figure that captures mutations across a section of the genome, and then scales it up to the whole murine reference genome size. On the other hand, whole genome sequencing of the colonies counts only those mutations called by the pipelines. Therefore, ‘blind spots’ or ‘masked regions’ in the genome would not contribute to mutation burden in colonies, but would be included in the scaling up of the mutation burden in Nanoseq. Thus the confidence interval (reflecting the degree of sampled genome) should be considered alongside the absolute mutation burden, particularly in the case of Nanoseq data that can vary considerably in the depth of sequencing, and thus, the confidence around any estimate of the extrapolated genome wide mutation burden.

We have now added the following to the paper (page 5 para 2): “We did note a non-significant trend towards higher mutation burden estimates from whole blood than HSC colonies - this is likely due to whole blood including lymphoid cells which have higher mutation burdens. Despite whole blood having a mixture of mature cell types and the different sequencing technologies used, these data confirm that somatic mutation rates in blood do not inversely scale with lifespan to the same degree as observed in colon.

• Line 152: “Overall, our data suggest that many long-term HSC and MPP lineages are established independently and in parallel during development, and that MPPs do not always arise from HSCs, contrary to classical haematopoiesis models.” It is important to note here that this summary point is not new. Several ideas in this section were already established by Patel et al. whose work deserves prominent acknowledgement.

Thank you for this point and it was not our intention to overlook prior work. As mentioned above, we have endeavored to more prominently acknowledge the work of Patel on the relationship and timing of MPP and HSC generation. We now cite Patel et al separately in the introduction, discussion, and within the results. Our data aligns with many of Patel et al.’s observations but also are distinct in nature. Additionally, we more extensively acknowledge other work that have observed variation in the development and lifelong persistence of HSCs and MPPs. These increased acknowledgements are listed below.

This process relies on a hierarchy of progenitors that successively amplify cellular output towards fully differentiated blood cells. All are believed to ultimately derive from a pool of rare haematopoietic stem cells (HSCs), a pool maintained in a relatively protected state to support blood production throughout life, with recent studies suggesting heterogeneity within these long-term self-renewing haematopoietic cells. (page 3 para 1)

Classical models of the haematopoietic differentiation hierarchy propose that MPPs derive from HSCs. In recent years, a more nuanced and dynamic picture has emerged, with the identification of additional self-renewing progenitor compartments. (page 7 para 1)

Classical models of blood production depict HSCs at the very top of the haematopoietic differentiation hierarchy, beneath which all blood cell types emanate. Recent studies suggest additional heterogeneity at the top of this haematopoietic hierarchy and more nuanced and dynamic haematopoietic self-renewing dynamics. (page 15 para 3)

• Line 156 to 186: To help readers understand the meaning of the inferences, I think it would be valuable to clearly explain that the transition rates governing the development of typical HSCs and MPPs are not equivalent to the transition rates governing the development of those specific HSCs and MPPs whose lineages are seen on the phylogenetic tree. This point is hinted at in the sentence “The apparent inconsistency in the results between young and old mice could perhaps be explained if the HSCs that produce the MPPs early in life are extinguished by old age.” However I think that most readers will need more than this sentence to understand the difference. Related to this point, can you say the extent to which the relative numbers of sampled MPPs and HSCs (which differ from the relative numbers of MPPs and HSCs in the pool) influence the inferred transition rates? As the analysis stands, it’s unclear to me how much the parameter estimates may be determined by sampling decisions rather than biology.

Thank you for this thoughtful point. To help the reader recognize our data is commenting on ancestral, typical HSCs and MPPs, and not just the sampled cells used to make our trees, we have clarified our statement on page 9 para 2 to the following:

This apparent inconsistency in the results between young and old mice could perhaps be explained if the HSCs that produce the MPPs early in life are extinguished by old age, and thus could not be sampled for inclusion in the phylogeny. Alternatively, the rate of HSCs that transition to MPPs may be greater earlier in life. Further work is required to explore this.

With respect to our HSC/MPP transitions, we appreciate that we have sampled roughly equal numbers of HSCs and MPPs for the hidden Markov tree modelling. However, qualitatively, the clustering of HSC tips and MPP tips on the phylogenetic trees will hold irrespective of the sampling regime. This banding of clades of HSCs and MPPs across all phylogenies in this study is, perhaps, the most important observation for divergence of these two populations early in life.

Nevertheless, we wished to check if our inferences for rejecting an “HSC-first” ontogeny would be affected by the sampling regime. For the old animals, we downsampled HSC and MPP numbers separately by a factor of 0.33, 0.5 and 0.66 and found that the HSC-first likelihood ratio test is still highly significant (Table R2.1).

Label	Cell Type totals across 3 animals		HSC Ratio	Total Cell Count	Log Likelihood		p_LRT_HSCFIRST
	HSC	MPP			hscfirst	full	
HSC(116)/MPP(322)	116	322	26.5%	438	-241.64374	-213.78404	8.36E-14
HSC(177)/MPP(322)	177	322	35.5%	499	-293.16328	-265.01198	6.21E-14
HSC(233)/MPP(322)	233	322	42.0%	555	-339.75516	-305.97841	2.05E-16
HSC(353)/MPP(322)	353	322	52.3%	675	-418.77899	-377.79256	1.38E-19
HSC(353)/MPP(212)	353	212	62.5%	565	-337.50523	-304.32629	3.76E-16
HSC(353)/MPP(161)	353	161	68.7%	514	-289.49436	-265.83296	6.02E-12
HSC(353)/MPP(106)	353	106	76.9%	459	-232.35209	-215.63024	7.34E-09

Table R2.1 HSC and MPP sample size adjustments and hidden Markov tree based ontogeny inference. The maximum likelihood of two nested models is considered “full” where 4 parameters are estimated representing the transition probabilities of EMB->HSC, EMB->MPP, HSC->MPP, MPP->HSC, and “hscfirst” where the transition probability of EMB->MPP is fixed at 0, and the other 3 parameters are estimated. For details see Supplementary Note 2.

Next, we wished to more formally address this. We have previously developed an HSC population simulator *Rsimpop* (used in Williams *et al*, Nature 2022), and we first updated this package to support the simulation of continuous differentiation between tissue compartments (embryonic cells, HSCs and MPPs). This allows the user to specify rates for migration or differentiation between any number of cellular compartments. Once the cell compartment specific population limits are reached, the death rates in the cell compartment balance any cell production. We then used this population simulator to simulate 3 phylogenetic trees (equal population size of HSC and MPP, 10x more HSCs than MPPs, and 10x more MPPs than HSCs).

We next checked if our hidden Markov tree model, which we used to explore if it was likely that HSCs and MPPs were being laid out independently, was sensitive to different background HSC/MPP population sizes, when HSCs and MPPs are sampled in roughly equal numbers. One example shown below (Figure R2.2) reassures that despite unbalanced population sizes, the hidden markov tree model can broadly capture the ground truth of transitions.

Figure R2.2 Simulation of unequal HSC and MPP population sizes and subsequent hidden Markov transition modelling Inferred Viterbi trees alongside the ground truth simulated trees (0:2 green is HSC and 0:3 purple is MPP). Informally we have a decent correspondence despite sampling bias.

Lastly, we considered if sampling decisions could influence our estimates of population size (N) or cell division rate. Both the *phylodyn* trajectories and approximate Bayesian computation (ABC) analyses rely on phylogeny topology (where we include branch lengths in our broad use of the term topology here) only, and thus are entirely 'blind' to if the underlying colonies are from HSCs or MPPs. The *phylodyn* analyses consider the HSC and MPP trajectories as entirely independent events. Given the similarity (or lack of drastic difference) in tree topology between HSC and MPPs, and that the *phylodyn* central estimates follow close trajectories, we can expect cell division and cell death rates to be similar between both populations. Additionally, the ABC analysis only uses the phylogeny topology as 'ground truth' but is totally agnostic to if specific branches are derived from HSCs or MPPs. Thus, these supporting models are driven by tree topology only and are 'blind' to sampling decisions or ratios between HSC or MPP. The selection of more colonies (either HSC or MPP) only serves to increase the resolution in topological structure of the phylogeny and the corresponding breadth of credibility intervals.

In summary, these additional analyses reassure us that the sampling approach used does not bias our main conclusion that HSC and MPP are essentially laid out independently early in life. In future work it would be beneficial to perform a large scale ABC analysis using the simulation approach that accounts for continuous differentiation between blood cell types.

• Line 172 to 176: The p values presented here are impressive-looking, however I am unsure how seriously to take them. The likelihood ratio test is used on six independent data points (only three

independent data points for the age-specific tests), but the likelihood ratio test's theoretical validity requires large sample sizes. I suggest either to note this potential caveat for the less statistically experienced readers or to leave aside the p values in order to focus on the Bayesian analysis.

The reviewer correctly points out that the likelihood ratio test is valid in the large sample size limit – specifically, twice the difference in the log-likelihood is asymptotically chi-square distributed according to Wilks' theorem. However, even for a single mouse it can be argued that each emerging clade, which we choose to identify by cutting at 25 mutations in molecular time, is a quasi-independent sample (especially prior to the saturation population sizes being reached). When combined over 3 mice there are 100s of these “independent” clades. Thus, in the terms of our simple model it makes sense to apply the LRT in the way that we have. Nonetheless, we concede that the p-value is for a statistical test for an idealised model that imperfectly models the biological system and so the impressive p-value should be interpreted with caution. Indeed the differing results we obtain for the old and young mice indicates that further work is required to convincingly assess the HSC-first hypothesis.

Additionally, in an effort to investigate the type 1 error rate, we simulated a collection of 100 groups of 3 simulated trees each with 200 tips. We then simulated an HSC-first discrete markov chain process down the tree. We find that the likelihood ratio test was quite conservative in this instance and reported only 1 simulation with a p-value < 0.05. Admittedly, this limited study does not explore the type 1 error rate at very low p-values but does indicate that the model p-values are not particularly permissive.

At the referee's suggestion, we have removed the p-values from the main text and focused on the underlying approach and conclusions (page 9).

• Line 248 to 251: Can you comment on the robustness of the estimates population size, and cell division and exit rates? In particular, how much do these estimates depend on your prior distributions? My sense is that these estimates could dramatically change with different priors. For example, what if the prior distribution on the population size were uniform between 10^2 and 10^7 rather than uniform between 10^2 and 10^5 as written? It's also worth noting that while inference of N/λ from phylogenies is standard, disentangling N and λ from phylogenies alone is notoriously problematic (see Louca and Pennell Nature 2020 for example).

We thank the reviewer for these important questions. Our response below will address these questions in sequence, as follows:

1. Address how the estimates of population size and birth/death rates may be affected by the prior distributions.
2. Explain why the generalised solution in Louca & Pennell 2020 does not apply to these data
3. Explain how the constraints of these data and our model make N and λ identifiable. We have added a new Supplementary Note 4 to the revised manuscript providing a mathematical explanation of how these data allow separation, or *identifiability*, and N and λ .

1) Prior distribution selection and robustness.

We agree that with ABC, the prior distributions can heavily influence the resultant posterior estimates. However, we consider our estimates to be robust, and the prior distributions included in our manuscript to be sound.

In this case of mouse HSC dynamics, the prior distribution can be constrained by some degree of ground truth. From flow cytometry experiments that have endeavoured to count every HSC in single mice, it is observed the mouse HSC compartment holds between 5,000 to 20,000 cells (Challen *et al* 2009). Thus, we placed an upper limit of 10^5 on the prior distribution, to reflect this 'biological plausibility'. By extension, selecting an upper limit of 10^7 would be biologically improbable. Mouse bone marrow contains on the order of 10^8 nucleated cells. Given the rarity of HSCs, it is not a biologically plausible scenario where HSCs were 10% or even 1% of total bone marrow.

That said, we also wished to assuage ourselves of the robustness of our estimates. We did run additional ABC simulations, notably with a broadened uniform prior distribution from 10^2 to 10^6 , with the expectation that our posterior estimates would converge on a similar value if the prior range of 10^2 - 10^5 was sound.

Indeed, with the extended prior range, we observe a population size estimate of $\sim 90,000$ HSCs, compared with the $\sim 70,000$ HSCs in the initial submission. These results assuage us that our population size estimate is not an artifact of prior distribution selection.

Figure R3.1. Comparison of joint probability distributions with differing prior distributions. A) For population size (N) prior distribution 10^2 - 10^5 , population size converges around 77,000 cells. B) With an extended prior distribution 10^2 - 10^6 , the estimated population size converges to similar magnitude, around 90,000 cells.

2) Comparison to generalized birth-death process described in Louca & Pennel

As the reviewer notes, Louca and Pennel importantly provide mathematical evidence that N and λ cannot be disentangled (ie are unidentifiable), however their proof is surrounding generalized birth-death models. Louca and Pennel 2020 is based on a version of the birth-death process in which the birth rate and death rate can vary continuously through time. There is also an additional parameter called the "sampling fraction", which is denoted by ρ .

We offer 2 concrete examples of the differences in our model and the generalized model described in Louca and Pennel.

(i) Louca and Pennel consider paleontological phylogenies, where they can have not known constraints on population size in the past. (For an example case, while we know how many species of songbird exist today, we have no way of knowing how many evolutionary ancestors of songbirds existed in phylogenetic history.) However, in the case of HSC dynamics, we know an organism grows from a single cell and we place constraints (prior ranges) on the number of HSCs that could have existed any time in the organismal past.

(ii) A generalized birth-death process allows birth and death rates to vary within an epoch of growth. Our model's birth and death rates do not vary; they are constrained.

In the species phylogeny setting of Louca and Pennell 2020, there is also a dependence of the sample size on the "sampling fraction" parameter and on the population size (at the time of sampling). This kind of sampling model is what Stadler 2009 ("On incomplete sampling under birth–death models and connections to the sampling-based coalescent") refers to as the "birth-death-sampling ρ process", in order to distinguish it from a second kind of sampling model (in which the sample size is specified essentially independently of the population size), referred to as the "birth-death-sampling m process" (m being the population size at the time of sampling - in the notation of that paper).

Our data is an example of a "birth-death-sampling m process" – our sample sizes are specified independent of the population size. The Stadler 2009 paper suggests that the Louca and Pennell 2020 sampling model is not directly relevant to our inference problem.

Finally, we note that even in the setting of the "birth-death-sampling ρ process", there are results which indicate that when the birth and death rates are piecewise-constant, these parameters can be identifiable. See Legried and Terhorst 2022.

3) Identifiability of N and λ within mouse HSC phylogenies.

To mathematically explain and justify the separation, or identifiability, of N and λ , we have added a new Supplementary Note 4 to the revised manuscript. A summary of this work is below, and so that the reviewer does not need to jump between documents, we include the entirety of Supplementary Note 4 at the end of this rebuttal letter document.

In the case of neutral deterministic growth models, we have exact formulas for the likelihood function, where the model parameter is a sequence of drift intensities (or equivalently, a sequence of effective population sizes). In the case of stochastic growth models which can be approximated by a neutral deterministic growth model, we can obtain approximate formulas for the likelihood function in which a sequence of effective population sizes is replaced by a parameter vector which includes birth rates and death rates as model parameters. We use approximate likelihood functions obtained in this way to address the issue of identifiability of parameters for various models. In particular, we are guided by the observation that whenever the population size is not too small, and the growth rate is not too close to zero, that a birth-death process with constant birth rate, and death rate, behaves much like a deterministic exponential growth model.

The observation from the *Phylodyn* trajectories that the log of the estimated effective population size increases linearly with time is difficult to explain except by a model of exponentially growing population (with constant growth rate). A birth death process with constant birth rate and death rate parameters, is a simple and (biologically) plausible model which behaves in the same way (to a close approximation) as the deterministic exponential growth model. The birth death process with constant rates is an exceptionally parsimonious explanation for the observed exponential trajectory. If we can accept this parsimonious explanation, then we can set aside the general

problem of making inferences about an arbitrary trajectory and restrict our attention to the very specific problem of making inferences about the parameters of the deterministic exponential growth model, or the parameters of the birth death process.

In general, if we were to plot the likelihood function over the space of parameter values, then the axes along which the likelihood function itself varies in value are the identifiable parameters, while those axes along which the likelihood function remains constant are non-identifiable parameters. In our Supplementary Note 4, parameter identifiability is defined more carefully, with some pointers to the literature. There we also discuss in more detail the implications of non-identifiability for parameter estimation in our current model. In particular, we discuss how additional constraints on the population trajectory can restore identifiability of the population size, and cell division rates.

In the case of a neutral deterministic growth model, the intensity of random drift is an identifiable parameter (at least at certain points along its trajectory), while the population size and the rates of cell division (and cell death) are non-identifiable. The drift intensity at time t refers to the ratio $\lambda(t)/N(t)$ of the rate of cell division (per cell per year) to the population size at time t . In Supplementary Note 4, the intensity of random drift is defined more precisely, for the case of a Moran-type population genetic model, with a population size that changes deterministically over time. Loosely speaking the population size and the rate of cell division (and cell death) are non-identifiable parameters because they do not appear separately in the likelihood function, but only in the particular combination represented by the trajectory for the drift intensity.

In the deterministic exponential growth model, the model parameters are the rate of cell division λ , the rate of cell death ν , and the size N of the ancestral population at the start of the epoch of exponential growth. From the formula for the likelihood function for this model (Equation 23 of Supplementary Note 4), it is apparent that the identifiable parameters are the population growth rate $(\lambda - \nu)$, and the ratio λ / N_A . In general, if the population size at the beginning of the epoch of exponential growth is known, then the rate of cell division λ , the rate of cell death ν , become identifiable parameters. In the special case where the epoch of exponential growth (at constant growth rate) extends all the way back to the founding individual (zygote cell), we know that the ancestral population was $N_A = 1$, at the moment of conception. So, in this case the rate of cell division λ , the rate of cell death ν , are identifiable parameters.

- Figure 3: presented estimates for lambda are 0.16, 0.17, and 0.14, but on line 769 it is stated that lambda has a uniform prior distribution between 0.01 and 0.15, indicating an error.

Thank you for catching this important mistype. Our initial submission Methods section erroneously stated the prior distribution for λ “ranged from 0.01 to 0.15 cell division per week.” We have corrected the Methods in revised manuscript to state the prior distribution for λ “ranged from 0.01 to 0.15 cell division per day.”

The input for *rsimpop* and ABC is in “per-day” units, so we state the cell division rate prior range in this unit when describing our Methods. In the remainder of the manuscript, including the Figures, we use “per-week” units as this time unit is convention for mouse studies. (It would be equally correct to state the prior ranged from 0.07 to 1.05 cell divisions per *week*, but these are not the values we use directly for ABC.)

- Line 268: It is stated that there is an “absence of selection on non-synonymous mutations (using dN/dS)” in mouse blood, which by contrast manifests “ubiquitously over time in humans”. Please can the authors clarify whether this mouse-human dN/dS comparison is merely a question of

sample size or, if not clarify, state that the comparison is not clear? Figure 3D shows that dN/dS for old mice hovers around 1.05 (although the actual number is not stated) and with a confidence interval that just covers one. On the other hand, the cited study is based on a much larger number of human HSC and it calculates a genome-wide estimate for dN/dS of 1.06 with a narrower confidence interval that does not cover one.

Thank you for this important critique. Our study initially included 1305 mouse colonies, and with the added colonies added during revision, now comprises 1845 whole genomes. With the added 540 additional genomes, and dN/dS did not meaningfully change with these additional colonies. Initial submission (n=1305): dN/dS = 1.038 (0.916 - 1.175). Revised manuscript (n=1845): dN/dS = 1.03 (0.919 - 1.143)

In contrast, in Mitchell *et al*, Nature 2022, we analysed 3579 colony whole genomes. However, the positive dN/dS ratio in that study was also detected in subsets of the data that were similar in number to our study. In the study's Extended Figure 9c (shown below) - when testing young (n=1512) colonies, or only old colonies (n=1461), dN/dS was significantly >1. While this was unsurprising in the older phylogenetic trees given the dramatic clonal expansions observed, we were surprised to also see this in young donors (save for cord blood), suggesting the rate of entry of driver mutations into the HSC pool is constant over the human lifespan.

Figure R3.2 (Extended Figure 9c (extract from Mitchell *et al*, Nature 2022)), Estimated number of driver mutations in the different datasets. The boxes show the estimate with whiskers showing the 95% CI. The numbers to the left give the numeric values for the estimates with 95%CI in brackets. 'n' is the number of cells included in each dataset.

It remains possible that with an additional ~1000 whole genomes from mouse, the confidence interval of our calculated dN/dS value may become smaller and thus become significant. We believe it is more likely to become significant if either the mouse lived for substantially longer and we were able to sample at such a later timepoint, or perhaps in the context of environments or perturbations that enhanced clonal selection to enable earlier detection. Indeed, in the mouse there is positive selection on genes that drive CH in humans, but the clones remain very small (1 in 1000 to 1 in 10,000 cells) and none of these mutations were detected on phylogenetic trees. Enough time has simply not passed for these clones to become of a significant size by the end of murine life. Overall, given the gross differences in topology between aged murine and human phylogenetic trees, and the lack of large clonal expansions, the consequences of any such positive selection are limited and likely not to be significant during the laboratory lifespan of mice.

- Line 307 to 335: The great literature on environmental exposure-altered clonal expansions in the blood (in both mouse and human) deserves more recognition.

As discussed above, environmental perturbations that truly drive clone expansion in humans are largely correlative. While we are unaware of many definitive studies that demonstrate causality, we acknowledge there is an enormous body of work in this space to which we can more appropriately direct the reader. In humans, smoking has been correlated with *ASXL1* clones, and chemoradiation has been associated with *TP53* and *PPM1D* clones (some examples: Coombs et al. 2017, Bolton et al. 2019). In mice, observations of environmental exposures have all been made by transplanting mice with large proportions of mutant cells (on the order of 10%) in mixtures with WT cells, and then treating the animals with a bolus of exposure (e.g. chemotherapy or infection). The dynamics of stem cell expansion likely differ when the starting populations are small as found in the natural setting. Nevertheless, we have endeavoured to acknowledge this work, and have cited three that recent reviews which cover a number of these models (King et al 2021, Florez 2022, Kapadia 2024), and some primary literature that provided the foundation for the hypotheses we tested in our native murine clone perturbation experiments (Hormaechea-Agulla Cell Stem Cell 2021; Heyde et al. Cell 2021, Meisel and Hinterleitner et al. 2018).

We have added the following statement to our discussion: Native murine clones do expand upon systemic exposures and recapitulate patterns previously observed in correlative humans studies [Coombs, Bolton] and in exposures administered following murine transplantation [Hsu, Hormaechea-Agulla, Heyde, Meisel] this extensive literature is reviewed in depth in King et al., Florez et al., and Kapadia et al.

- Line 349: This inference of selection uses the assumption that the stem cell population has constant size, however, Figure 3 claims a continuously growing stem cell pool which expands by several orders of magnitude throughout life. Can the inference be adapted to the setting of a changing population size?

The evolutionary framework for HSC dynamics assumes a constant selective advantage, as the Referee correctly points out. One could use approximate Bayesian computation to infer strength of selection in the context of growing population sizes. We undertook this in Williams *et al* Nature 2022, to estimate the strength of selection for very early-in-life *DNMT3A* mutation acquisitions, as these clones were proliferating during a background of ongoing population expansion of embryogenesis. This has the effect of reducing the strength of selection required for clonal expansion.

In this study, given that the stem cell pool may be growing at different rates early and later in life, estimates of fitness would be affected by the time in life when the mutation is acquired. We have no information on this as none of these driver mutations were captured in phylogenetic trees due to their small clone sizes. Furthermore, in an ideal scenario, one would estimate selection on a *per* driver mutation basis using the pattern of inter-coalescent intervals on the phylogenetic tree, or from duplex sequencing data where the same variant is observed across several different mice. However, as the number of clones from each specific variant is low, we have only been able to estimate the average selective advantage across all variants detected by duplex sequencing. Given the paucity of data, we do not believe it would be a fruitful effort to develop more complex models to take into account population growth. However, we have added a caveat to the discussion to acknowledge that background population growth may allow clones with weaker driver mutations to establish as follows: Indeed, in the short-lived mouse, variants with weaker fitness (<50%) might have insufficient time to enter exponential, deterministic growth within the population, given that clones are not established until $t_{years} > \frac{1}{s}$ (ref. ²), although any background

growth in population size could circumvent this, allowing for weaker variants to fix in the population. (page 15, para 2)

- Line 371: Can you clarify, is s the clone growth rate? Also, by establishment, do you mean fixation or reaching exponential, deterministic growth? The latter option makes sense in the context of your sentence, but the former appears to be the topic of the cited paper.

Apologies for the confusion. Here, “ s ” is the difference between the birth and death rates (i.e. $s = B-D$). In scenarios where there is no interference between clones and where stochastic effects are small due to the clone being large, then yes, the *growth* rate of the clone would also be “ s ”. Establishment refers to when approximate deterministic growth is attained which happens when the clone reaches a size $n = \lambda/s$ which takes a time of about $1/s$ years. Thus, it does not refer to fixation. However, the mathematics of the result justifying the time required for ‘fixation’ or how long a clone takes to get to establishment are effectively the same, so the cited reference is appropriate in our context.

Rebuttal Appendix. Here we reproduce Supplementary Note 4 that is now included in the manuscript text for the reviewer.

Supplementary Note 4: Inferring population sizes and cell division rates from cell sample phylogenies

Introduction

The aim of this exercise is to understand the apparent identifiability of the model parameters, seen in Bayesian inferences about the dynamics of HSC populations in mice (and other systems with similar population dynamics), when the phylogeny of a sample of descendent cells is the only available data. These Bayesian inferences were performed using ABC (approximate Bayesian computation) methods. Here the term approximate Bayesian computation refers to a class of Monte Carlo methods for generating samples from posterior distribution, which avoid computation of the likelihood function, by relying on simulation of the model. The more descriptive term *likelihood-free* Bayesian computation is also used. These methods include rejection sampling (pioneered by ³), and various regression methods (introduced by ⁴).

The ABC results reported here were obtained by using the *rsimpop* package⁵ to perform simulations of models of cell population dynamics, and then using the *abc* package⁶ to compute (approximate) marginal posterior densities for modal parameters. The *rsimpop* package allows us to specify a wide range of stochastic growth models based on an underlying birth-death process.

In the case of neutral deterministic growth models, we have exact formulas for the likelihood function, where the model parameter is a sequence of *effective population sizes* (or a sequence of *drift intensities*). For these models, efficient Monte Carlo methods⁷ are available for sampling from the exact posterior distribution of the model parameters. In the case of stochastic growth models which can be approximated by a neutral deterministic growth model, we can obtain an approximate formulas for the likelihood function in which sequence of effective population sizes is replaced by a parameter vector which includes birth rates and death rates as model parameters. We will use approximate likelihood functions obtained in this way to address the issue of identifiability of parameters for various models.

Likelihood functions for neutral models given phylogeny data

When we have genome sequences from a sample of single cells taken from an individual donor, we can construct a phylogeny for the sample, with the mutations assigned to branches. From this phylogeny, we can obtain an ultrametric tree, in which the relative lengths of the branches can be

estimated (taking account of the number of mutations assigned to each branch). We also know the age t_S of the donor at the time point at which the sample was taken. (Here age is measured from the moment of conception.)

From the phylogeny on a sample of n cells, together with the estimated absolute branch lengths, we can label the internal nodes (coalescent event) with integers $2, 3, \dots, n$, where n is the label on the most recent node (closest to the time of sample collection), and where 2 is the label on the earliest node on the phylogeny (the root node). We let S be the sequence of node heights $(S(n), S(n-1), \dots, S(2))$, where $S(r)$ is the height (time in days or years, measured backwards from sample collection) of internal node r . These node heights are determined by the branch lengths. The same information is contained in the sequence T of inter-coalescent interval durations $(T(n), T(n-1), \dots, T(2))$. The inter-coalescent interval duration $T(r)$ is the duration (in days or years) of the time interval during which exactly r lines of descent remain.

We begin by allowing the neutral model to take a very general form, which can be viewed as a generalisation of the neutral Moran model⁸. For now, we measure time t forward from conception ($t = 0$) when the population of cells contains a single founder cell ($N_0 = 1$), which is the zygote. This time t coincides with age (measured from conception). The sequence of distinct time points (ages) at which the population size changes, together with the age t_C at the time of sample collection, is recorded as $t = (t_1, t_2, \dots, t_C)$, where $0 < t_1 < t_2 < \dots < t_C$. So we have a sequence of $C-1$ population events which occur before sample collection. We assume that at each population event, these changes in population size occur instantaneously, so that we can define a population size N_k which persists throughout the time interval $[t_k, t_{k+1})$ from event k to the moment immediately preceding the next event.

At each of these events ($k = 1, 2, \dots, C-1$) at which the population size changes, we allow the number of *births* b_k (cell division) to be either 0 or 1, and the number of *deaths* d_k (cells which leave the stem cell population, either via cell deaths, or via cell differentiation events) to any integer value from 0 up to N_{k-1} (the size of the population when it enters event k).

Note that in a birth-death process it is more usual to assume that each event is either a birth event (where $b_k = 1$, and $d_k = 0$) or a death event (where $b_k = 0$, and $d_k = 1$). However, it turns out that while the analysis outlined below is greatly complicated if we allow b_k to exceed 1, when we relax the constraints on d_k we encounter very little additional difficulty. We have the following recursion for the population size

$$N_k = N_{k-1} + b_k - d_k, \quad (1)$$

for $k = 1, 2, \dots, C-1$, subject to the constraints that either $b_k = 0$ or $b_k = 1$, and $0 \leq d_k \leq N_{k-1}$. There is one more sequence which it is useful for us to define here. This is the sequence of *drift intensities*, $\xi = (\xi_1, \xi_2, \dots, \xi_{C-1})$, where

$$\xi_k = \left(\frac{N_k}{2}\right)^{-1} b_k = \frac{2b_k}{N_k(N_k-1)}, \quad (2)$$

for $k = 1, 2, \dots, C - 1$. Recall that if there was no birth (cell division) at event k , then $b_k = 0$, and therefore $\xi_k = 0$. Notice that here we are using the conventional notation $\binom{n}{k}$, for binomial coefficients. In particular we have

$$\binom{n}{2} = \frac{n(n-1)}{2}. \quad (3)$$

We can define the function

$$b(t) = \sum_{k=1}^{C-1} b_k \delta(t, t_k),$$

which represents the intensity of birth events. We can also define the function

$$\xi(t) = \sum_{k=1}^{C-1} \xi_k \delta(t, t_k), \quad (4)$$

which represents the intensity of random drift.

We can express the drift intensity function as

$$\xi(t) = \left(\frac{N(t)}{2}\right)^{-1} b(t) = \frac{2b(t)}{N(t)(N(t)-1)}, \quad (5)$$

which is in agreement with the earlier definition (Equation 4). The trajectory of the intensity of random drift, as specified by the drift intensity function $\xi(t)$ (Equations 4 and 5), takes us a step closer to our goal of deriving an expression for the likelihood function for the sample phylogeny data. However, in order to express the likelihood function in its most familiar and convenient form, we need to express the trajectory $\xi(t)$ (and the related trajectories $N(t)$, and so on) as functions of time s measured backwards from the time point at which the sample was collected ($s = 0$). The

relationship between the forward time t (age from conception) and the backwards time s , is given by

$$s = t_C - t,$$

and hence $t = t_C - s$.

So we can represent the backwards time trajectory for population size as the function

$$\tilde{N}(s) = N(t_C - s).$$

Similarly, we can represent the backwards time trajectories for other quantities of interest as follows

$$\tilde{b}(s) = b(t_C - s)$$

and

$$\tilde{\xi}(s) = \xi(t_C - s).$$

Now that we have this definition of the (reverse time) population size function $\tilde{N}(s)$, we can express the (reverse time) drift intensity function as

$$\tilde{\xi}(s) = \left(\frac{\tilde{N}(s)}{2}\right)^{-1} \tilde{b}(s) = \frac{2\tilde{b}(s)}{\tilde{N}(s)(\tilde{N}(s)-1)}, \quad (6)$$

which is simply the reverse time version of Equation 5.

Recall that we defined the sequence t of distinct (forward) times (ages) at which the population size changes, together with the age t_C at the time of sample collection, $t = (t_1, t_2, \dots, t_C)$, where $0 < t_1 < t_2 < \dots < t_C$. The same sequence of time points, representing population events, which we have labelled with forward times (ages) t_k , can also be labelled with reverse times $s_k = t_C - t_k$, for $k = 1, 2, \dots, C-1$. We now define the sequence s of distinct reverse times at which the population size changes, together with the time $s_0 (= t_C)$ at which conception occurred, $s = (s_0, s_1, s_2, \dots, s_{C-1})$, where $s_k = t_C - t_k$, for each event k . Therefore, we have $s_0 > s_1 > s_2 > \dots > s_{C-1} > 0$. The function $\tilde{\xi}(s)$ (and the function $\xi(t)$) is completely determined by the sequence pair (t, ξ) , and also by the (equivalent) sequence pair (s, ξ) .

When the phylogeny, with (estimated) absolute branch lengths, is the only available data, the likelihood function of the model parameter given the data, is (up to a constant factor) equal to the joint probability density

$$p_n(T(n), T(n-1), \dots, T(2); s, \xi) = \prod_{r=2}^n f_r(T(r)|S(r+1); s, \xi), \quad (7)$$

where

$$S(r) = T(n) + T(n-1) + \dots + T(r),$$

and where each factor

$$f_r(w|s; s, \xi) = \binom{r}{2} \tilde{\xi}(s+w) \cdot R_r(w|s; s, \xi), \quad (8)$$

is the (marginal) probability density of the waiting time to the next coalescent event, starting from time point s , when r lines of descent remain (each of which can be traced back from the sample). The function $\tilde{\xi}(s)$ is the drift intensity at time s (measured backwards from the time of sample collection). The function

$$R_r(w|s; s, \xi) = \exp \left[- \binom{r}{2} \int_{u=s}^{u=s+w} \tilde{\xi}(u) du \right], \quad (9)$$

gives the probability that the waiting time to the next coalescent event (starting from time point s , when r lines of descent remain) is exceeds w . We could describe $R_r(w|s; s, \xi)$ as the *reliability* function (or survival function), and interpret $T(r)$ as a kind of *failure time* (at which one line of descent fails to persist).

Strictly speaking, Equations 8 and 9 represent an approximation which is valid whenever the entire sample phylogeny lies within a time interval throughout which the intensity of random drift $\tilde{\xi}(s)$ remains small (the effective population size remains large). See refs. ^{9,10} for derivation of the properties of the (reverse-time) genealogical process.

We want to draw attention to a feature of the likelihood function represented by Equations 7, 8, and 9. From the likelihood function (Equations 7 and 8) it is evident that, while segments (spanning certain time intervals) of the trajectory for the drift intensity (represented variously as a sequence pair s, ξ , or as a function of time), may constitute an identifiable parameter (when we have a phylogeny on a large enough sample), the trajectory for the population size, and the trajectory for

the intensity of birth events, in the absence of additional constraints, are non-identifiable parameters. This is because the population sizes and the counts of birth events do not appear separately in the likelihood function, but only in the particular combination represented by the trajectory for the drift intensity.

In Section 3 below, parameter identifiability is defined more carefully, with some pointers to the literature. We also discuss in more detail the implications of non-identifiability for parameter estimation in our current model. In particular we will discuss how additional constraints on the population trajectory can restore identifiability of the population size, and the intensity of birth events.

Parameter estimation and identifiability

We usually make some further assumptions about the possible trajectories which the population is allowed to follow through time. In the case of a deterministic growth model, we assume that the sequence pair (s, ξ) of event times and drift intensities belongs to a family of trajectories, in which the individual trajectory is completely determined by a parameter vector ϕ . (Typically this parameter vector is of low dimension.) We say that the family of trajectories is parametrised by ϕ . Here we have in mind models of deterministic exponential growth, where the model parameters include rates of cell division and rates of cell death.

In the case of a stochastic growth model, we assume that the sequence pair (s, ξ) is drawn from a distribution which belongs to some family of distributions. Within this family of distributions, the specific distribution is completely determined by a parameter vector ϕ . We say that the family of distributions is parametrised by ϕ . Here we have in mind models based on a birth death process, where the model parameters again include rates of cell division and rates of cell death.

In order to emphasise the dependence on the parameter vector ϕ , it is convenient to use the notation

$$\begin{aligned} L(\phi|T) &= p_n(T(n), T(n-1), \dots, T(2); \phi) \\ &= \prod_{r=2}^n f_r(T(r)|S(r+1); \phi), \end{aligned} \tag{10}$$

for the likelihood function specified by Equations 7 and 8.

We say that a parameter vector ϕ is *non-identifiable* whenever there is a mapping ϑ (to a vector of lower dimension) for which the likelihood function $L(\phi|T)$ depends on the parameter vector ϕ only through $\theta = \vartheta(\phi)$. In other words $\vartheta(\phi_1) = \vartheta(\phi_2)$ implies that $L(\phi_1|T) = L(\phi_2|T)$. If there is no such mapping ϑ , then we say that the parameter vector ϕ is *identifiable*. When the parameter vector ϕ is *identifiable*, we may also refer to the components of this vector as *identifiable* parameters. See ref. ¹¹ (*non-identifiability* is introduced in Section 3.15, on page 70, and discussed further on pages 72 and 74).

If such a mapping ϑ (to a vector of lower dimension) exists (so that ϕ is non-identifiable), then this means (loosely speaking) that from the fixed data T , we can not learn anything about the unobserved parameter vector ϕ , beyond what we can learn about the (lower dimensional) parameter vector θ . We can state this more precisely. First, we can always (leaving aside technical issues and pathological cases) express the prior density $\pi(\phi)$ for the parameter vector ϕ , in the form

$$\pi(\phi) = \pi(\phi|\theta)\pi(\theta). \quad (11)$$

If ϕ is non-identifiable, and $\theta = \vartheta(\phi)$ is identifiable, then the posterior density $\pi(\phi|T)$ of the parameter vector ϕ is of the form

$$\pi(\phi|T) = \pi(\phi|\theta)\pi(\theta|T). \quad (12)$$

As a consequence, we also have

$$\pi(\phi|T, \theta) = \pi(\phi|\theta). \quad (13)$$

This means that if we knew the (lower dimensional) parameter vector θ , then the observed data T would tell us nothing more about the (higher dimensional) parameter vector ϕ .

First we consider a family of models where the population trajectory includes prolonged epochs during which birth events and death events occur equally often, so that the population size remains stable. Then we consider neutral models where the trajectory includes epochs of (deterministic) exponential population growth (Section 5). Finally, we consider birth-death processes, without an upper boundary (Section 6), and with an upper boundary (Section 7) on

the population size, and how these stochastic growth models can be approximated by deterministic growth models.

Epochs of stable effective population size

First we consider a family of models where the population trajectory includes prolonged epochs during which the population size remains stable. Suppose that across the time interval $[a, b]$, the population size remains constant at N_A . In order to maintain a constant population size, the birth rate β_A must be balanced by an equal death rate.

The observed inter-coalescent interval durations $T(r)$, which fall within the time interval $[a, b]$, contribute factors to the likelihood function which are of the form

$$f_r(T(r)|S(r+1); \phi) = \left(\frac{r}{2}\right)^{\frac{2\beta_A}{N_A}} \cdot \exp\left[-\left(\frac{r}{2}\right)^{\frac{2\beta_A}{N_A}} T(r)\right], \quad (14)$$

where $\phi = (N_A, \beta_A)$ is the parameter vector of the model.

From the expression on the right hand side of Equation 14, it appears that the only identifiable parameter is the ratio β_A/N_A .

Epochs of exponential population growth

Now we turn to neutral models where the trajectory includes epochs of exponential population growth. Suppose that the (forward time) estimated trajectory $\hat{\xi}(t)$ of the drift intensity appears to fit an exponential growth path across the time interval $[t_A, t_C]$, where t_C is the time (age) at which the sample of n genome-sequenced cells was collected. The estimated trajectory $\hat{\xi}(t)$ at time t can be interpreted as a kind of average drift intensity over some interval centred on the time point t . The (forward time) estimated trajectory is

$$\hat{\xi}(t) = \hat{k} \cdot \exp[\hat{\rho}(t - t_A)], \quad (15)$$

which is based on point estimates $\hat{\rho}$ (for the growth rate) and \hat{k} (for the initial drift intensity). Notice that when $\hat{\rho}$ is positive, the drift intensity declines exponentially, with increasing age t .

If we measure time backwards from sample collection, then the (reverse time) estimated trajectory $\hat{\xi}^{\text{rev}}(s)$ of the drift intensity appears to fit an exponential growth path across the time interval $[0, s_A]$. The (reverse time) estimated trajectory is

$$\hat{\xi}(s) = \hat{k} \cdot \exp[\hat{\rho}(s_A - s)], \quad (16)$$

where $s_A = t_C - t_A$ is the time measured backwards from sample collection to the time point at which the epoch of exponential growth began. Notice that when $\hat{\rho}$ is positive, the drift intensity increases exponentially, with increasing time s .

There is this one very simple model of population growth, in which births occur at a constant rate λ , and deaths occur at a constant rate ν , which results in an exponential trajectory. This is an exceptionally parsimonious explanation for the observed exponential trajectory. If we can accept this parsimonious explanation, then we can set aside the general problem of making inferences about an arbitrary trajectory $\xi(t)$ for the intensity of random drift (the reciprocal of the effective population size), and restrict our attention to the very specific problem of making inferences about the parameters of the deterministic exponential growth model, or the parameters of the birth death process.

Having observed an (approximately) exponential trajectory for the drift intensity (and its reciprocal, the effective population size), from age t_A , up to the point of sample collection (at age t_C), we have arrived at a parsimonious explanation which we now examine in more detail. The population size has been growing at a constant growth rate ρ , while the birth rate has remained constant at a value λ , and the death rate has remained constant at a value ν , which yields the constant growth rate $\rho = \lambda - \nu$. Now we can express the trajectory for the population size $N(t)$, forward in time across the epoch of exponential growth (from age t_A to age t_C) as

$$N(t) = N_A \exp[\rho(t - t_A)], \quad (17)$$

while the forward time trajectory for the drift intensity is

$$\xi(t) = \frac{2\lambda}{N_A} \cdot \exp[-\rho(t - t_A)], \quad (18)$$

where N_A is the size of the ancestral population at age t_A (when the epoch of exponential growth begins).

We now return to time measured backwards from sample collection. The reverse time trajectory for the population size is

$$\tilde{N}(s) = N_A \exp[\rho(s_A - s)], \quad (19)$$

where $s_A = t_C - t_A$ is the time measured backwards from sample collection to the time point at which the epoch of exponential growth began. The reverse time trajectory for the drift intensity is

$$\tilde{\xi}(s) = \frac{2\lambda}{N_A} \cdot \exp[-\rho(s_A - s)]. \quad (20)$$

The (marginal) probability density $f_r(w|s; \phi)$ of the waiting time to the next coalescent event (starting from time point s , when r lines of descent remain), is in this case

$$f_r(w|s; \phi) = \left(\frac{r}{2}\right) \frac{2\lambda}{N_A} \cdot \exp[\rho(w + s - s_A)] \cdot R_r(w|s; \phi), \quad (21)$$

where $\phi = (\lambda, \nu, N_A)$ is the parameter vector of this model, and where

$$R_r(w|s; \phi) = \exp\left[-\left(\frac{r}{2}\right) \frac{2\lambda}{N_A} \cdot \frac{1}{\rho} \exp[\rho(s - s_A)](e^{\rho w} - 1)\right], \quad (22)$$

is the reliability function.

The observed inter-coalescent interval durations $T(r)$, which fall within the time interval $[0, s_A]$ (the epoch of exponential growth), contribute factors to the likelihood function which are of the form

$$\begin{aligned} & f_r(T(r)|S(r+1); \phi) \\ &= \left(\frac{r}{2}\right) \frac{2\lambda}{N_A} \cdot e^{-\rho(U(r)-T(r))} \cdot \exp\left[-\left(\frac{r}{2}\right) \frac{2\lambda}{N_A} \cdot \frac{1}{\rho} e^{-\rho U(r)}(e^{\rho T(r)} - 1)\right], \end{aligned} \quad (23)$$

where $U(r) = s_A - S(r+1)$.

The parameter vector of this model is $\phi = (\lambda, \nu, N_A)$, where λ is the birth rate, ν is the death rate, and N_A is the size of the ancestral population at the start of the epoch of exponential growth. (This occurs at age t_A , which precedes sample collection by time interval of duration $s_0 = t_C - t_A$.) From the formula for this factor of the likelihood function, it appears that the parameter vector ϕ is *non-identifiable*, while the parameter vector $\theta = (N_A/\lambda, \rho)$ is *identifiable*. The components of the parameter vector θ are the ratio N_A/λ , and the difference $\rho = \lambda - \nu$ (the population growth rate).

In the special case where the epoch of exponential growth (at constant growth rate ρ) extends all the way back to the founding individual (zygote cell), we know $N_A = 1$, and we know that (reverse) time $s_A = s_C$ (age $t_A = 0$) corresponds to the moment of conception. In this special case, the unobserved parameters λ and ν , are identifiable. More generally, if the population size at the

beginning of the epoch of exponential growth N_A is known with certainty, then the parameter vector $\theta = (\lambda, \nu)$ is *identifiable*.

In the case of a sample of single cell genome sequences obtained from blood-derived colonies, from a mouse (or any species with similar HSC dynamics), the parameter N_A is the size of the ancestral population of HSCs at age t_A (when the epoch of exponential growth begins); or if the time t_A is even earlier, then N_A is the size of the population of embryonic cells existing at this time which are ancestral to the HSCs. Unfortunately we do not have direct observations of the ancestral HSC population size N_A (at the age t_A when the epoch of exponential growth begins).

However, we can place some bounds on the value of N_A . First of all there is an upper bound M_A , on N_A , which can be obtained from embryological observations. We know the approximate number of cells in the embryo at age t_A . If some differentiation has already occurred, we may be able to exclude some cell types as HSC ancestors, and thus perhaps obtain an upper bound M_A which is somewhat lower than the average total number of cells in a mouse embryo at age t_A . Secondly, we have a lower bound on N_A , which we can obtain directly from the phylogeny. This is the number of lines of descent n_A present on the tree at time t_A .

The linear birth-death process

A linear birth-death process is a simple stochastic growth model in which birth events and death events occur at constant rates (birth rate λ and death rate ν) per individual (cell) per unit of time (day or year). Therefore the total rate of birth (respectively death) events in the population at each time point is proportional to the total number of individuals in the population at that time point (hence a *linear* birth-death process. The total size $N(t)$ of the population at each time point is determined by the (stochastic) sequence of events (births and deaths) up to that time point. For the properties of the linear birth-death process, see ref. ¹², and ref ¹³, pages 174–177.

Whenever the population size is not too small, and the growth rate is not too close to zero, the linear birth-death process behaves much like deterministic exponential growth. The trajectory for the population size $N(t)$ is well approximated by Equation 17, with growth rate $\rho = \lambda - \nu$, provided that the birth rate λ exceeds the death rate ν , so that ρ is positive.

In the case of an epoch of stochastic growth (under a linear birth-death process) it is important to bear in mind that the formula for the factors of the likelihood function in Equation 21, is an approximation, which can break-down. A conclusive argument about the identifiability of the

model parameters should be based on an exact formula for the likelihood function for the linear birth-death process, when the phylogeny is the only available data.

The birth-death process with an upper boundary on population size

If a mouse lives long enough, we would expect that the propensity of the mouse HSC population to grow exponentially will eventually be checked by the physical constraints on the space available to accommodate the HSC cells within the bone marrow.

In the case of a model where the population undergoes deterministic exponential growth until an upper boundary N_B on population size is reached, the phylogeny may contain additional information about the time T_B at which the population first hits the upper boundary N_B . Such information can be present only if the sample of cells has been collected from the population at a time point after the time T_B . In this case, the hitting time parameter T_B occurs in the likelihood function.

In the case of a model where the population undergoes deterministic exponential growth until an upper boundary N_B on population size is reached. The hitting time T_B is determined by model parameters (N_A/N_B and $\rho = \lambda - \nu$). Using Equation 17, we can obtain

$$\frac{N_B}{N_A} = \exp[\rho(T_B - t_A)], \quad (24)$$

and therefore

$$T_B = t_A + \frac{1}{\rho} \ln \left(\frac{N_B}{N_A} \right). \quad (25)$$

When the population reaches the upper boundary on population size, the marginal birth rate and the marginal death rate must be equal ($\delta_B = \beta_B$). The parameter vector of the model is now $\phi = (\lambda, \nu, N_A, N_B, \beta_B)$.

As usual we inspect the formula for the likelihood function in order to discover which parameters may be identifiable, and which are clearly non-identifiable. The factors of the likelihood function representing the epoch of exponential growth are of the form given in Equation 21, in which the parameter combinations λ/N_A and ρ appear. The factors of the likelihood function representing the epoch of stable population size are of the form given in Equation 14, in which the parameter combination β_B/N_B appears. We have also seen from Equation 24 that the ratio N_B/N_A is

determined by the parameter ρ and the hitting time T_B . The hitting time T_B is a change point, which also appears in the likelihood function. Therefore, from the formulas for the factors of the likelihood function, it appears that the parameter vector $\theta = (\rho, \lambda/N_A, \beta_B/N_B, N_B/N_A)$ is identifiable. Notice also that by combining the last three components of θ , we obtain

$$\frac{N_B}{N_A} \cdot \frac{\xi_B}{\xi_A} = \frac{N_B}{N_A} \cdot \frac{\beta_B}{N_B} \cdot \frac{N_A}{\lambda} = \frac{\beta_B}{\lambda}.$$

So the ratio β_B/λ is also identifiable.

In the special case where N_A is known for certain, the parameter vector $\theta = (\lambda, \nu, N_B, \beta_B)$ is identifiable. As already discussed in Section 5, when the epoch of exponential growth (at constant growth rate ρ) extends all the way back to the founding individual (zygote cell), we know $N_A = 1$. So, in this case, the parameters λ , ν , N_B , and β_B , are all identifiable, and amenable to estimation from the phylogeny of a sample.

References

1. Chen, J., Astle, C. M. & Harrison, D. E. Genetic regulation of primitive hematopoietic stem cell senescence. *Exp. Hematol.* **28**, 442–450 (2000).
2. Kimura, M. & Ohta, T. The Average Number of Generations until Fixation of a Mutant Gene in a Finite Population. *Genetics* **61**, 763–771 (1969).
3. Pritchard, J. K., Seielstad, M. T., Perez-Lezaun, A. & Feldman, M. W. Population growth of human Y chromosomes: a study of Y chromosome microsatellites. *Mol. Biol. Evol.* **16**, 1791–1798 (1999).
4. Beaumont, M. A., Zhang, W. & Balding, D. J. Approximate Bayesian computation in population genetics. *Genetics* **162**, 2025–2035 (2002).
5. Williams, N. *et al.* Life histories of myeloproliferative neoplasms inferred from phylogenies. *Nature* **602**, 162–168 (2022).
6. Csilléry, K., François, O. & Blum, M. G. abc: an R package for approximate Bayesian computation (ABC). *Methods Ecol. Evol.* **3**, 475–479 (2012).

7. Lan, S., Palacios, J. A., Karcher, M., Minin, V. N. & Shahbaba, B. An efficient Bayesian inference framework for coalescent-based nonparametric phylodynamics. *Bioinformatics* **31**, 3282–3289 (2015).
8. Moran, P. A. P. Random processes in genetics. *Proc. Camb. Philos. Soc.* **54**, 60–71 (1958).
9. Kingman, J. F. C. On the Genealogy of Large Populations. *J Appl Probab* **19A**, 27–43 (1982).
10. Griffiths, R. C. & Tavaré, S. Sampling theory for neutral alleles in a varying environment. *Philos. Trans. R. Soc. Lond. Ser. B* **344**, 403–410 (1994).
11. O’Hagan, A. & Forster, J. *Bayesian Inference*. vol. 2B (Arnold, London, UK, 2004).
12. Kendall, D. G. Stochastic Processes and Population Growth. *J. R. Stat. Soc. Ser. B Methodol.* **11**, 230–282 (1949).
13. Moran, P. A. P. *An Introduction to Probability Theory*. (Oxford University Press, Oxford, UK, 1968).

30th November 2024

We are grateful for the overall enthusiasm of the reviewers. We have addressed all of the comments as detailed below. Referee comments are shown in black text. Our responses are in blue text with any changes made to the manuscript highlighted in red text. Text included from the manuscript is highlighted, with amended text in red.

Referee #1 (Remarks to the Author):

I appreciate the authors thoughtful response to the technical concerns raised in the original review.

We thank the Referee for their positive comments and address their remaining queries below.

I have several additional very minor queries:

1. Please include a short justification for why only female mice were used for the phylogeny inference experiment and clarify in the methods whether these mice were nulliparous or multiparous.

We chose to study female mice because it was easier to obtain more aged females, as fewer males tend to have survive to 30 months during the period of ageing undertaken. Given the older aged mice sampled were female, we then chose to study younger females for internal consistency. We confirm that all mice were nulliparous.

We have added the following to the Methods (page 24, para 1): "Female mice were chosen for internal consistency and due to their slightly longer lifespan. All mice were nulliparous".

2. Do the authors feel the gene distribution of driver mutation by strain across C57BL/6 ; B6FVBF1/J; HET3 (Fig 4A,C,D) represents a different propensity to driver mutations across different strains or is random chance? This is a potentially interesting observation as it would suggest that mouse genetic background may play a role in shaping somatic evolution.

We thank the Referee for this interesting point. It is plausible that mouse genetic background can affect somatic evolution patterns, especially considering literature on potential strain-specific differences in HSC cell cycling rates (Chen et al, Experimental Hematology 2000).

However, we are apprehensive to draw conclusions about differences in gene distribution between the strains in this study. The three strains used were aged at different academic institutions. While all the animal facilities are outstanding and AAALAC-approved, we envision that their environmental differences, compounded over at least 24 months of aging, make these cohorts not directly comparable. (Animals facilities were Baylor College of Medicine, University of Minnesota, Jackson Labs, and the National Institute of Aging.)

This caveat aside, we did directly compare the driver gene distributions for strains of similar ages (using a more robust non-parametric test, as described below in response to comment 3). For C57/Bl6 24-month mice versus HET3 24-month mice, there is enrichment for *Asx1* clones in HET3 ($p=0.016$). In 37-month C57/BL6 versus 30-37month B6FVBF1/J mice, there is a statistical difference in *Bcor1* clones ($p=0.011$). However, these statistical differences could reflect either differences between strains or differences in their respective environments.

In a future study, it would be interesting to study CH across different strains in the context of perturbations, as the latter will enhance the detection of CH, and this may highlight how any potential strain-specific differences in the stem cell compartment may influence the consequences on population growth in response to positive selection in the bone marrow. We include the following in the Discussion (page 12 para 3), “Alternatively, mouse strains with higher HSC turnover⁵⁰ or exposed to more sustained systemic insults may better mimic human patterns of CH.”

3. For Figure 5, I have concerns about how the gene level enrichment is calculated. Per the methods "Gene-level enrichment was measured using a Fisher's exact test on the number and mutant and wildtype reads, normalized for coverage differences between samples." I think it would be more standard (and more robust) to consider individual mice as independent replicates and compare cases vs controls with a given gene mutated as the input to the fisher exact test rather than considering each sequencing read as an independent experiment.

For example, for figure 5B - the figure appears to indicate that there is a significant driver enrichment (two asterixis) for CUX1 - when there are 3/9 uninfected mice with mutations in this gene and 3/10 infected mice with mutations in this gene. A fisher exact test comparing 3/9 to 3/10 would yield a p=1. Similarly in 5C, there appear to be 3/7 mice with TET2 in the vehicle group and 6/8 mice with TET2 in the cisplatin group - fisher test comparing the number of mice would yield a p=0.31 for this comparison. For Fig 5D - if the CUX1 is significant for driver enrichment with 5FU treatment how is Bcor not significant? It is a bit hard to tell with the over plotting but it looks like a similar number of mice in the 5FU group have this mutation and there are no mice in the vehicle group with this mutation.

If the authors wish to account for the VAF size, there are also non-parametric tests that could be used to compare two groups of mice where the maximum VAF of a mutation in a given gene could be used and a mouse that does not have any mutation in a given driver gene could be included as 0.

Thank you for raising this important point. We had indeed aggregated mutant and wildtype reads across different mice within the control or treatment groups. To normalise for differences in sequencing depth between samples, all samples were normalised by VAF to be 30,000X depth. Whilst this approach did not account for any inter-individual differences or numbers of animals in each group, it did take into account clone size which we felt was important to consider when assessing the utility of the model for CH. It also appropriately recognises the issue of zero-clones in some mice where clones may not actually be absent, but simply be below our duplex detection limit. We fully accept the Referee's valid point here. We have now utilised a non-parametric test to account for differences in VAF between the control and experimental groups, using the Mann-Whitney U test. This can be applied to zero-clone settings (**Figure R1.1**). We have opted for this rather than a parametric approach, as it avoids the distributional assumptions associated with, for example, utilisation of a GLMM Poisson regression method.

Figure R1.1. Gene-level enrichment assessment in perturbation experiments. The results of the Fischer's exact test, which was included in the initial submission, is shown per gene. Results using a non-parametric Mann-Whitney U test, are shown adjacent per gene.

We have updated Figure 5 “Driver enrichment” in 5A, B, C and D to reflect the Mann-Whitney U test results presented above. The updated Figure 5 is reproduced on the following page.

We have adjusted the methods (page 36) to reflect this change as follows: Differences in clone burden between control and treated cohorts was quantified using a Mann-Whitney test on cumulative VAFs per sample. Gene-level enrichment was measured using a Mann-Whitney test on the maximum VAF for a given gene per sample, normalised for coverage differences between samples.

Lastly, regarding Figure 5D and the significance of *Cux1* and *Bcor*, in our submitted manuscripts, we noticed there was an accidental shifting of the 3 columns of this figure panel. This resulted in the dots and asterisks being erroneously offset by a row. This error has been fixed in the revised manuscript (both *Cux1* and *Bcor* do indeed have significance) and we thank the reviewer for noting this. The updated Figure 5 is reproduced on the following page.

4. For Fig 5 - please clarify in the methods or the figure legend that there is no correction for multiple hypothesis testing and also what the asterixis means eg (*) vs (**).

We apologise for these omissions. We can confirm that there was no correction for multiple hypothesis testing. * is p value <0.05 and ** is p value <0.01. We have amended the legend as follows (page 19):

Figure 5. Haematopoietic perturbation modulates selection landscapes. Clonal haematopoiesis prevalence in aged mice following **A)** normalised microbial experience (NME), **B)** *M. avium* infection, **C)** cisplatin treatment, and **D)** 5-FU myeloablation. At final sampling, aged mice were 30 months old for the NME experiments in panel A), and were 25 months old for the perturbation experiments in panels B), C), and D). Enrichment of clonal prevalence and dN/dS ratios departing from parity following treatment are shown for each gene. Survival curves and experimental endpoint blood counts are displayed for B) and C), using log-rank and two-sided *t* tests, respectively. * denotes p-value <0.05 and ** denotes p-value <0.01, with no correction for multiple hypothesis testing. Treatment schedules are as displayed or described in Methods.

Figure 5

5. For Figure 5B/ 5C /5D - there appear to be notable differences between the different control cohorts by gene - for example for TET2 there are 0/9 "uninfected" mice in 5B compared to 3/7 "vehicle" mice in 5C. For DNMT3A there are 0 mice in 5B Uninfected / 2 mice in 5C vehicle / and 3 mice in 5D "vehicle. Do the authors have thoughts about a source of this variation? If the vehicle mice and uninfected mice were compared using the gene level enrichment method described in the current draft(using a Fisher's exact test on the number and mutant and wildtype reads, normalized for coverage differences between samples), I worry that there could be gene level enrichment differences between these control groups which would not seem biologically plausible if they are the same age.

Thank you for this important observation. We have looked into this peculiarity. We wonder if some differences are because the control mice in 5B (uninfected with *M. avium*) were housed in a different facility than the control mice in 5C/D (vehicle in cisplatin and 5-FU experiments). *M. avium* experiments are conducted in a particular 'biohazard' animal suite, given the infectious nature of the pathogen. This biohazard animal suite also houses animals from various investigators utilising a range of murine pathogens, including influenza A virus, Group A strep bacteria, and the *Trichuris* (whipworm) and *Trypanosoma* (Chagas disease) parasite. Hence, this animal room is considered more "dirty" compared to the standard animal rooms in which regular mouse husbandry and experiments are performed.

Our group has previously encountered phenotypic differences between these two animal suites, and thus, always house control animals in the same suite for comparability. As an illustrative example, in the following experiment (reproduced from another project), wildtype mice were transplanted with 2% mutant-*DNMT3A* bone marrow + 98% wildtype-*DNMT3A* bone marrow. The recipient mice were then housed in either the standard SPF mouse facility (TMF, n=26) or in the biohazard suite (n=30). Intriguingly, mice housed in the biohazard suite had differing clonal chimerism behaviour compared to those housed in TMF (**Rebuttal Figure 1.2**). While the clone-specific trend observed in these data (i.e. the biohazard room correlates with greater *DNMT3A* clonal expansion) is the opposite to that observed in Figure 5, these data illustrate that housing facility can have a systemic effect on mice.

Rebuttal Figure R1.2: *DNMT3A* clone expansion post-transplant by animal facility. Wildtype mice aged 8-12 weeks were transplanted with 2% *DNMT3A* mutant bone marrow, then randomly housed in two animal facilities at Baylor College of Medicine. Peripheral blood was collected for mutant chimerism analysis at the timepoints indicated. Groups were compared by 2-way ANOVA and **** indicates $p < 0.0001$.

Within our data, we note that the control mice in Figure 5B (uninfected, biohazard suite) had on average 1.7 mutant clones detected, as did control mice in 5C (vehicle, TMF suite). Control mice in 5D (vehicle, TMF suite) had on average 1.3 mutant clones identified. In each experimental setting, mutant clones were statistically more frequent in mice that were exposed to infection (5B) or chemotherapy (5C/5D). It is possible that differences in sterility between control groups due to the different housing facilities could account for some of the differences, or other batch effects, although overall, across all genes tested, there were more frequent mutant clones present during infection/treatment despite the differences in the control group.

Lastly, while the “infected” controls were not administered *M. avium*, the entire cohort was housed in the biohazard facility with potentially different environmental milieu. Thus, we do not feel it would be appropriate to test control groups statistically against each other given the different laboratory conditions. That caveat aside, using a gene-level Mann-Whitney test (as described above in response #3), when control groups in 5B (biohazard room) and 5C (TMF) are compared, there is borderline enrichment for *Tet2* clones ($p=0.048$).

For the control groups in 5C vs. 5D, that were housed in the same facility, we can confirm that no genes were present at significantly different distributions using a non-parametric Mann-Whitney U test (that retains individual mouse-level mutant clone data), as described above in the response to Reviewer 1 #3.

To explicitly detail the different housing conditions of the *M. avium* experiment, we have added the following to the Methods (page 35 para 2): “For *Mycobacterium avium* infection, mice were infected with 2×10^6 colony-forming units of *M. avium* delivered intravenously as previously described⁹². **Infected and uninfected control mice were housed in a BSL-2 biohazard animal suite following infection.** Mice were infected once every 8 weeks (twice in total) to ensure chronic infection.”

Referee #2 (Remarks to the Author):

Thank you for the comprehensive revision, I have no further specific comments.

Very many thanks.

Referee #3 (Remarks to the Author):

I thank the authors for their replies. The responses to my specific comments are mostly satisfactory, although I feel that at certain points in the manuscript the authors still do not acknowledge prior literature in a sufficiently transparent way. In particular, I strongly agree with Reviewer 1 that the finding that HSCs and MPPs give rise to independent peripheral blood populations in a lifelong manner is not novel: this fact is described in great detail by Patel et al. in an endogenous (non-transplantation) setting and I feel that the authors' acknowledgement of this directly relevant prior work with the sentence: "In recent years, a more nuanced and dynamic picture has emerged, with the identification of additional self-renewing progenitor compartments" is simply insufficient. I worry that this wording will be interpreted as a deliberate downplaying of Patel et al.'s insights. In fact, this sentence is the only place in the relevant results section where Patel is acknowledged, and the authors end this narrative again by saying that their data challenge classical hematopoiesis models, without notifying the reader that the model has already been challenged in an identical manner by prior work. The discussion I find more transparent.

Our sincere apologies that our manuscript might give the impression that prior work is under acknowledged. Whilst we had cited Patel *et al.*'s work 5 times in the previous version (pages 3, 7, 16, and 17), we have now changed the wording in the above referenced section as follows (Page 5, para 4): The classic view of haematopoietic differentiation, based largely on transplantation studies, held that MPPs derived from HSCs^{25,26}. Recent barcoding and single-cell approaches in non-perturbed haematopoiesis suggest a more nuanced picture with multiple self-renewing progenitor populations^{2,4}. And on page 6, para 4: Overall, our data indicate that many long-term HSC and MPP lineages are established independently and in parallel during early development, and that MPPs do not always arise from HSCs, **contrary to classical views, but consistent with recent barcoding evidence^{2,4}**.

Perhaps more glaringly, in their rebuttal the authors have refused to address the concern that Chin et al.'s finding that the end-of-life mutation burden in single mouse HSCs is relatively low in comparison with humans is not acknowledged. Their argument is that "given the above paper sequences 6 single-cell derived whole genomes from (presumably) one 24-month mouse, we strongly believe that the data, whilst the first description of the difference between mutation burden in equivalently aged mouse/human, is arguably preliminary given the limited sample set." I think this response is highly problematic. As the authors themselves and many others have shown, mutation accumulation in HSCs is predictable and scales linearly with time with relatively limited variance among cells. It is unsurprising that major biological effects can be discerned with 6 colonies from one mouse – the authors have no reason to dismiss this finding. Furthermore, the authors do not fairly acknowledge prior measurements of CH in mice. Their citation of Chin et al.'s work in the introduction "with preliminary data suggesting a lower rate of CH" is an understatement of the fact that Chin et al have unambiguously demonstrated that CH is more limited in mice compared to humans.

We apologise for understating Chin *et al.*'s findings. This was not our intention. We have now endeavored to more clearly acknowledge the prior findings. We have removed the word "preliminary" on page 3 where Chin's work is cited in the introduction, "...with initial data showing

a lower rate of CH.¹¹". We also cite the paper in the Results section in relation to their important observation that transplantation enhances clonal expansions (page 9): "These data are consistent with a previous report identifying expanded clones in mice following bone marrow transplantation¹¹."

We fully agree that the data in Chin *et al.* provide important prior evidence of a lower rate of CH in mice. However, there are many differences between that report and our data that expand our understanding of murine native CH.

(i) The authors observed CH in two (2%) unperturbed C57BL/6 24-month mice using single-strand consensus sequencing. Our duplex sequencing method, which we estimate to be 10x more sensitive, identifies a much greater prevalence of low burden CH across unmanipulated mice of different ages. We identify low CH clones in 80% of unperturbed C57BL/6 mice (54/67). In fact, beyond 3 months of age, 91% of individual mice had evidence of CH (51/56).

(ii) The mutation burden reported by Chin *et al.* for 24-month mice is 33-100% greater than what we observe (200 mutations in Chin *et al.* versus 100-150 mutations in this study, for 24-month old mice). It is possible that these reflect batch-specific differences. Alternatively, variant calling approaches may have accounted for the differences. Our approach leveraged shared variants across colonies from the same individual, and mice of different ages, to enhance removal of germline SNPs and sequencing artifacts.

(iii) Chin *et al.* report that the mouse HSC mutation rate is 8.5-fold higher than human HSCs, derived from a single mouse age. Our data, using >1500 whole genomes across >20 individual mice of different ages (1 day old to 30 months), provides greater sensitivity and identifies that the murine HSC mutation rate is just under 3-fold higher than human HSCs.

(iv) Our study explores the impact of several different haematopoietic exposures and insults on the landscape of CH, many of which are similar to what humans are exposed to. 97% of animals (29/30) mice experiencing different haematopoietic insults harboured CH, and often at larger clone sizes, in contrast to 25% of transplanted mice in Chin *et al.*

Taken together with our additional data on (i) ontogeny and timing of specification of HSC versus MPP, (ii) their similar contributions to peripheral blood, (iii) the HSC number and cell division rates in mouse (which we validate orthogonally), and (iv) the fitness of CH drivers in mice versus human, we do believe our study adds substantial depth and breadth, as well as novelty, to our understanding of the clonal landscape and population dynamics of haematopoiesis in the laboratory mouse.

We hope that we have now clearly acknowledged the important contribution of Chin *et al.*

Overall, no technical issues with this paper remain, but the acknowledgement of prior work is still inadequate in the places outlined above. The most critical aspect raised in my original review – a relative lack of novelty which was also clearly articulated by Reviewer 1 – has not substantially changed in my assessment.